# Beyond the Exploration-Exploitation Trade-off: A Hidden State Approach for LLM Reasoning in RLVR

## Abstract

A prevailing view in Reinforcement Learning with Verifiable Rewards (RLVR) interprets recent progress through the lens of an exploration-exploitation trade-off, a perspective largely shaped by token-level metrics. We re-examine this perspective, proposing that this perceived trade-off may not be a fundamental constraint but rather an artifact of the measurement level. To investigate this, we shift the analysis to the semantically rich hidden-state space, adopting Effective Rank (ER) to quantify exploration and proposing its novel first- and second-order derivatives, named ER Velocity and ER Acceleration, to capture exploitation dynamics. Our analysis reveals that in the semantic space, exploration and exploitation could be *decoupled* (Sec. 4). This finding reveals an opportunity to enhance both capacities simultaneously. This insight motivates our method, Velocity-Exploiting Rank-Learning (VERL), the *first* to operationalize the principle of synergistic exploration-exploitation enhancement by directly shaping the RL advantage function. The key innovation is leveraging the theoretically stable ERA as a predictive meta-controller to create a synergistic, dual-channel incentive structure. Instead of forcing a trade-off, VERL prospectively amplifies rewards for exploration to preempt overconfidence and reinforces exploitative gains to consolidate reasoning. Experiments across diverse LLMs and reasoning benchmarks show consistent gains, including up to 21.4% absolute accuracy improvement on the challenging Gaokao 2024 dataset. The code is available at `https://anonymous.4open.science/r/coding`.

## 1 Introduction

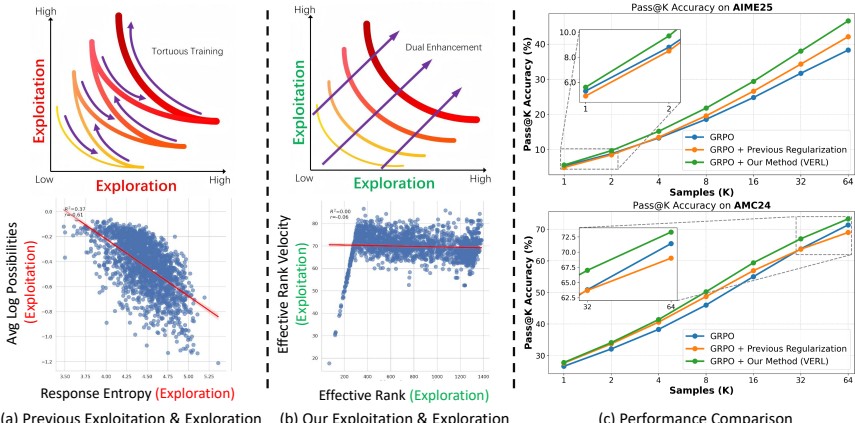

Figure 1: Comparative analysis with the responses of DeepSeek-R1-Distill-Qwen-7B in simpleRL test dataset (Zeng et al., 2025). (a) Traditional metrics for exploitation & exploration constrained by negative coupling, leading to meandering progress for both capabilities. (b) Our metrics are mutually independent. (c) Training regularization with our metrics demonstrates stronger performance in both exploitation (small K) and exploration (large K).

Recent advancements in Reinforcement Learning with Verifiable Rewards (RLVR) have significantly improved the reasoning abilities of Large Language Models (LLMs). A dominant narrative emerging from these recent works (Chen et al., 2025b; Yue et al., 2025; Deng et al., 2025a; Agarwal et al., 2025) interprets this progress through the lens of balancing *exploration* (searching for diverse reasoning paths) and *exploitation* (refining the most promising known strategies). However, this paradigm is almost exclusively rooted in a token-level analysis in action space, where exploration is captured by high-entropy token distributions and exploitation by high-confidence, low-entropy ones. This has inevitably led to the widespread assumption of an inherent trade-off between the two, as the token-level output distribution, which only reflects the model's hesitation about the next-token prediction, is seen as unable to be simultaneously uniform and sharp.

While convenient, this token-centric viewpoint introduces significant limitations. Equating exploration with mere token-level entropy faces an intrinsic dilemma (Fu et al., 2025; Qiao et al., 2025; Agarwal et al., 2025): excessively high entropy risks generating incoherent noise, while overly low entropy stifles exploration it aims to encourage. Similarly, defining exploitation via hand-crafted heuristic rewards (Chen et al., 2025a; Li et al., 2025a; Bensal et al., 2025) produces brittle models with poor generalizability as they simply learn to chase surface-level proxies. More fundamentally, these token-level proxies are misaligned with how reasoning actually happens (Wei et al., 2022; Yao et al., 2023): solutions emerge over multi-token *semantic chunks* (concepts, subgoals), not isolated tokens, and a single token cannot correspond to a meaningful greedy decision about a reasoning strategy. More related works are discussed in Sec. D. While many works (Cheng et al., 2025a; Deng et al., 2025b) are aware to consider both exploration and exploitation as in Fig. 1a, their continued reliance on token-level metrics invariably traps them in a cycle of "balancing" the trade-off, instead of doubting its existence. This raises a critical question: *Is the exploration–exploitation trade-off intrinsic to reasoning, or merely an artifact of token-level measurement?*

To answer this, we move beyond token-level statistics and study exploration and exploitation directly in the semantically rich hidden-state space of response-level trajectories, where prior work has shown that transformer representations encode meaningful linguistic and reasoning structure (Jing et al., 2025; Sajjad et al., 2022; Valeriani et al., 2023; Matthews et al., 2024; Zhang et al., 2025). At this level, we define exploration and exploitation : Effective Rank (ER) measures how broadly a hidden-state trajectory spreads across semantic directions, corresponding to representation-level exploration, while its temporal derivative, Effective Rank Velocity (ERV), measures how the same trajectory refines semantic content along its path, corresponding to representation-level exploitation. Concretely, we are the first to apply ER in an RL context and use it to quantify exploration by measuring the semantic diversity of hidden-state representations: high ER indicates that the model is activating diverse semantic directions and widening its search over possible solutions. To capture exploitation more precisely, which we define as the efficiency with which a trajectory converges toward a solution in representation space, we further introduce Effective Rank Acceleration (ERA), the second-order temporal change of ER, which captures the trend of the velocity, indicating whether reasoning is accelerating toward a solution or saturating in a stagnant regime. Equipped with these semantic-trajectory tools, we uncover a striking result: in the semantic space, exploration and exploitation exhibit near-zero correlation (Fig. 1b, bottom). This contrast provides strong evidence that the trade-off is not an inherent property of RLVR for reasoning but an artifact of biased token-level measurements. It further reveals that these two capacities are not fundamentally antagonistic but can, in fact, *be decoupled and enhanced simultaneously* (Fig. 1c).

Building on this core insight, we propose Velocity-Exploiting Rank-Learning (VERL), a method that moves beyond the trade-off between the two capacities by directly shaping the RL advantage function using ER and ERV. Instead of acting as a switch between the two capacities in lower dimension, VERL functions as a *tuner* synergistically enhances both capacities in higer dimension. Its key innovation is leveraging ERA as a meta-control variable, a choice justified by our theoretical proof of its remarkable $\mathcal{O}(1)$ growth stability (Sec. 3). Specifically, VERL uses ERA as a dynamic signal to enhance the training incentives; Specifically, VERL uses ERA to create a synergistic, dual-channel incentive structure. Instead of switching between modes, it prospectively shapes the reward to simultaneously encourage exploration (via ER) to preempt overconfidence, while also reinforcing exploitative gains (via ERV) to consolidate the reasoning path. This unique stability makes ERA a robust signal to guide training, allowing VERL to simultaneously encourage exploration from productive-potential states while preventing overfitting to local optima. As a result, VERL delivers significant performance gains across diverse models and tasks, achieving up to a **21.4%** absolute accuracy improvement on the challenging Gaokao 2024 benchmark.

**Contributions.** *(i)* We are the first to probe the exploration-exploitation relationship in the semantically rich hidden-state space. By adopting "ER" to quantify exploration and proposing novel metrics "ERV" and "ERA" for exploitation, we empirically demonstrate that these two capacities are decoupled, moving beyond the conventional token-level trade-off. *(ii)* We present VERL, a method that leverages ERA to manage exploration and exploitation in a unified framework, enabling the simultaneous enhancement of both capabilities. *(iii)* Our extensive experiments demonstrate the efficiency, generality, and versatility of VERL across different RL architectures.

## 2 PRELIMINARIES

### 2.1 PROBLEM FORMULATION AND NOTATIONS

We adopt a reinforcement learning perspective on training LLMs for reasoning tasks. The LLM is modeled as a policy $\pi_\theta(\cdot|x)$, parameterized by $\theta$, which generates a reasoning trajectory for a given prompt $x$ sampled from a distribution $\mathcal{P}_x$. The model's output is a sequence of reasoning steps $y_{0:T} = (y_0, y_1, \ldots, y_T)$, constrained to a maximum length $L_{\max}$ (*i.e.*, $T < L_{\max}$). The quality of this trajectory is evaluated by a scalar reward function $r(x, y)$. The objective is to find the optimal policy $\pi_\phi$ that maximizes the expected reward:

$$\phi = \arg\max_\theta \mathbb{E}_{x \sim \mathcal{P}_x} \mathbb{E}_{y \sim \pi_\theta(\cdot|x)} [r(x, y)] \text{ s.t. } |y| \leqslant L_{\max}. \tag{1}$$

Conventionally, optimizing this objective at the token level is framed as a fundamental exploration-exploitation trade-off. The policy must explore diverse and potentially novel reasoning pathways to discover high-reward solutions. Concurrently, it must exploit known strategies by reinforcing correct and reliable reasoning patterns that consistently yield high rewards.

### 2.2 REINFORCEMENT LEARNING BASELINE FRAMEWORKS

**Proximal Policy Optimization (PPO)** from Schulman et al. (2017) is a standard RL algorithm that seeks to maximize a clipped surrogate objective function. This objective prevents excessively large changes that would destabilize training, which is defined as:

$$\mathcal{L}_{\text{PPO}}(\theta) := \mathbb{E}_{x \sim \mathcal{P}_x, y \sim \pi_{\theta_{\text{old}}}(y|x)} \left\{ \sum_{t=1}^{|y|} \min \left[ \rho_t(\theta) A_t, \text{clip}(\rho_t(\theta), 1 - \epsilon_{\text{low}}, 1 + \epsilon_{\text{high}}) A_t \right] \right\}, \tag{2}$$

where $\rho_t(\theta) := \frac{\pi_\theta(y_t|x, y_{<t})}{\pi_{\theta_{\text{old}}}(y_t|x, y_{<t})}$ is the probability ratio between the current and old policies, and $A_t$ is the estimated advantage, often calculated using Generalized Advantage Estimation (GAE) from Schulman et al. (2015), with clipping (hyperparameter $\epsilon$) to mitigate excessive deviation.

**Group Relative Policy Optimization (GRPO)** from Shao et al. (2024) computes a baseline directly from the rewards of multiple trajectories. For a given prompt, it samples a group of G responses, obtains their corresponding rewards $\{r_1, ..., r_G\}$, and normalizes these rewards to compute the advantage for each response:

$$A_{i,t} := \frac{r_i - \text{mean}(\{r_j\}_{j=1}^G)}{\text{std}(\{r_j\}_{j=1}^G)}. \tag{3}$$

GRPO would assign a single rule-based reward to the entire output sequence, and the resulting group-relative advantage is uniformly propagated to all tokens, then updated as in Eq. 2.

### 2.3 CHARACTERIZING HIDDEN STATE REPRESENTATIONS

**Response Hidden States.** LLM would generate responses token by token in an autoregressive manner. The token $y_t$ output at step $t$ represents the current explicit state, while the corresponding output in the intermediate layers is referred to as the hidden state $z_t$. As the sequence of explicit states forms the final response, simultaneously, the hidden states $\{z_t\}_{t=1}^T$, ordered by their output sequence, collectively form the hidden states matrix $\mathbf{Z} \in \mathbb{R}^{T \times D}$, where $T$ is the output length and $D$ is the feature dimension, representing the semantic trajectory. To align with the semantic space, we focus on the hidden state of the final layer in this paper.

**Dataset Hidden States.** Following the definition in Skean et al. (2025), for a dataset containing $N$ prompts, after obtaining a single vector representation for the $i$-th response by averaging its token hidden states $\bar{z}_i = \frac{1}{T_i} \sum_{t=1}^{T_i} z_{i,t}$, we stack these $N$ mean embeddings in the dataset hidden states

matrix $\bar{\mathbf{Z}} \in \mathbb{R}^{N \times D}$, to represent the overall semantic distribution of the entire dataset.

## 3 A HIDDEN-STATE PERSPECTIVE ON REPRESENTATIONAL DYNAMICS

### 3.1 STATIC METRIC: EFFECTIVE RANK (ER)

According to Roy & Vetterli (2007), the Effective Rank (ER), which is denoted by $\mathrm{erank}(\mathbf{Z})$ for a response, is computed based on the normalized singular values of its non-padding hidden states $\mathbf{Z}$. Let $\sigma_j$ be the $j$-th singular value of $\mathbf{Z}$, and $p_j = \frac{\sigma_j}{\sum_k \sigma_k}$ be the normalized singular values. The ER is then given by:

$$\mathrm{ER} := \mathrm{erank}(\mathbf{Z}) = \exp\left[ -\sum_j p_j \log(p_j) \right]. \tag{4}$$

To quantify a model's reasoning breadth, we treat exploration as a measurable semantic property. Our primary metric for this is ER, which measures the **effective dimensionality** of the hidden-state space a model occupies during a response. A high ER signals that the model is leveraging a rich and diverse set of internal features, which is a direct signature of exploratory behavior. A low ER, in contrast, points to a collapsed, simpler representation, indicating the model is not exploring widely. By capturing this dispersion of embeddings, ER provides a more nuanced view of exploration than conventional rank, which merely counts dimensions without considering their diversity.

**Theorem 3.1.** *Suppose we have a matrix of embeddings $\mathbf{Z} \in \mathbb{R}^{T \times D}$. Then the ER of $\mathbf{Z}$ is a lower bound of conventional rank of $\mathbf{Z}$:*

$$1 \leqslant \mathrm{erank}(\mathbf{Z}) \leqslant \mathrm{rank}(\mathbf{Z}) \leqslant min\{T, D\}. \tag{5}$$

**Remark 3.2.** Conventional rank offers a discrete count of available dimensions but fails to capture the geometric complexity essential for true exploration. In contrast, ER provides a nuanced, continuous measure of "effective dimensionality". In reasoning, this distinction is critical: conventional rank may count many potential paths, but ER reveals how uniformly the model is actually exploring them. A high ER reflects a more uniform distribution, signaling a broader and more effective exploration of the solution space.

### 3.2 DYNAMIC METRICS: EFFECTIVE RANK VELOCITY (ERV) AND ACCELERATION (ERA)

In this section, we develop temporal higher-order metrics, termed ERV and ERA, to characterize the dynamics of a policy's information gain together. Corresponding to the first and second-order temporal differences of a representational metric, these tools measure how the quality of hidden states evolves, revealing whether the reasoning process is expanding, stabilizing, or saturating.

**Definition 3.3. (First-Order Temporal Difference: ERV)** *To quantify the rate of change for a given metric $M$, such as the ER or the conventional rank of the hidden states matrix, we define the first-order temporal difference, denoted $\Delta_M^{(1)}$. This metric captures the "velocity" of information gain by measuring how the metric's value at a given step deviates from its historical average. Let $m_t$ be the value of metric $M$ computed on the token sub-sequence from the start to position $t$. For a sequence of length $T$ and a difference stride $s$, let the set of evaluation time steps be $\mathcal{T} = \{s, 2s, \ldots, Ks\}$, where $K = \lfloor (T-1)/s \rfloor$. The overall first-order difference is defined as:*

$$\Delta_M^{(1)} := \frac{1}{K-1} \sum_{j=2}^{K} \delta_{j \cdot s}, \quad \text{where} \quad \delta_{j \cdot s} := m_{j \cdot s} - \frac{1}{j-1} \sum_{k=1}^{j-1} m_{k \cdot s}. \tag{6}$$

*Equivalently, writing the consecutive-step increments as $\Delta m_{r \cdot s} := m_{r \cdot s} - m_{(r-1) \cdot s}$, a simple algebraic rearrangement yields*

$$\delta_{j \cdot s} = \frac{1}{j-1} \sum_{r=2}^{j} (r-1) \, \Delta m_{r \cdot s}, \quad j \geqslant 2, \tag{7}$$

*showing that each $\delta_{j \cdot s}$ is in fact a time-weighted average of local consecutive differences, with larger weights assigned to more recent steps.* Our primary metric for exploitation is ERV, which is designed to capture the rate of information gain. It is the average of instantaneous differences ($\delta_{j \cdot s}$), where

each difference contrasts the complexity of the current token chunk with the cumulative average of all preceding ones. This formulation directly operationalizes our definition of exploitation: a large ERV demonstrates that the model is successfully enriching its representation at a rate that outpaces its historical trend, signifying a deepening and productive line of inquiry. Conversely, a small ERV signals that exploitation of the current path is becoming less effective.

**Definition 3.4. (Second-Order Temporal Difference: ERA)** *To measure the rate of change of the velocity, we define the second-order temporal difference $\Delta_M^{(2)}$, which represents the "acceleration" of the metric's evolution. It reveals whether the process of representation formation is speeding up or stabilizing. It is computed as the average change between consecutive instantaneous differences:*

$$\Delta_M^{(2)} := \frac{1}{K-2} \sum_{j=3}^{K} [\delta_{j \cdot s} - \delta_{(j-1) \cdot s}]. \tag{8}$$

A positive $\Delta_M^{(2)}$ signifies an accelerating growth rate, indicating that the diversification of the representation is speeding up. A negative value suggests this growth is decelerating, implying that the representation's quality is approaching stability or saturation.

### 3.3 SCALING PROPERTIES OF REPRESENTATIONAL DYNAMICS

In the preceding sections (Sec. 3.1 and 3.2), we introduced metrics for analyzing the hidden states of individual responses. We now analyze the scaling properties of these dynamics at two distinct levels of granularity: across an entire dataset as a function of its size ($N$), and within a single reasoning trajectory as a function of its length ($T$). The following proposition provides a unified theoretical model for both scenarios.

**Proposition 3.5.** *Assume a hidden-state matrix is composed of $k$ approximately orthogonal row vectors. The Effective Rank (ER) and its first-order difference (ERV) scale linearly with $k$, such that $ER = \mathcal{O}(k)$ and $\Delta_{ER}^{(1)} = \mathcal{O}(k)$. The second-order difference (ERA) is independent of $k$, with a scaling order of $\Delta_{ER}^{(2)} = \mathcal{O}(1)$.*

**Remark 3.6.** This proposition offers a dual interpretation of how our metrics scale under ideal conditions: **At the dataset level**, $k$ represents the number of questions $N$. The proposition implies that as a dataset grows with semantically distinct responses (approaching orthogonality), its overall representational diversity (ER) should increase proportionally. The constant acceleration (ERA) suggests a stable, predictable growth pattern for the dataset's semantic volume. **At the response level**, $k$ represents the sequence length $T$. The proposition suggests that for an ideal reasoning process where each step contributes novel information (making token embeddings approach orthogonality), the trajectory's semantic complexity (ER) and information-gain velocity (ERV) should also grow linearly with its length. In this context, a constant ERA becomes a signature of a robust and non-saturating reasoning process.

## 4 DECOUPLING EXPLORATION AND EXPLOITATION IN REASONING

In this section, we first investigate the changing trends of the hidden states matrix rank (both ER and conventional rank) during regular RL training. Specifically, we utilized the Qwen (Hui et al., 2024) and Llama (Dubey et al., 2024) models for our experiments, employing GRPO (Shao et al., 2024) reinforcement learning paradigm. The training dataset followed the configuration in Zeng et al. (2025), which comprises 8k hard-level 3 to 5 mathematical problems from MATH datasets, each accompanied by a verifiable reference answer.

### 4.1 ANALYSIS OF RESPONSE-LEVEL METRICS

During each training step, we quantitatively analyzed the representational dynamics of hidden states within that batch as depicted in Fig. 2, and provided more and diverse details in App. H.1.

**Semantic space of hidden states move beyond the exploration-exploitation trade-off towards stable enhancements.** While RL consistently improves performance, it interacts differently with distinct base models, evidenced in the divergent trends of the ER (first column in Fig. 2), which measures the total information within a response. For instance, the Qwen model exhibits an increasing ER, suggesting more exploratory reasoning, whereas the one of Llama model decreases,

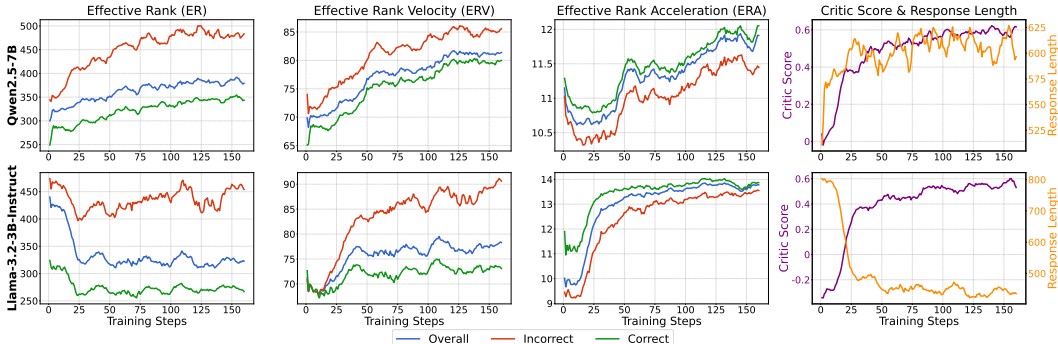

Figure 2: Response-level metrics during GRPO post-training, smoothed with a 10-step rolling window. Metrics are shown for the Overall batch, as well as for subsets of Correct and Incorrect samples. The rightmost column displays the average Critic Score (reward) and Response Length per batch.

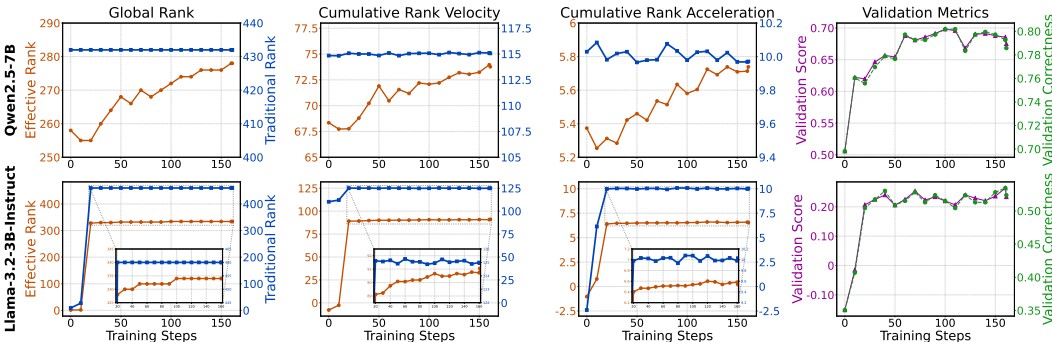

Figure 3: Visualization of dataset-level metrics during GRPO post-training. The figure compares Traditional metrics with our proposed metrics. Also shown are the Validation Score and sample Correctness, both averaged over the validation dataset.

indicating more concise, exploitative reasoning. Despite these differences in exploratory behavior, the ERV (second column) demonstrates a consistent upward scaling trend for all models, which suggests while the models' intrinsic exploratory tendencies differ, RL fine-tuning universally enhances their exploitation capabilities by consistently accelerating the rate of information gain.

**ERA distinguishes correct reasoning.** For both the zero-order metric (ER) and the first-order metric (ERV), incorrect reasoning paths consistently score higher than correct ones. This suggests that excessive exploration (high ER) with new excessive information (high ERV) will potentially derail the reasoning process and lead to incorrect answers. Conversely, for the second-order metric (ERA), correct reasoning trajectories consistently exhibit higher values, which implies the acceleration of information gain—the ability to increasingly speed up the representational evolution—is the key to guide the policy towards a correct solution, distinguishing robust reasoning from flawed exploration.

### 4.2 ANALYSIS OF DATASET-LEVEL METRICS

Following the framework established in Sec. 2, we extend our analysis from the response level to the entire validation dataset. By computing the dataset hidden states matrix $\overline{\mathbf{Z}}$, we examine its zero-, first-, and second-order rank dynamics to understand how the policy's overall representational space evolves. The trends are visualized in Fig. 3. While key experiments are shown here, we refer the reader to App. H.2 for a more diverse range of studies.

**Policy optimization correlates with expanding dataset-level diversity.** Across the training process, we observe a strong positive correlation between performance metrics (accuracy and critic score on the validation set) and the dynamics of dataset-level ER. As the model improves, the zero-order $\mathrm{erank}(\overline{\mathbf{Z}})$ and its first- and second-order differences consistently scale up. This indicates that as the policy is updated, it develops a more diverse and complex repertoire of reasoning strategies for the same set of problems. The increasing ERV and ERA suggest the model becomes progressively more efficient at navigating and expanding this richer semantic space to discover correct solutions.

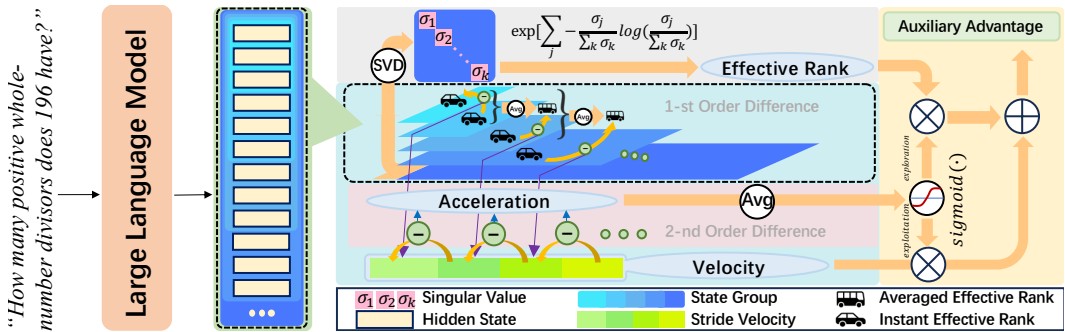

Figure 4: Overview of VERL. Exploration is quantified by computing the ER of the rolling-done hidden states via SVD, while exploitation is captured through EMA-smoothed first-order difference (ERV) on per-step rolling hidden state and extended to second-order difference (ERA). Finally, exploration and exploitation are adaptively integrated to derive the auxiliary advantage.

**ER reveals refinement beyond the limits of conventional rank.** During late-stage training, a plateauing conventional rank suggests the model has settled on a fixed number of linearly independent reasoning "directions". Yet, a simultaneously rising ER points to a more subtle optimization. This trend reveals that the model is improving the quality of its existing solution space by making the "magnitudes" of these directions more uniform. In essence, instead of finding new pathways, the model learns to utilize its established ones more equitably, reducing representational redundancy and fostering a more sophisticated and distributed reasoning capability.

## 5 VELOCITY-EXPLOITING RANK-LEARNING (VERL)

Building upon the insights from Sec. 3 and the empirical observations in Sec. 4, we find that conventional RL objectives overlook the intrinsic hidden-state dynamics that more faithfully characterize exploration and exploitation. This oversight may lead to inefficient training, where policies either wander in unproductive exploration or collapse prematurely to suboptimal reasoning paths. To address this, we propose a novel method named Velocity-Exploiting Rank-Learning (VERL), which refines advantage by incorporating the nuanced dynamics of hidden states, enabling simultaneous enhancement of exploration and exploitation capacities.

### 5.1 STABLE REPRESENTATION DEVIATION INDICATOR

Concretely, we first formalize the representational metrics, letting $\mathcal{M} = \{M_0, M_1, M_2\}$ denote the set of metrics derived from the hidden states, where $M_0$ is the 0-order ER, $M_1 := \Delta_M^{(1)}$ is its first-order temporal difference (ERV), and $M_2 := \Delta_M^{(2)}$ is its second-order temporal difference (ERA). To create a stable guidance signal, having already computed scalar values $\{m_0, m_1, m_2\}$ for each trajectory, we normalize these values against their historical trends by maintaining an Exponential Moving Average (EMA) $\bar{\mu}_k$ for each metric $M_k$. The relative deviation for each metric is:

$$d_k := \frac{m_k - \bar{\mu}_k}{|\bar{\mu}_k| + \epsilon}, \quad k \in \{0, 1, 2\}, \tag{9}$$

where $\epsilon$ is a small constant for numerical stability. This deviation $d_k$ quantifies how the current trajectory's representational structure diverges from the policy's recent average behavior.

### 5.2 BEYOND TRADE-OFF FOR EFFICIENT TRAINING VIA ERA

Our analysis in Sec. 3.1 and 3.2 reveals that ER ($M_0$) and ERV ($M_1$) are effective proxies for exploration and exploitation, respectively. Crucially, these two metrics are also almost independent of each other, as shown in Fig. 1c bottom. This decoupling is key, as it allows us to combine them into a single objective to enhance both capabilities simultaneously. $M_1$ measures the confidence of the current step as analysis above, so the subtraction of $M_1$, namely ERA ($M_2$), would predict the evolution of confidence in subsequent steps. Meanwhile, theoretical analysis (Prop. 3.5) and empirical observations (Fig. 3) indicate that $M_2$ remains approximately constant across trajectories. Thus, $M_2$ can serve as a meta-level signal to guide training.

An increasing ERV indicates that the model is acquiring progressively more informative evidence, reflecting its growing confidence. However, our preliminary experiments reveal that trajectories exhibiting excessively high confidence often achieve high in-domain performance (as evidenced by the results on the MATH dataset in Tab. 3, where performance suffers without ERA's dynamic unification or when using a simple 50/50 sum "$\beta = 0.5$".) while compromising out-of-domain generalization (across most datasets), suggesting severe overfitting. This implies that overconfident trajectories reduce the opportunity to learn from less confident yet potentially informative samples. To mitigate this, we employ $M_2$ as the predictive signal to combine exploration and exploitation, strategically encouraging exploring lower-confidence samples when trajectories exhibit excessively high future confidence, thereby enhancing training efficiency and robustness.

Specifically, we define two orthogonal unit vectors of the weights first, an exploration-focused vector $\mathbf{w}_{\text{explore}} = [1, 0]$, which targets $M_0$, and an exploitation-focused one $\mathbf{w}_{\text{exploit}} = [0, 1]$ of $M_1$. The dynamic weight $\mathbf{w}_{\text{dyn}}$ is interpolated by the relative deviation of the second-order metric $d_2$:

$$\mathbf{w}_{\text{dyn}} := \beta \cdot \mathbf{w}_{\text{explore}} + (1 - \beta) \cdot \mathbf{w}_{\text{exploit}}, \quad \text{where} \quad \beta := \text{sigmoid}(d_2). \tag{10}$$

The interpolation coefficient $\beta$ is adaptively determined by the second-order metric $d_2$ through a sigmoid mapping. A high $M_2$ ($d_2 \gg 0$) means overconfidence in the future, risking overfitting to in-domain patterns; thus, VERL increases $\beta$ to favor the exploration profile $M_0$. In contrast, when $M_2$ is low ($d_2 \leqslant 0$), namely limited confidence and reasoning saturation, VERL decreases $\beta$ to emphasize the exploitation profile $M_1$. As $M_2$ typically fluctuates around zero, VERL jointly enhances exploration and exploitation. The final auxiliary advantage $\Phi$ is defined as:

$$\Phi := w_{\text{dyn},0} \cdot \tanh(d_0) + w_{\text{dyn},1} \cdot \tanh(d_1), \tag{11}$$

where $w_{\text{dyn},0}$ and $w_{\text{dyn},1}$ are the first and second entries of the dynamic weight vector $\mathbf{w}_{\text{dyn}}$, respectively. The $\tanh$ function bounds the magnitude of $\Phi$ while preserving its sign, thereby stabilizing training. This formulation rewards trajectories that exceed the historical average and penalizes those that fall short, guiding the policy with adaptive reasoning dynamics while mitigating risks of stagnation and overconfidence.

## 5.3 Advantage Shaping via Representational Dynamics

We refine the policy learning signal by shaping the advantage through a representational auxiliary term. Let $A^{(0)}$ denote the original advantage from GRPO or PPO with GAE, and let $\Phi_i$ be the sequence-level auxiliary signal defined in Sec. 5.2. The shaped objective replaces the original advantage with a refined estimate $\hat{A}_t$, defined directly within the surrogate loss:

$$\mathcal{L}^{\text{shaped}}(\theta) = \mathbb{E}_{x \sim \mathcal{P}_x, \, y \sim \pi_{\theta_{\text{old}}}} \left[ \sum_{t=1}^{|y|} \min\left( \rho_t(\theta) \, \hat{A}_t, \, \text{clip}(\rho_t(\theta), 1 - \epsilon, 1 + \epsilon) \, \hat{A}_t \right) \right],$$

$$\text{where} \qquad \hat{A}_t = A_t^{(0)} + \min\left( \max(0, \Phi_i), \, \frac{|A_t^{(0)}|}{\kappa} \right), \, \rho_t(\theta) = \frac{\pi_\theta(y_t \mid x, y_{<t})}{\pi_{\theta_{\text{old}}}(y_t \mid x, y_{<t})}. \tag{12}$$

This formulation adds a strictly positive, clipped bonus to trajectories exhibiting desirable representational dynamics, while preserving the sign and stability of the original advantage. The mechanism applies consistently across advantage structures: for GRPO the shaping is applied once per trajectory, and for PPO+GAE it applies to every token within the trajectory.

## 6 Experiments

### 6.1 Experiment Settings

*(i)* **Dataset.** The same datasets as in Sec. 4 are used. *(ii)* **Reward.** Our rule-based reward function assesses mathematical correctness and `\boxed{}` formatting. Correct answers receive a +1.0 reward if formatted, and +0.5 if not. Incorrect answers are penalized with -0.5 if formatted and -1.0 otherwise. *(iii)* **Training.** Based on verl (Sheng et al., 2025) and vLLM (Kwon et al., 2023) framework, we set batch size as 48, generating 4 rollouts per prompt for GRPO and 1 rollout for PPO, and set the maximum length $L_{\max}$ to 1536. More details of the experiments are provided in App. G.

Table 1: Performance comparison of models on mathematical reasoning benchmarks (Pass@1). "+ GRPO" and "+ PPO" denote RL fine-tuning by GRPO and PPO framework respectively. "w/ VERL." indicates incorporating our VERL with original RL type. Δ represents the performance contrast between original RL method and its VERL variant. See App. H.3 for full details.

| Model | AIME24 | AIME25 | AMC23 | AMC24 | ASDiv | Carp_En | CMATH | Gaokao 2024_I | Gaokao 2024_Mix | Gaokao MathCloze | GSM8K | MAWPS | Olympiad Bench | SVAMP | TabMWP | Avg. |
|---|---|---|---|---|---|---|---|---|---|---|---|---|---|---|---|---|
| **Llama-3.2-3B-Instruct** | 0.0 | 0.0 | 25.0 | 11.1 | 74.6 | 26.5 | 10.2 | 14.3 | 14.3 | 6.8 | 66.6 | 86.9 | 12.7 | 74.1 | 41.4 | 31.0 |
| + GRPO | 3.3 | 0.0 | 27.5 | 8.9 | 88.8 | 45.0 | 28.3 | 21.4 | 20.9 | 23.7 | 80.7 | 96.0 | 16.7 | 87.7 | 71.7 | 41.4 |
| + GRPO w/ VERL. | 13.3 | 6.7 | 25.0 | 11.1 | 89.3 | 45.4 | 46.2 | 14.3 | 22.0 | 22.9 | 81.7 | 96.0 | 17.6 | 87.8 | 72.3 | 43.4 |
| Δ_GRPO | +10.0 | +6.7 | -2.5 | +2.2 | +0.5 | +0.4 | +17.9 | -7.1 | +1.1 | -0.8 | +1.0 | +0.0 | +0.9 | +0.1 | +0.6 | +2.0 |
| + PPO | 10.0 | 3.3 | 22.5 | 13.3 | 87.9 | 46.4 | 21.2 | 7.1 | 16.5 | 20.3 | 81.4 | 95.5 | 17.8 | 86.8 | 71.0 | 40.1 |
| + PPO w/ VERL. | 10.0 | 3.3 | 25.0 | 11.1 | 88.7 | 46.0 | 30.7 | 14.3 | 19.8 | 27.1 | 82.9 | 95.7 | 17.3 | 85.8 | 71.3 | 41.9 |
| Δ_PPO | +0.0 | +0.0 | +2.5 | -2.2 | +0.8 | -0.4 | +9.5 | +7.2 | +3.3 | +6.8 | +1.5 | +0.2 | -0.5 | -1.0 | +0.3 | +1.9 |
| **Qwen2.5-7B** | 6.7 | 0.0 | 45.0 | 15.6 | 91.4 | 55.8 | 86.7 | 42.9 | 33.0 | 49.2 | 85.8 | 95.4 | 25.8 | 88.5 | 82.8 | 53.6 |
| + GRPO | 10.0 | 6.7 | 55.0 | 26.7 | 94.8 | 60.2 | 91.7 | 14.3 | 34.1 | 64.4 | 90.2 | 97.6 | 36.1 | 92.8 | 91.3 | 57.7 |
| + GRPO w/ VERL. | 13.3 | 10.0 | 50.0 | 28.9 | 95.0 | 60.8 | 90.7 | 35.7 | 35.2 | 69.5 | 89.2 | 97.7 | 35.4 | 92.9 | 91.9 | 59.8 |
| Δ_GRPO | +3.3 | +3.3 | -5.0 | +2.2 | +0.2 | +0.6 | -1.0 | +21.4 | +1.1 | +5.1 | -1.0 | +0.1 | -0.7 | +0.1 | +0.6 | +2.1 |
| + PPO | 6.7 | 3.3 | 50.0 | 33.3 | 94.9 | 59.6 | 89.8 | 28.6 | 31.9 | 63.6 | 89.1 | 97.3 | 36.1 | 92.8 | 90.8 | 57.9 |
| + PPO w/ VERL. | 10.0 | 6.7 | 52.5 | 33.3 | 94.8 | 60.0 | 90.3 | 28.6 | 34.1 | 66.9 | 90.2 | 97.8 | 36.1 | 92.5 | 90.6 | 59.0 |
| Δ_PPO | +3.3 | +3.3 | +2.5 | +0.0 | -0.1 | +0.4 | +0.5 | +0.0 | +2.2 | +3.3 | +1.1 | +0.5 | +0.0 | -0.3 | -0.2 | +1.1 |

Table 2: Performance comparison of instruction-tuned models under diverse decoding settings (Pass@$k$). For full details, please refer to App. H.4.

| Model | MATH500 (Pass@16) | AMC23 (Pass@128) | AMC24 (Pass@128) | AIME24 (Pass@256) | AIME25 (Pass@256) | Avg. |
|---|---|---|---|---|---|---|
| **Llama-3.2-3B-Instruct** | 79.8 | 93.5 | 51.5 | 40.0 | 30.0 | 58.96 |
| + GRPO | 80.2 | 95.4 | 60.6 | 40.0 | 30.0 | 61.24 |
| + GRPO w/ VERL. | 80.6 | 95.7 | 59.0 | 50.0 | 36.7 | 64.40 |
| Δ_GRPO | +0.4 | +0.3 | −1.6 | +10.0 | +6.7 | +3.16 |
| + PPO | 82.2 | 94.5 | 57.0 | 46.7 | 36.7 | 63.42 |
| + PPO w/ VERL. | 82.4 | 94.7 | 57.8 | 46.7 | 40.0 | 64.32 |
| Δ_PPO | +0.2 | +0.2 | +0.8 | +0.0 | +3.3 | +0.90 |
| **Qwen2.5-7B** | 90.6 | 98.4 | 73.7 | 60.0 | 60.0 | 76.54 |
| + GRPO | 90.8 | 97.8 | 78.3 | 56.7 | 50.0 | 74.72 |
| + GRPO w/ VERL. | 91.4 | 98.3 | 79.0 | 63.3 | 60.0 | 78.40 |
| Δ_GRPO | +0.6 | +0.5 | +0.7 | +6.6 | +10.0 | +3.68 |
| + PPO | 91.2 | 98.6 | 74.3 | 53.3 | 56.7 | 74.82 |
| + PPO w/ VERL. | 91.4 | 98.0 | 74.4 | 56.7 | 66.7 | 77.44 |
| Δ_PPO | +0.2 | −0.6 | +0.1 | +3.4 | +10.0 | +2.62 |

## 6.2 MAIN RESULTS

**VERL Generalizes across Multiple Benchmarks of Varying Difficulty.** As summarized in Tab. 1 (Full details in App. H.3), VERL leads to consistent performance gains across benchmarks of different difficulty levels, ranging from elementary school problems (e.g., ASDiv) to Olympiad-level challenges (e.g., OlympiadBench). The improvements are particularly pronounced on benchmarks that demand multi-step reasoning rather than simple arithmetic. VERL achieves up to 21.4% and 10.0% absolute accuracy improvements on Gaokao 2024_I and AIME24 (in Tab. 1), respectively.

**VERL Generalizes across RL Algorithms and Base Models.** VERL is a plug-and-play method that can be integrated with different RL algorithms to enhance their performance. As shown in Tab. 1, applying VERL to GRPO and PPO improves the average results on 15 benchmarks for both the Llama and Qwen series, demonstrating its strong generalization ability.

**Gains in Both Exploration and Exploitation.** As shown in Tab. 2 (Full details in App. H.4), VERL yields larger improvements on Pass@$k$ (a measure of exploration) than on Pass@1 (a measure of exploitation), particularly on more challenging benchmarks. Since Pass@1 reflects exploitation and Pass@$k$ reflects exploration, the combined results of Tab. 1 and Tab. 2 demonstrate that VERL effectively enhances both abilities. For a detailed case study, see Sec. I.

**Performance Degradation on Some Datasets** As shown in Tab. 1, the minor drop on CMATH (-1.0) with Qwen occurs at a high-performance saturation level ( 91%), likely reflecting statistical variance rather than capability degradation. Meanwhile, the drop on Gaokao (-7.1) with Llama is attributable to the optimization dynamics specific to GRPO, as VERL achieves a substantial +7.2 gain on the exact same benchmark under the PPO setting (Row 6). Crucially, these isolated fluctuations are outweighed by the consistent improvements in Average accuracy across all models (e.g., +2.0% for Llama) and the significant breakthroughs on challenging OOD reasoning tasks (e.g., AIME24 +10.0%). This confirms that VERL's benefits in promoting robust reasoning significantly exceed the cost of minor local variance.

## 6.3 ABLATION ANALYSES

We conduct ablation studies on the key hyperparameters or components of VERL: the effectiveness of ERA, the stride length ($s$) for temporal dynamics, the advantage clipping factor ($\kappa$), and the composition of the auxiliary shaping signal.

Table 3: Pass@1 performance with variant $\beta$. "Adapted $\beta$" denotes $\beta := \text{sigmoid}(d_2)$. In this paper, all results in the table are reported in percentage (%), with **Bold** indicating the best performance.

| Training Strategy | Score Avg | In Domain | | | Out of Domain | | | | Hard Problems | | | | |
|---|---|---|---|---|---|---|---|---|---|---|---|---|---|
| | | MATH | MATH500 | Avg | Gaokao | CN Middle School | CMATH | Avg | AIME24 | AIME25 | AMC23 | AMC24 | Avg |
| GRPO | 0.36 | **51.4** | 46.2 | 48.80 | **23.7** | 28.7 | 28.3 | 26.90 | 3.3 | 0.0 | **27.5** | 8.9 | 9.93 |
| GRPO+VERL ($\beta = 0.5$) | **0.38** | 51.2 | 47.2 | 49.20 | 21.2 | 36.6 | 38.7 | 32.17 | 10.0 | 0.0 | **27.5** | 8.9 | 11.60 |
| GRPO+VERL (Adapted $\beta$) | **0.38** | 50.9 | **51.2** | **51.05** | 22.9 | **38.6** | **46.2** | **35.90** | **13.3** | **6.7** | 20.0 | **11.1** | **12.78** |

(a) Different Strides (s).  (b) Different advantage clip ($\kappa$).  (c) Different shaping signals.

Figure 5: Comparison of various hyperparameters with Llama-3.2-3B-Instruct. It shows that the model performs best with a stride of 40 in (a) and with $\kappa = 2$ in (b). We adopt these settings for all subsequent experiments. Moreover, (c) indicates that using only one signal, either exploration or exploitation, leads to suboptimal performance, demonstrating the effectiveness of our method.

**Analysis of the Effectiveness of ERA.** As shown in Tab. 3, the comparison between GRPO without/with $\beta = 0.5$ shows that incorporating hiddenstate-level rewards provides consistent gains over the token-level baseline, demonstrating the advantage of leveraging richer internal representations during optimization. The gap between the fixed $\beta$ and the Adapted $\beta$ variant indicates that dynamic adjustment of ERA provides a more reliable estimate of when to emphasize exploration versus exploitation.

**Analysis of Stride** ($s$)**.** The stride s determines the granularity for calculating temporal difference metrics. As depicted in Fig. 5 (a), VERL's performance improvement is robust across various stride values, indicating the underlying signal is not overly sensitive to sampling frequency. We find that $s = 40$ yields optimal validation rewards, striking an effective balance between capturing significant temporal shifts and avoiding noise from minor token-level fluctuations.

**Analysis of Advantage Clip** ($\kappa$)**.** The advantage clipping factor $\kappa$ stabilizes training by ensuring our auxiliary term acts as a refinement rather than a dominant signal. It constrains the shaping bonus to a fraction of the original advantage, preventing it from overpowering the primary task reward. The results in Fig. 5 (b) show that VERL consistently enhances performance for all tested values of $\kappa$ underscoring its stability. Optimal performance is achieved at $\kappa = 2$, which provides a sufficiently strong and well-proportioned signal to guide the policy without destabilizing the learning process.

**Analysis of Shaping Signals** ($\Phi$)**.** As shown in Fig. 5 (c), compared to the full formulation in Eq. 11, using only the exploration-related term prevents the model from exploiting high-reward trajectories, leading to earlier bottlenecks and inferior final performance. In contrast, using only the exploitation-related term yields higher initial returns but quickly plateaus due to insufficient exploration. When combining both terms, the model achieves more stable training and superior final performance.

## 7 CONCLUSION

We challenge the conventional exploration-exploitation capacities trade-off in LLM reasoning blamed on token-level analysis and shift focus to their decoupled relation in hidden-state representations (measured as semantic diversity and information gain velocity, respectively). We introduce ER, ERV and ERA to quantify the dynamics of semantic complexity, with ERA as a stable indicator distinguishing correct from flawed reasoning. We further propose Velocity-Exploiting Rank-Learning (VERL) method, which uses ERA as a meta-controller to adaptively shape the advantage function, moving beyond the trade-off for simultaneous enhancement. Extensive experiments validate VERL's superior out-of-domain generalization and performance on complex reasoning tasks.

## ETHICS STATEMENT

The authors of this paper have read and agree to adhere to the ICLR Code of Ethics. Our research is focused on the fundamental analysis of internal representations in Large Language Models and the development of a novel reinforcement learning algorithm. This work does not involve human subjects, and we did not collect any new datasets containing personally identifiable or sensitive information. The experiments were conducted using publicly available and widely-used mathematical reasoning benchmarks (such as MATH and GSM8K), which are standard in the field and do not raise immediate privacy or data bias concerns in the context of this study. While we acknowledge that any advancement in LLM reasoning capabilities could be applied in various ways, our work is foundational and aimed at improving the robustness and efficiency of AI systems. We do not foresee any direct negative societal impacts stemming from this research.

## REPRODUCIBILITY STATEMENT

To ensure the reproducibility of our findings, we have provided comprehensive details throughout the paper and its appendices. The core methodology of our proposed Velocity-Exploiting Rank-Learning (VERL) is described in Sec. 5, with a concrete implementation outlined in Algorithm 1. All experimental settings, including the base models used (Llama and Qwen series), datasets, reward function design, and key hyperparameters for both GRPO and PPO training, are detailed in Sec. 6.1 and further expanded upon in App. G. The theoretical foundations for our proposed metrics (ER, ERV, and ERA) are established in Sec. 3, with complete mathematical proofs for our claims provided in App. H. As stated in the abstract, the source code to replicate our experiments will be made publicly available upon publication of this work.

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

CONTENTS OF APPENDIX

# A  NOTATIONS

| Symbol | Description |
|---|---|
| $\pi$ | Large language model policy |
| $A_{i,t}$ | Group-relative advantage for the $t$-th token in the $i$-th response in group |
| $A^{(0)}$ | Original advantage estimation |
| $\hat{A}$ | Reshaped advantage value |
| $z_t$ | Hidden state corresponding to the $t$-th step of the output token |
| $\bar{z}_i$ | Single vector representation for the $i$-th response by averaging its token hidden states |
| $\mathbf{Z}_c$ | Mean-centered hidden state matrix |
| $\overline{\mathbf{Z}}_{1:n}$ | Dataset-level hidden states matrix formed by the first $n$ prompts |
| $\mathbf{Z}$ | Response-level hidden states matrix |
| $\Delta_M^{(i)}$ | The $i$-order temporal difference for metric $M$ |
| $\mathcal{M}$ | Set of metrics derived from the hidden states |
| $M_i$ | T$i$-order temporal difference of ER, exactly the different metrics |
| $m_t$ | Value of metric $M$ computed on the token sub-sequence from the start to position $t$ |
| $\epsilon_{\text{high/low}}$ | Hyperparameter for the upper/lower bound used for clipping |
| $\epsilon$ | Small constant for numerical stability |
| $r_j$ | Reward of the $j$-th response |
| $\mathbf{w}_{\text{explore/exploit/dyn}}$ | Exploration-focused profile/Exploitation-focused profile/Dynamic-weighted profile |
| $w_{dyn,i}$ | The $i$-th scalar of $\mathbf{w}_{\text{dynamic}}$ |
| $\text{rank}(\cdot)/\text{erank}(\cdot)$ | Conventional rank/Effective rank function |
| $\mathbf{SVD}(\cdot)$ | a function to calculate the singular values |
| $\delta_n^{(i)}$ | Instantaneous $i$-Order Difference for step $n$ |
| $s$ | The stride for effective rank velocity calculation |
| $y_t$ | The $t$-th step (token) of the model's response |
| $y_{i:j}$ | Sequence of reasoning steps from $i$ to $j$ |
| $\bar{\mu}_k$ | Exponential Moving Average for metric $M_k$ |
| $\lambda_j(\cdot)$ | The $j$-th eigenvalues of the given matrix |
| $\beta$ | Interpolation coefficient for VERL training |
| $G$ | Size of sampled group in GRPO |
| $\theta$ | Large language model policy's parameter |
| $\phi$ | The parameter corresponding to the optimal policy |
| $\Phi$ | Auxiliary advantage |
| $t$ | Time step |
| $T$ | Output length |
| $L_{\max}$ | The maximum length of model's output |
| $x$ | Prompt |
| $S$ | Sample times per prompt |
| $\mathcal{P}_x$ | Distribution of prompts |
| $\rho_t$ | Probability ratio between the current and old policies for $t$-step of the output |
| $p_j$ | The $j$-th normalized singular values |
| $\mathcal{L}_{\text{PPO}}(\cdot)$ | The optimization objective for PPO applied to policy |
| $D$ | Feature dimension of hidden states |
| $N$ | The size of the dataset |
| $\sigma_j$ | The $j$-th singular value of matrix |
| $\mathbf{p}$ | Singular value distribution |
| $\mathcal{T}$ | Set of time steps |
| $d_k$ | Deviation for metric $M_k$ |
| $\kappa$ | Advantage clipping factor |
| $\Delta$ | Performance difference between the baseline RL method and its VERL variant |
| $H(\cdot)$ | Shannon entropy function |
| $\mathbf{q}_i$ | The $i$-th row of the dataset matrix |
| $\overline{\mathbf{K}}$ | Gram matrix of dataset matrix |
| $U_t$ | Uncentered Gram matrix |

## B  THE USE OF LARGE LANGUAGE MODELS (LLMs)

We used large language models for text polishing.

## C  ALGORITHM

---

**Algorithm 1** VERL: Training

---

1: **Input:** $\mathcal{D} = \{x^i\}_{i=1}^N$, prompt $x^i$, policy model $\pi_\theta$, hidden-state dimension $D$, sample times per prompt $S$.
2: **Parameters:** EMA factor $\gamma$, relative deviation stabilizing factor $\epsilon \ll 1$, RL fine-tuning stabilizing factor $\kappa$.
3: **Initialize:** Randomly initialize policy parameters $\pi_\theta$, historical averages of metrics $\bar{\mu}_{\text{ER}} = \bar{\mu}_{\text{ERV}} = \bar{\mu}_{\text{ERA}} = 0$, exploration capacity profile $\mathbf{w}_{\text{explore}} = [1, 0]$, exploitation capacity profile $\mathbf{w}_{\text{exploit}} = [0, 1]$.
4: **Output:** A well-trained policy model $\pi_\theta$.
5: **repeat**
6:     **for** $x^i \in \mathcal{D}$ **do**:                        // Pick a sample from dataset
7:         **for** 1 **to** $S$ **do**:                // Rolling $S$ times for one sample
8:              $y_0^i \leftarrow x^i$, $\mathbf{Z}_0^i \leftarrow \varnothing$, $t \leftarrow 1$
9:              **repeat**                        // Generation process
10:                 $y_t^i, \mathbf{z}_t^i \sim \pi_\theta(\cdot|y_{t-1}^i)$
11:                 $y_t^i \leftarrow [y_{t-1}^i; y_t^i]$           // Concatenate token sequence
12:                 $\mathbf{Z}_t^i \leftarrow [\mathbf{Z}_{t-1}^i; \mathbf{z}_t^i] \in \mathbb{R}^{t \times D}$
13:                 $\boldsymbol{\sigma}_t^i \leftarrow \text{SVD}(\mathbf{Z}_t^i)$
14:                 $j \leftarrow |\boldsymbol{\sigma}_t^i|$
15:                 $p_{j,t}^i \leftarrow \sigma_{j,t}^i / \sum_j \sigma_{j,t}^i$
16:                 $\text{erank}_t^i \leftarrow \exp\left(-\sum_j p_{j,t}^i \log p_{j,t}^i\right)$
17:                 **If** $t > 1$ **then**: $\delta_{\text{ERV},t} \leftarrow \text{erank}_t^i - \frac{1}{t-1}\sum_{k=1}^{t-1} \text{erank}_k^i$
18:                 **If** $t > 2$ **then**: $\delta_{\text{ERA},t} \leftarrow \delta_{\text{ERV},t} - \delta_{\text{ERV},t-1}$
19:                 $t \leftarrow t + 1$
20:              **until** rolling done the sentence;       // $t - 1$ is the final timestep while rolling done
21:              $A_{origin}^i \leftarrow$ base RL evaluating on $y_{t-1}^i$
22:              $m_{\text{ER}}^i \leftarrow \text{erank}_{t-1}^i$                   // Calculating ER metric
23:              $m_{\text{ERV}}^i \leftarrow \frac{1}{t-2}\sum_{t=2}^{t-1} \delta_{\text{ERV}}^t$           // Calculating ERV metric
24:              $m_{\text{ERA}}^i \leftarrow \frac{1}{t-3}\sum_{t=3}^{t-1} \delta_{\text{ERA}}^t$           // Calculating ERA metric
25:              $\bar{\mu}_k \leftarrow \gamma\bar{\mu}_k + (1-\gamma)m_k^i$, $k \in \{\text{ER}, \text{ERV}, \text{ERA}\}$
26:              $d_k^i \leftarrow \frac{m_k^i - \bar{\mu}_k}{|\bar{\mu}_k| + \epsilon}$, $k \in \{\text{ER}, \text{ERV}, \text{ERA}\}$
27:              $\beta^i \leftarrow \text{sigmoid}(d_{\text{ERA}}^i)$
28:              $\mathbf{w}_{\text{dyn}}^i \leftarrow \beta^i \mathbf{w}_{\text{explore}} + (1-\beta^i)\mathbf{w}_{\text{exploit}}$
29:              $w_{\text{dyn,ER}}^i \leftarrow$ the first scalar value of $\mathbf{w}_{\text{dyn}}^i$
30:              $w_{\text{dyn,ERV}}^i \leftarrow$ the second scalar value of $\mathbf{w}_{\text{dyn}}^i$
31:              $\Phi^i \leftarrow w_{\text{dyn,ER}}^i \tanh(d_{\text{ER}}^i) + w_{\text{dyn,ERV}}^i \tanh(d_{\text{ERV}}^i)$
32:              $\hat{A}^i \leftarrow A_{origin}^i + \min\left(\max(0, \Phi^i), \frac{|A_{origin}^i|}{\kappa}\right)$
33:         **end for**
34:     **end for**
35:     Update $\theta$ via base RL objective with $\hat{A}^i$
36: **until** $\theta$ converges;

---

# D   RELATED WORK

## D.1   REINFORCEMENT LEARNING WITH VERIFIABLE REWARDS

To unlock the full reasoning potential of Large Language Models (LLMs), Reinforcement Learning with Verifiable Rewards (RLVR) has become a prominent training paradigm. This approach was notably employed by DeepSeek-R1-Zero (Guo et al., 2025) and , which executes complex reasoning processes through actions such as reflection and validation. Following the success of DeepSeek-R1, a significant body of research has investigated the efficacy of RLVR on popular open-source LLMs, including Qwen (Yang et al., 2024), Mistral (Jiang et al., 2024), and LLaMA (Dubey et al., 2024).

This has fostered an optimistic view that RLVR can not only enhance existing model capabilities but also enable the acquisition of novel reasoning knowledge, facilitating a path toward continuous self-improvement (Zeng et al., 2025; Yu et al., 2025). RLVR training has been shown to grant LLMs controllable output length for efficient inference (Yan et al., 2025; Cheng et al., 2025b), deepen their reasoning pathways (Bensal et al., 2025), mitigate their weaknesses (Liang et al., 2025a;b), enable the use of external tools (Rainone et al., 2025; Jin et al., 2025), and even facilitate unsupervised reasoning (Zuo et al., 2025). However, Some studies (Yue et al., 2025) argue that while RLVR significantly improves the confidence and reliability of model reasoning, it may inadvertently constrain the model's exploratory capacity. The core of this issue lies in RLVR's optimization objective: maximizing expected rewards. This objective function inherently biases the policy gradient toward reinforcing known trajectories that lead to high rewards (i.e., "exploitation"), while suppressing the exploration of unknown paths that may offer potentially higher returns but also carry greater risk (i.e., "exploration"). Consequently, the outputs of RLVR-optimized models often remain confined within the sampling distribution of the base model, suggesting the paradigm excels at refining existing knowledge rather than generating new knowledge. This trade-off between exploration and exploitation constitutes a central challenge in the contemporary RLVR landscape.

## D.2   EXPLORATION AND EXPLOITATION IN RLVR OF LLM

Recent perspectives (Wang et al., 2025; Cui et al., 2025) on the exploration-exploitation dilemma have predominantly been shaped by analyses at the token level, focusing on the prediction distribution over the vocabulary. From this viewpoint, higher entropy in the token-level prediction—indicating a more uniform distribution over the next token—is interpreted as a sign of greater exploratory behavior, as it suggests a capacity for more diverse responses. This has led to the adoption of techniques (Deng et al., 2025a; Cheng et al., 2025a) such as entropy regularization to explicitly encourage the policy to explore novel reasoning paths. Conversely, lower entropy in the token-level prediction is taken to signify higher model confidence in its reasoning chain, thus representing strong exploitation. Subsequent work (Fu et al., 2025) has also utilized metrics derived from the top-k probabilities of the token prediction to quantify confidence. For instance, some approaches (Damani et al., 2025; Qiao et al., 2025) leverage the model's internal "confidence" signals to dynamically evaluate and filter the quality of reasoning steps, while others have employed high confidence as a feedback signal to enable unsupervised reinforcement learning (Li et al., 2025b). Ultimately, however, these confidence-based metrics are not fundamentally different from entropy. This token-level standard of measurement introduces an endogenous contradiction: classifying behavior as either exploratory or exploitative requires the introduction of a prior assumptions, a practice that is disadvantageous for LLM research.

In this paper, we depart from this paradigm. We shift the analysis from the token level to the semantic space at the response level. This approach allows us to decouple the intertwined elements of exploration and exploitation, aiming to achieve a simultaneous enhancement of both during reinforcement learning.

## D.3   REPRESENTATION DYNAMICS IN DEEP REINFORCEMENT LEARNING

Beyond RLVR for language models, there is a line of classical deep RL work that explicitly studies how neural representations evolve during training and how this affects exploration and sample efficiency. State representation learning (SRL) for control aims to construct low-dimensional, action-dependent embeddings that preserve task-relevant dynamics while discarding nuisance variation. This work Lesort et al. (2018) provide a comprehensive overview of SRL methods for robotics and control, emphasizing how compact latent states can improve both data efficiency and stability of downstream RL algorithms. More recently, several works have directly analyzed the feature dynam-

ics of deep RL agents. The work Kumar et al. (2020) identify an "implicit under-parameterization" phenomenon in value-based deep RL: repeated bootstrapping updates lead to a collapse in the effective rank of value-network features, which in turn correlates with degraded performance in both online and offline settings. The work Lyle et al. (2022) further study capacity loss, showing that networks trained on non-stationary targets can lose their ability to fit new value functions over time, and proposing Initial Feature Regularization (InFeR) to stabilize the feature subspace and improve performance on sparse-reward Atari tasks. Complementary to these analyses, Wang et al. (2022) systematically measure multiple representational properties (e.g., dynamics-awareness, orthogonality) across thousands of agent–task combinations, and relate them to transfer performance, highlighting that not all good control policies arise from equally useful representation geometries.

# E DETAILS OF THEOREMS

## E.1 PROOF OF THEOREM 3.1

*Suppose we have a matrix of embeddings $\mathbf{Z} \in \mathbb{R}^{T \times D}$. Then the effective rank of $\mathbf{Z}$ is a lower bound of $\mathrm{rank}(\mathbf{Z})$:*

$$1 \leqslant \mathrm{erank}(\mathbf{Z}) \leqslant \mathrm{rank}(\mathbf{Z}) \leqslant \min\{T, D\} \tag{13}$$

*Proof.* Let the singular value distribution of the matrix $\mathbf{Z}$ be $\mathbf{p} = (p_1, p_2, \ldots, p_{min\{T,D\}})$. The Shannon entropy of this distribution $H(\mathbf{p})$ is bounded. Its minimum is 0, which occurs when only one element of $p$ is 1 and all others are 0. Its maximum is $\log k$, where $k$ is the number of non-zero singular values, and this occurs when the distribution is uniform ($p_j = 1/k$ for all non-zero values). The lower bound is established from the minimum entropy value:

$$\mathrm{erank}(\mathbf{Z}) = \exp(H(\mathbf{p})) \geqslant \exp(0) = 1 \tag{14}$$

Equality holds if and only if the singular value distribution is $(1, 0, \ldots, 0)$, meaning $\mathbf{Z}$ has only one non-zero singular value. For the upper bound, let $k = \mathrm{rank}(\mathbf{Z})$ be the number of non-zero singular values of $\mathbf{Z}$. The entropy of the distribution $p$ is calculated only over these $k$ values and is maximized when they are uniform. Therefore

$$H(\mathbf{p}) \leqslant \log k \tag{15}$$

Applying the exponential function to this inequality gives:

$$\mathrm{erank}(\mathbf{Z}) = \exp(H(\mathbf{p})) \leqslant \exp(\log k) = k = \mathrm{rank}(\mathbf{Z}) \tag{16}$$

This establishes that the effective rank is upper-bounded by the conventional rank. The final inequality, $\mathrm{rank}(\mathbf{Z}) \leqslant \min\{T, D\}$, is a standard property of matrix rank. Equality for $\mathrm{erank}(\mathbf{Z}) = \mathrm{rank}(\mathbf{Z})$ holds if and only if the non-zero singular values are all equal, corresponding to a uniform singular value distribution over its support.

## E.2 PROOF OF PROPOSITION 3.5

*The zero-order metric and first-order difference of the effective rank scales linearly with the number of responses, $\Delta_M^{(0)} = \mathcal{O}(N), \Delta_M^{(1)} = \mathcal{O}(N)$. The second-order difference of the effective rank is constant and does not depend on $N$, yielding a scaling order of $\Delta_M^{(2)} = \mathcal{O}(1)$.*

*Proof.* Without loss of generality, we take the effective rank for example. We adopt the provided definition of effective rank for a representation matrix $\mathbf{Z}$ with singular values $\{\sigma_i\}$:

$$\mathrm{erank}(\mathbf{Z}) = \exp\left(-\sum_j p_j \log(p_j)\right), \quad \text{where} \quad p_j = \frac{\sigma_j}{\sum_k \sigma_k} \tag{17}$$

Our analysis focuses on the dataset matrix $\overline{\mathbf{Z}} \in \mathbb{R}^{N \times D}$, whose rows $\{\mathbf{q}_i\}_{i=1}^N$ are the mean token embeddings of $N$ responses. The singular values $\sigma_i(\overline{\mathbf{Z}})$ of $\overline{\mathbf{Z}}$ are the square roots of the eigenvalues of the Gram matrix $\overline{\mathbf{K}} = \overline{\mathbf{Z}}\overline{\mathbf{Z}}^\top$; i.e., $\sigma_j(\overline{\mathbf{Z}}) = \sqrt{\lambda_j(\overline{\mathbf{K}})}$. Given that the rows of $\overline{\mathbf{Z}}$ are nearly

orthogonal, the Gram matrix $\overline{\mathbf{K}}$ is strongly diagonal-dominant. Its eigenvalues can be approximated by its diagonal entries:

$$\lambda_j(\overline{\mathbf{K}}) \approx \overline{\mathbf{K}}_{jj} = \|\mathbf{q}_j\|^2 = \frac{1}{T} \quad \text{for } j = 1, \ldots, N \tag{18}$$

The matrix has $N$ significant eigenvalues, each approximately equal to $1/T$. The singular values of $\overline{\mathbf{Z}}$ are the square roots of the eigenvalues of $\overline{\mathbf{K}}$:

$$\sigma_j(\overline{\mathbf{Z}}) = \sqrt{\lambda_j(\overline{\mathbf{K}})} \approx \sqrt{\frac{1}{T}} = \frac{1}{\sqrt{T}} \quad \text{for } j = 1, \ldots, N \tag{19}$$

To calculate the effective rank, we first normalize these singular values to form a probability distribution $\{p_j\}$. The sum of singular values is:

$$\sum_{k=1}^{N} \sigma_k(\overline{\mathbf{Z}}) \approx \sum_{k=1}^{N} \frac{1}{\sqrt{T}} = \frac{N}{\sqrt{T}} \tag{20}$$

The individual probabilities are therefore:

$$p_j = \frac{\sigma_j}{\sum_k \sigma_k} \approx \frac{1/\sqrt{T}}{N/\sqrt{T}} = \frac{1}{N} \tag{21}$$

The distribution $\mathbf{p} = \{p_1, p_2, \ldots, p_N\}$ is a uniform distribution over $N$ states. The Shannon entropy of this distribution is maximal:

$$H(\mathbf{p}) = -\sum_{j=1}^{N} p_j \log(p_j) = -\sum_{j=1}^{N} \frac{1}{N} \log\left(\frac{1}{N}\right) = -N\left(\frac{-\log(N)}{N}\right) = \log(N) \tag{22}$$

The effective rank is the exponential of this entropy: $\text{erank}(\overline{\mathbf{Z}}) = \exp(H(\mathbf{p})) = \exp(\log(N)) = N$. In the maximal prompt entropy regime, the effective rank of the dataset matrix scales as $\mathcal{O}(N)$.

We adapt them to our context by defining the metric's value at "time" $n$ as the Effective Rank computed on the dataset matrix formed by the first $n$ prompts, denoted $\overline{\mathbf{Z}}_{1:n}$. Let $m_n = \text{erank}(\overline{\mathbf{Z}}_{1:n})$. From our previous analysis, we established a crucial result that forms the basis of this derivation: for maximal cases, the effective rank of a dataset with n prompts scales linearly with $n$.

$$m_n = \text{erank}(\overline{\mathbf{Z}}_{1:n}) \approx n \tag{23}$$

We will use this linear approximation to derive the scaling orders of the difference metrics, assuming a stride of $s = 1$ for simplicity. The first-order difference quantifies the average "velocity" of change in the metric relative to its historical mean. Instantaneous First-Order Difference ($\delta_n^{(1)}$) is the value at step $n$ minus the average of all preceding values.

$$\delta_n^{(1)} = m_n - \left(\frac{1}{n-1} \sum_{k=1}^{n-1} m_k\right) \tag{24}$$

Substituting our approximation $m_k \approx k$:

$$\delta_n^{(1)} \approx n - \left(\frac{1}{n-1} \sum_{k=1}^{n-1} k\right) \tag{25}$$

$$\delta_n^{(1)} \approx n - \left(\frac{1}{n-1} \cdot \frac{(n-1)n}{2}\right) = n - \frac{n}{2} = \frac{n}{2} \tag{26}$$

The instantaneous difference grows linearly with $n$. Overall First-Order Difference $\Delta_{\text{erank}}^{(1)}$: This is the average of the instantaneous differences over the entire dataset of size $N$.

$$\Delta_{\text{erank}}^{(1)} = \frac{1}{N-1} \sum_{n=2}^{N} \delta_n^{(1)} \approx \frac{1}{N-1} \sum_{n=2}^{N} \frac{n}{2} \tag{27}$$

$$\Delta_{\text{erank}}^{(1)} \approx \frac{1}{2(N-1)} \left( \left( \sum_{n=1}^{N} n \right) - 1 \right) = \frac{1}{2(N-1)} \left( \frac{N(N+1)}{2} - 1 \right) \tag{28}$$

For large $N$, the expression is dominated by the highest power of $N$:

$$\Delta_{\text{erank}}^{(1)} \sim \frac{N^2/4}{N} = \frac{N}{4} \tag{29}$$

The first-order difference of the effective rank scales linearly with the number of prompts, $\Delta_{\text{erank}}^{(1)} = \mathcal{O}(N)$. As for second-order difference, we compute the change in Instantaneous Differences between consecutive values of $\delta_n^{(1)}$.

$$\delta_n^{(1)} - \delta_{n-1}^{(1)} \approx \frac{n}{2} - \frac{n-1}{2} = \frac{1}{2} \tag{30}$$

This change is a constant, indicating a linear increase in the first-order difference. Overall Second-Order Difference $\Delta_{\text{erank}}^{(2)}$:

$$\Delta_{\text{erank}}^{(2)} = \frac{1}{N-2} \sum_{n=3}^{N} \left( d_n^{(1)} - d_{n-1}^{(1)} \right) \approx \frac{1}{N-2} \sum_{n=3}^{N} \frac{1}{2} \tag{31}$$

$$\Delta_{\text{erank}}^{(2)} \approx \frac{1}{N-2} \cdot (N-2) \cdot \frac{1}{2} = \frac{1}{2} \tag{32}$$

The second-order difference of the effective rank is constant and does not depend on $N$, yielding a scaling order of $\Delta_{\text{erank}}^{(2)} = \mathcal{O}(1)$.

# F  ADDITIONAL THEORETICAL SUPPORT FOR EXPLORATION AND EXPLOITATION METRICS

In this section we formalize the relationship between our proposed hidden-state metrics (*Effective Rank* and *Effective Rank Velocity*) and the classical notions of exploration and exploitation in reinforcement learning. We first show that the *old* token-level metrics (average log probability and response entropy) are algebraically coupled, whereas our *new* hidden-state metrics are not. We then provide a representation-level justification for interpreting Effective Rank as a measure of semantic exploration, and Effective Rank Velocity as a measure of representation-level exploitation that is strongly correlated with greedy value improvement under the PPO-style architecture used in RLVR. Throughout, we consider a conditional language model $p_\theta(y \mid x)$ and a Transformer backbone that produces hidden states $z_t \in \mathbb{R}^D$ at each time step $t$ for a given prompt $x$ and generated response $y_{1:T}$.

## F.1  PREVIOUS VS. OUR EXPLORATION–EXPLOITATION METRICS

In this subsection, we formalize the difference between the *previous* token-level metrics used in prior RLHF/RLVR work and the *our* hidden-state metrics proposed in this paper. For a given prompt $x$ and generated response $y_{1:T}$, let $\pi_\theta(\cdot \mid x, y_{<t})$ denote the model's token-level policy distribution at step $t$, i.e. the softmax over the vocabulary induced by the logits at that position.

**Previous metrics (token-level action space, log-probability, and entropy).**   We define the *average log probability* of an response and the *response entropy* as

$$\text{AvgLogProb}(x, y_{1:T}) := \frac{1}{T} \sum_{t=1}^{T} \log \pi_\theta\big(y_t \mid x, y_{<t}\big), \tag{33}$$

$$\text{RespEnt}(x, y_{1:T}) := \frac{1}{T} \sum_{t=1}^{T} H\big(\pi_\theta(\cdot \mid x, y_{<t})\big), \quad H(p) := -\sum_v p(v) \log p(v). \tag{34}$$

Thus $\mathrm{RespEnt}$ is the *token-level* entropy averaged over the response: at each step we compute the Shannon entropy of the vocabulary distribution and then average over time. At the response level sematic space, we consider $x$ drawn from a prompt distribution $p(x)$ and, for the purpose of analysis, responses $y_{1:T}$ drawn *on-policy* from the model $p_\theta(\cdot \mid x)$:

$$\mathcal{L}_{\text{avg-log}}(\theta) := \mathbb{E}_{x \sim p(x)} \, \mathbb{E}_{y_{1:T} \sim p_\theta(\cdot|x)} \Big[ \mathrm{AvgLogProb}(x, y_{1:T}) \Big], \tag{35}$$

$$\mathcal{H}_{\text{resp}}(\theta) := \mathbb{E}_{x \sim p(x)} \, \mathbb{E}_{y_{1:T} \sim p_\theta(\cdot|x)} \Big[ \mathrm{RespEnt}(x, y_{1:T}) \Big]. \tag{36}$$

**Proposition F.1** (Token-level exploitation and exploration are tightly coupled)**.** *Under on-policy sampling $y_{1:T} \sim p_\theta(\cdot \mid x)$, the corpus-level average log probability $\mathcal{L}_{avg\text{-}log}(\theta)$ and response entropy $\mathcal{H}_{resp}(\theta)$ satisfy*

$$\mathcal{L}_{avg\text{-}log}(\theta) \;=\; -\mathcal{H}_{resp}(\theta). \tag{37}$$

*In particular, under the same sampling distribution, any change of the model that increases token-level exploitation in action space as measured by $\mathcal{L}_{avg\text{-}log}$ necessarily decreases $\mathcal{H}_{resp}$ by the same amount (and vice versa).*

*Proof.* Fix a prompt $x$ and a time step $t$. Conditioned on $x$ and the history $y_{<t}$, the next token $y_t$ is drawn from $\pi_\theta(\cdot \mid x, y_{<t})$. Taking the expectation of $\log \pi_\theta(y_t \mid x, y_{<t})$ under this distribution yields:

$$\mathbb{E}_{y_t \sim \pi_\theta(\cdot|x,y_{<t})} \big[ \log \pi_\theta(y_t \mid x, y_{<t}) \big] = \sum_v \pi_\theta(v \mid x, y_{<t}) \log \pi_\theta(v \mid x, y_{<t}) = -H(\pi_\theta(\cdot \mid x, y_{<t})). \tag{38}$$

Now consider a full response $y_{1:T} \sim p_\theta(\cdot \mid x)$. By the law of iterated expectations,

$$\mathbb{E}_{y_{1:T} \sim p_\theta(\cdot|x)} \big[ \log \pi_\theta(y_t \mid x, y_{<t}) \big] = \mathbb{E}_{y_{<t} \sim p_\theta(\cdot|x)} \Big[ \mathbb{E}_{y_t \sim \pi_\theta(\cdot|x,y_{<t})} \big[ \log \pi_\theta(y_t \mid x, y_{<t}) \big] \Big] \tag{39}$$

$$= -\mathbb{E}_{y_{<t} \sim p_\theta(\cdot|x)} \Big[ H(\pi_\theta(\cdot \mid x, y_{<t})) \Big], \tag{40}$$

where we used equation 38 in the last step. Averaging over $t = 1, \dots, T$ and dividing by $T$ gives

$$\mathbb{E}_{y_{1:T} \sim p_\theta(\cdot|x)} \Big[ \mathrm{AvgLogProb}(x, y_{1:T}) \Big] = \mathbb{E}_{y_{1:T} \sim p_\theta(\cdot|x)} \Big[ \frac{1}{T} \sum_{t=1}^{T} \log \pi_\theta(y_t \mid x, y_{<t}) \Big] \tag{41}$$

$$= -\mathbb{E}_{y_{1:T} \sim p_\theta(\cdot|x)} \Big[ \frac{1}{T} \sum_{t=1}^{T} H(\pi_\theta(\cdot \mid x, y_{<t})) \Big] \tag{42}$$

$$= -\mathbb{E}_{y_{1:T} \sim p_\theta(\cdot|x)} \Big[ \mathrm{RespEnt}(x, y_{1:T}) \Big]. \tag{43}$$

Finally, taking expectation over prompts $x \sim p(x)$ on both sides of equation 43 yields

$$\mathcal{L}_{\text{avg-log}}(\theta) = -\mathcal{H}_{\text{resp}}(\theta), \tag{44}$$

which is exactly Eq. equation 37. This shows that under on-policy sampling, the two token-level metrics are related by a fixed negative sign and thus cannot be decoupled in the action space. $\square$

**Our metrics (hidden-state Effective Rank and velocity).** The next proposition shows that these two hidden-state metrics in Sec. 3.1 and 3.3 are *structurally decoupled* at the level of trajectories: knowing the final Effective Rank alone does not determine ERV, and conversely.

**Proposition F.2** (Hidden-state metrics are structurally decoupled)**.** *Fix $K \geq 3$ evaluation steps. Consider the map that associates to each Effective Rank trajectory $m = (m_1, \dots, m_K) \in \mathbb{R}^K$ its final value*

$$\mathrm{ER}_{final}(m) := m_K \tag{45}$$

*and its Effective Rank velocity*

$$\mathrm{ERV}(m) := \frac{1}{K-1} \sum_{j=2}^{K} \Big( m_j - \frac{1}{j-1} \sum_{k=1}^{j-1} m_k \Big). \tag{46}$$

*Then:*

1. *There is no function $f : \mathbb{R} \to \mathbb{R}$ such that $\mathrm{ERV}(m) = f(\mathrm{ER}_{final}(m))$ for all trajectories $m \in \mathbb{R}^K$.*

2. *There is no function $g : \mathbb{R} \to \mathbb{R}$ such that $\mathrm{ER}_{final}(m) = g(\mathrm{ERV}(m))$ for all trajectories $m \in \mathbb{R}^K$.*

*Equivalently, $\mathrm{ER}_{final}$ and $\mathrm{ERV}$ are not functionally dependent: they capture genuinely different aspects of the Effective Rank sequence.*

*Proof.* We view $\mathrm{ER}_{final}$ and $\mathrm{ERV}$ as real-valued functions on $\mathbb{R}^K$. The proof is purely algebraic and does not rely on any monotonicity of $m_j$.

Step 1: ERV is a non-trivial linear functional. Introduce the shorthand

$$\Delta m_j := m_j - m_{j-1}, \quad j \geqslant 2. \tag{47}$$

A direct calculation shows that each increment $\delta_j$ can be written as

$$\delta_j = \frac{1}{j-1} \sum_{r=2}^{j} (r-1) \, \Delta m_r, \quad j \geqslant 2, \tag{48}$$

so that ERV is a linear functional of $m$:

$$\mathrm{ERV}(m) = \sum_{j=1}^{K} \alpha_j m_j, \tag{49}$$

for some fixed coefficients $\alpha_1, \ldots, \alpha_K$ that depend only on $K$ (and $s$) and satisfy $\sum_{j=1}^{K} \alpha_j = 0$ and $\alpha_j \neq 0$ for at least two indices $j$ (e.g. $\alpha_1 \neq 0$ and $\alpha_K \neq 0$). In particular, ERV is *not* proportional to the projection onto any single coordinate $m_j$.

Step 2: No functional dependence of ERV on the final ER. Suppose, for contradiction, that there exists a function $f : \mathbb{R} \to \mathbb{R}$ such that

$$\mathrm{ERV}(m) = f(\mathrm{ER}_{final}(m)) = f(m_K) \quad \text{for all } m \in \mathbb{R}^K. \tag{50}$$

Fix any constant $c \in \mathbb{R}$. Consider the affine subspace

$$\mathcal{A}_c := \{ m \in \mathbb{R}^K : m_K = c \}. \tag{51}$$

On this subspace, $\mathrm{ER}_{final}(m) \equiv c$ is constant, so by assumption $\mathrm{ERV}(m) \equiv f(c)$ must also be constant. However, ERV is a non-trivial linear functional that depends on at least one coordinate $m_j$ with $j < K$. Therefore, restricted to $\mathcal{A}_c$, the map $m \mapsto \mathrm{ERV}(m)$ varies with those coordinates and cannot be constant. This yields a contradiction. Hence no such $f$ exists.

Step 3: No functional dependence of final ER on ERV. The argument is symmetric. Suppose there exists $g : \mathbb{R} \to \mathbb{R}$ such that

$$m_K = \mathrm{ER}_{final}(m) = g(\mathrm{ERV}(m)) \quad \text{for all } m \in \mathbb{R}^K. \tag{52}$$

Fix any constant $c \in \mathbb{R}$ and consider the affine subspace

$$\mathcal{B}_c := \{ m \in \mathbb{R}^K : \mathrm{ERV}(m) = c \}. \tag{53}$$

Since ERV is a non-trivial linear functional, $\mathcal{B}_c$ is an affine hyperplane of codimension 1, and $m_K$ can vary freely among its points. Yet the assumed relation $m_K = g(\mathrm{ERV}(m)) = g(c)$ would force $m_K$ to be constant on $\mathcal{B}_c$, which is impossible. Thus no such $g$ exists.

Combining the two steps, we conclude that $\mathrm{ER}_{final}$ and $\mathrm{ERV}$ are not functionally dependent on each other. $\qquad \square$

**Summary.** Proposition F.1 shows that the classical token-level metrics—average log probability and response entropy—are *algebraically coupled* under on-policy sampling and cannot be varied independently. In contrast, Proposition F.2 demonstrates that the proposed hidden-state metrics—terminal Effective Rank and Effective Rank velocity—are structurally decoupled: they are distinct functionals of the Effective Rank trajectory and capture complementary aspects of exploration (semantic diversity level) and exploitation (semantic diversity gain speed).

## F.2 EFFECTIVE RANK AS SEMANTIC EXPLORATION

We now formalize the interpretation of Effective Rank as a measure of semantic diversity and uncertainty in the hidden-state space, and hence as a representation-level proxy for exploration in LLM reasoning. We assume that the hidden states are semantic representations in the sense that downstream semantic properties can be approximately recovered as linear functionals of the hidden vectors. This is standard in representation learning and probing work on large language models.

**Assumption F.3** (Semantic linear decodability). There exists a collection of $K$ semantic features $s^{(1)}, \ldots, s^{(K)}$ (e.g., semantic roles, entity identities, factual attributes, intermediate reasoning states) such that for each time step $t$ and feature index $k$ we have

$$s_t^{(k)} \approx w_k^\top h_t, \quad w_k \in \mathbb{R}^D. \tag{54}$$

That is, semantic features are approximately linearly decodable from hidden states.

**Assumption F.4** (Bounded energy and finite support). For a given trajectory $Z_{1:T}$, there exists an orthonormal basis of semantic directions $\{e_1, \ldots, e_D\}$ such that each hidden state admits a decomposition

$$h_t = \sum_{i=1}^{D} a_{t,i} \, e_i, \tag{55}$$

with $\sum_{t=1}^{T} a_{t,i}^2 < \infty$ for all $i$, and only finitely many coordinates $a_{t,i}$ carry task-relevant semantic variation.

**Proposition F.5** (Effective Rank as semantic diversity and uncertainty). *Let $Z_{1:T}$ be a hidden-state trajectory satisfying Assumptions F.3–F.4. Let $Z_{1:T} = U\Sigma R^\top$ be its SVD, and $\mathrm{ER}(Z_{1:T})$ its Effective Rank. Then:*

1. *If the trajectory uses exactly $k$ orthogonal semantic directions with equal energy and no others, i.e. the singular values satisfy $\sigma_1 = \cdots = \sigma_k > 0$ and $\sigma_{k+1} = \cdots = \sigma_r = 0$, then $\mathrm{ER}(Z_{1:T}) = k$.*

2. *More generally, if the singular value spectrum of $Z_{1:T}$ becomes more* spread out *over more directions in the sense of majorization (i.e. the normalized singular value vector becomes more uniform over a larger support), then $\mathrm{ER}(Z_{1:T})$ increases.*

*Consequently, $\mathrm{ER}(Z_{1:T})$ is a basis-invariant, strictly increasing measure of the number of independent semantic directions that are effectively used by the hidden states, and thus a natural representation-level proxy for semantic exploration and uncertainty.*

*Proof.* We proceed in two parts.

**(1) Equal-energy $k$-dimensional semantic subspace.** Suppose $Z_{1:T}$ uses exactly $k$ orthogonal semantic directions with equal energy. Then, up to permutation, the non-zero singular values satisfy

$$\sigma_1 = \cdots = \sigma_k = c > 0, \qquad \sigma_{k+1} = \cdots = \sigma_r = 0. \tag{56}$$

The normalized singular values are thus

$$q_i = \begin{cases} 1/k, & i = 1, \ldots, k, \\ 0, & i > k, \end{cases} \tag{57}$$

and the entropy of $q$ is

$$H(q) = -\sum_{i=1}^{k} \frac{1}{k} \log \frac{1}{k} = \log k. \tag{58}$$

Therefore

$$\mathrm{ER}(Z_{1:T}) = \exp(H(q)) = \exp(\log k) = k. \tag{59}$$

This shows that, in the idealized case of exactly $k$ equi-energic semantic directions, Effective Rank matches the true semantic dimensionality $k$.

**(2) Monotonicity under majorization.** Consider two hidden-state trajectories $Z$ and $\tilde{Z}$ with normalized singular value spectra $q$ and $\tilde{q}$, respectively. Suppose that $q$ is *majorized* by $\tilde{q}$ (denoted $q \prec \tilde{q}$), meaning intuitively that $\tilde{q}$ is "more spread out" and therefore more uniform across a larger support.

It is a standard result in information theory that the Shannon entropy $H(\cdot)$ is Schur-concave: if $q \prec \tilde{q}$, then $H(q) \leqslant H(\tilde{q})$ with strict inequality whenever $q \neq \tilde{q}$. Therefore

$$\mathrm{ER}(Z) = \exp(H(q)) \leqslant \exp(H(\tilde{q})) = \mathrm{ER}(\tilde{Z}), \tag{60}$$

with strict inequality when the majorization is strict. In words, whenever the singular value spectrum becomes more spread out across more directions, the Effective Rank strictly increases.

Combining the two parts, we see that Effective Rank equals the number of equi-energic semantic directions in the idealized case and increases whenever the representation distributes energy over more orthogonal directions. Since, by Assumption F.3, semantic features are linearly decodable along such directions, $\mathrm{ER}(Z_{1:T})$ provides a basis-invariant measure of how many independent semantic dimensions are explored by the hidden states and how evenly they are used. This justifies its interpretation as a representation-level exploration and uncertainty metric. □

### F.3 Effective Rank Velocity as Semantic Exploitation

Building on Sec. 3.1, where Effective Rank (ER) is shown to measure the number and uniform use of semantic directions in the hidden-state space, we now give a representation-only justification for interpreting Effective Rank Velocity (ERV) as *semantic exploitation*.

Throughout this subsection we fix a single trajectory $Z_{1:T}$ and a stride $s$. Let the evaluation positions be $t_j = js$ for $j = 1, \ldots, K$ with $K = \lfloor (T-1)/s \rfloor$, and write

$$m_j := \mathrm{ER}\big(Z_{1:t_j}\big), \qquad j = 1, \ldots, K. \tag{61}$$

Thus, $\{m_j\}_{j=1}^K$ is the ER trajectory of the growing prefixes of the same response.

**ERV as a recency-weighted sum of ER increments.** For convenience we recall the notation of Def. 3.3 with $M = \mathrm{ER}$. Define the local ER increments

$$\Delta m_r := m_r - m_{r-1}, \quad r \geqslant 2. \tag{62}$$

Def. 3.3 introduces the "instantaneous difference"

$$\delta_j := m_j - \frac{1}{j-1} \sum_{k=1}^{j-1} m_k, \qquad j \geqslant 2, \tag{63}$$

and the first-order temporal difference (ERV) as

$$\Delta_{\mathrm{ER}}^{(1)} := \frac{1}{K-1} \sum_{j=2}^K \delta_j. \tag{64}$$

The following lemma makes explicit that ERV is a recency-weighted average of the consecutive ER increments.

**Lemma F.6** (Recency-weighted velocity of ER). *For any sequence $(m_j)_{j=1}^K$, the instantaneous differences admit the representation*

$$\delta_j = \frac{1}{j-1} \sum_{r=2}^j (r-1)\,\Delta m_r, \qquad j \geqslant 2, \tag{65}$$

*and hence ERV can be written as*

$$\Delta_{ER}^{(1)} = \sum_{r=2}^K w_r\,\Delta m_r, \quad w_r := \frac{r-1}{K-1} \sum_{j=r}^K \frac{1}{j-1} > 0. \tag{66}$$

*In particular, ERV is a positive linear combination of the local ER increments $\Delta m_r$, assigning larger weights to more recent steps.*

*Proof.* Eq. 65 is exactly in Def. 3.3 with $M = $ ER, obtained by expressing $\delta_j$ in terms of the increments $\Delta m_r$ via telescoping. Plugging Eq. 65 into the definition of $\Delta_{\text{ER}}^{(1)}$ and exchanging the order of summation yields

$$\Delta_{\text{ER}}^{(1)} = \frac{1}{K-1} \sum_{j=2}^{K} \frac{1}{j-1} \sum_{r=2}^{j} (r-1)\,\Delta m_r = \sum_{r=2}^{K} \left[ \frac{r-1}{K-1} \sum_{j=r}^{K} \frac{1}{j-1} \right] \Delta m_r, \qquad (67)$$

which gives Eq. 66 with $w_r$ as stated. Since $r - 1 > 0$ and the harmonic tail $\sum_{j=r}^{K}(j-1)^{-1}$ is positive, we have $w_r > 0$ for all $r$. □

**Semantic exploitation as positive ER drift in a fixed semantic subspace.**   App. F.1 states that, up to an orthonormal change of basis, the hidden states can be written as $h_t = \sum_i a_{t,i} e_i$ with bounded energy along each semantic direction $e_i$, and that ER is a strictly increasing, basis-invariant measure of how many semantic directions are effectively used and how evenly energy is distributed among them. In particular, if the set of active directions (support of the singular value spectrum) is kept fixed and the spectrum becomes more uniform (in the sense of majorization), then ER strictly increases. Motivated by this, we isolate an idealized *semantic exploitation* regime in which the trajectory has already selected a semantic subspace and is refining it.

**Definition F.7** (Semantic exploitation regime). Let $(m_j)_{j=1}^{K}$ be the ER trajectory of a response, and let $q^{(j)}$ denote the normalized singular value vector of $Z_{1:t_j}$. We say that steps $j = 2, \ldots, K$ form a *semantic exploitation regime with rate $\mu > 0$* if:

1. (Fixed semantic support) The support of $q^{(j)}$ is independent of $j$, i.e., the set of active semantic directions is fixed.

2. (Uniformization within the support) For every $j \geqslant 2$, $q^{(j)}$ is more uniform than $q^{(j-1)}$ on this fixed support, in the sense of majorization, so that by Prop. F.5 we have $m_j - m_{j-1} = \Delta m_j \geqslant \mu$ for some $\mu > 0$.

Intuitively, condition (i) says that the model has committed to a particular semantic subspace (a line of reasoning), and condition (ii) says that it keeps redistributing energy within this subspace to make use of all its semantic directions more evenly. This is precisely the notion of "refining a promising strategy" in representation space.

**ERV lower-bounds the semantic exploitation rate.**   Under Def. F.7, ER experiences a persistent positive drift along the trajectory. The next proposition shows that ERV is a quantitative lower bound on this drift, and thus a natural measure of semantic exploitation strength.

**Proposition F.8** (ERV as a lower bound on semantic exploitation rate). *Assume the hidden states satisfy Assumptions F.3 and F.4 and that steps $j = 2, \ldots, K$ form a semantic exploitation regime with rate $\mu > 0$ in the sense of Def. F.7, so that $\Delta m_j \geqslant \mu$ for all $j \geqslant 2$. Then*

$$\Delta_{ER}^{(1)} \geqslant \frac{\mu K}{4}. \qquad (68)$$

*In particular, ERV is strictly positive and grows linearly with the length $K$ of the exploitation segment.*

*Proof.* By Eq. 65 and the assumption $\Delta m_r \geqslant \mu$ we obtain, for each $j \geqslant 2$,

$$\delta_j = \frac{1}{j-1} \sum_{r=2}^{j} (r-1)\,\Delta m_r \geqslant \frac{1}{j-1} \sum_{r=2}^{j} (r-1)\,\mu = \frac{\mu}{j-1} \sum_{r=2}^{j} (r-1) = \frac{\mu j}{2}. \qquad (69)$$

Averaging over $j$ then yields

$$\Delta_{\text{ER}}^{(1)} = \frac{1}{K-1} \sum_{j=2}^{K} \delta_j \geqslant \frac{1}{K-1} \sum_{j=2}^{K} \frac{\mu j}{2} = \frac{\mu}{2(K-1)} \sum_{j=2}^{K} j. \qquad (70)$$

For $K \geqslant 2$ we have $\sum_{j=2}^{K} j \geqslant \frac{K(K-1)}{2}$, so

$$\Delta_{\text{ER}}^{(1)} \geqslant \frac{\mu}{2(K-1)} \cdot \frac{K(K-1)}{2} = \frac{\mu K}{4}, \qquad (71)$$

which proves Eq. 68. □

Thus, in an idealized regime where the model has already discovered a useful semantic subspace and is consistently enriching it, ERV provides a strictly positive, linearly growing lower bound on the rate at which semantic complexity within that subspace is being exploited.

# G IMPLEMENTATION DETAILS

## G.1 TRAINING AND EVALUATION DETAILS

We typically use the same set of hyperparameters to train and evaluate all models in the SimpleRL-Zoo series (Zeng et al., 2025) in the default main experiment setting.

**Training.** We conduct all experiments with 4 A800-PCIE-80GB GPUs. For GRPO and PPO, we use a prompt batch size of 48 with a maximum rollout length of 1536 tokens. Training is performed using a mini-batch size of 24. For GRPO, we generate 4 rollouts per prompt. For PPO, we use DeepSeek-R1-Distill-Qwen-1.5B (Guo et al., 2025) as the value model and generate 1 rollout per prompt. The default sampling temperature is set to 1.0, and the clip ratio is 0.2. For all actor models ranging from 3B to 8B parameters, we use a learning rate of 1e-6 and a KL loss coefficient of 1e-4. For critic models in PPO, we use a learning rate of 1e-5. For our training datasets, we follow the same setup as in Zeng et al. (2025), where the data is filtered from GSM8K (Cobbe et al., 2021) and MATH (Hendrycks et al., 2021) configured with different difficulty levels for models of varying capabilities. We tested using the checkpoint model trained up to 120 steps.

**Evaluation.** We build our evaluation script based on that of Zeng et al. (2025), using a temperature of 0.6 and a maximum generation length of 2048 tokens. To ensure consistency, we adopt the same prompt template used during training. For most benchmarks, we report Pass@1 results. However, for benchmarks like AIME 2024, which contains fewer problems, we report both Pass@1 and average accuracy (Pass@256), computed over 256 generated samples per problem.

**Base Models.** To demonstrate the universality of our insights and methods, we conduct zero RL training experiments on Llama-3.2 (3B), Llama-3.1 (8B) (Dubey et al., 2024), Mistral-v0.3-7B (Jiang et al., 2024), and Qwen-2.5 (1.5B, 3B, 7B) (Hui et al., 2024). For value model in PPO, we use DeepSeek-R1-Distill-Qwen-1.5B (Guo et al., 2025) for all experiments.

**Benchmark.** We evaluate on a diverse suite of mathematical reasoning benchmarks. These include standard benchmarks such as GSM8K (Cobbe et al., 2021), MATH (Hendrycks et al., 2021), AS-Div (Miao et al., 2020), Carp (English Version) (Zhang et al., 2023), MAWPS (Koncel-Kedziorski et al., 2016), SVAMP (Patel et al., 2021), TabMWP (Lu et al., 2023), and OlympiadBench (He et al., 2024); Chinese mathematics collections like CMATH (Wei et al., 2023) and Gaokao 2024; and benchmarks from mathematics competitions, including the 2024/2025 AIME and the 2023/2024 AMC.

## G.2 EFFICIENT INCREMENTAL COMPUTATION OF HIGHER-ORDER METRICS

A naive computation of the temporal difference metrics would be computationally prohibitive. Our method's feasibility hinges on an efficient, incremental algorithm that computes the required metrics without redundant operations on the growing hidden state matrix $\mathbf{Z} \in \mathbb{R}^{T \times D}$.

The effective rank is derived from the singular values of the mean-centered hidden state matrix $\mathbf{Z}_c$. These are equivalent to the square roots of the eigenvalues of the centered Gram matrix $G = \mathbf{Z}_c \mathbf{Z}_c^\top$. Instead of recomputing $G_t$ from scratch at each time step t, our algorithm incrementally updates two sufficient statistics: the uncentered Gram matrix $U_t = Z_{1:t} Z_{1:t}^\top$ and the sum of hidden state vectors $s_t = \sum_{i=1}^{t} z_i$. When extending the analysis window, the new uncentered Gram matrix $U_t$ is constructed from the prior matrix $U_{t-s}$ and the new chunk of hidden states $\Delta Z_t = Z_{t-s+1:t}$. This update follows a recursive block matrix structure:

$$U_t = \begin{pmatrix} U_{t-s} & Z_{1:t-s}(\Delta Z_t)^\top \\ (\Delta Z_t)Z_{1:t-s}^\top & (\Delta Z_t)(\Delta Z_t)^\top \end{pmatrix} \quad (72)$$

From the efficiently updated $U_t$ and $s_t$, we can directly construct the centered Gram matrix $G_t$. Letting $\boldsymbol{\mu}_t = s_t/t$ be the mean vector and $\mathbf{1}_t$ be a column vector of ones, $G_t$ can be expressed as:

$$G_t = U_t - (Z_{1:t}\boldsymbol{\mu}_t)\mathbf{1}_t^\top - \mathbf{1}_t(Z_{1:t}\boldsymbol{\mu}_t)^\top + (\boldsymbol{\mu}_t^\top \boldsymbol{\mu}_t) \cdot (\mathbf{1}_t \mathbf{1}_t^\top) \quad (73)$$

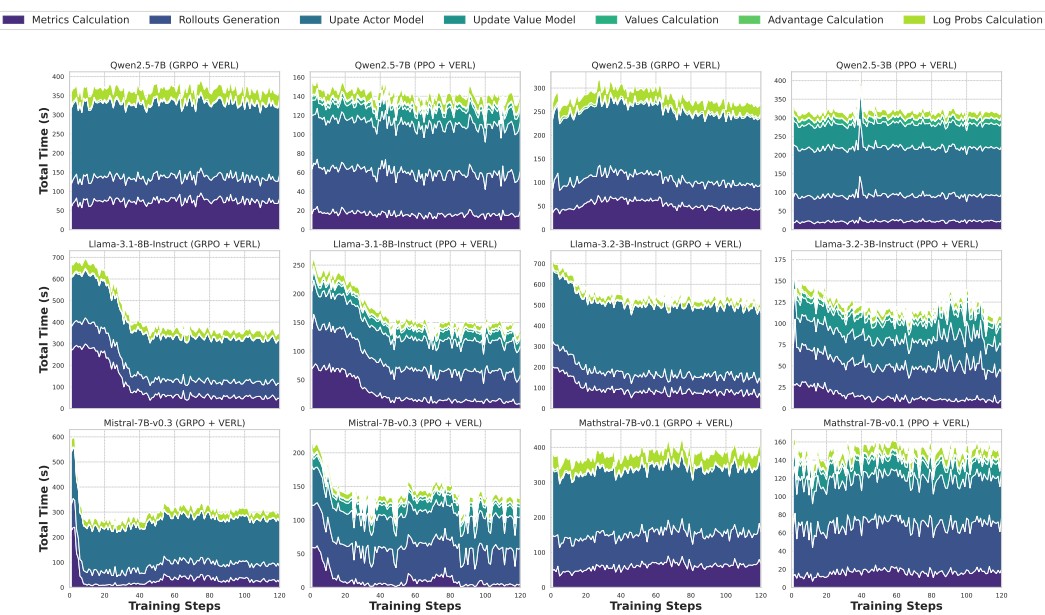

Figure 6: Time overhead of the main computation of RL Training.

This allows for the calculation of $G_t$ without re-accessing the full history of hidden states. The eigenvalues $\{\lambda_j\}$ of $G_t$ are then used to derive the effective rank. First, the singular values of the centered matrix are obtained, $\sigma_j = \sqrt{\lambda_j}$. These are normalized to form a probability distribution, $p_j = \sigma_j / \sum_k \sigma_k$. The effective rank is then the exponential of the Shannon entropy of this distribution:$\mathrm{erank}(Z_{c,t}) = \exp\left(-\sum_j p_j \log p_j\right)$. This pipeline efficiently yields a sequence of effective rank values, $m_{j \cdot s} = \mathrm{erank}(Z_{c,j \cdot s})$, at each stride $s$. From this sequence, we compute the instantaneous first-order difference, $\delta$, which compares the current value to the running average of all preceding values. This is defined recursively as:$\delta_{j \cdot s} = m_{j \cdot s} - \frac{1}{j-1}\sum_{k=1}^{j-1} m_{k \cdot s}$.

The computational advantage of this incremental approach is substantial. While the total cost for the series of eigenvalue decompositions $\mathcal{O}(T^4/s)$, is common to both methods, the cost of matrix construction differs significantly. The naive method's recalculation totals $\mathcal{O}(DT^3/s)$, whereas our incremental update method reduces this to $O(DT^2)$. This reduction of the polynomial dependency on sequence length T from cubic to quadratic is critical, as this term is multiplied by the large hidden dimension $D$, making it the dominant factor in practical performance and rendering the dense calculation of temporal dynamics feasible. In the worst-case regime where the sequence length $T$ exceeds the hidden dimension $D$, and both $D$ and the stride $s$ can be treated as constants. The naïve approach that reconstructs matrices independently at each stride has a matrix-construction cost scaling as $\mathcal{O}(T^2)$, VERL's incremental Gram/covariance updates scale as $\mathcal{O}(T)$. So asymptotically, our implementation has a strictly better dependency on $T$ than a naïve SVD-based design.

### G.3 TIME OVERHEAD OF VERL TRAINING

We conducted post-training with Zero RL on several base models. The Fig. 6 illustrates the time associated with each computational stage. The 'metrics calculation' component, which represents the cost of computing metrics for hidden states, accounts for an insignificant portion of the total processing time. This demonstrates that our method does not introduce substantial time overhead. To further stress-test the worst-case scenario, we deliberately compute ER, ERV, and ERA on the CPU rather than the GPU, and still observe that the additional time overhead remains negligible.

## H MORE EXPERIMENTS

### H.1 ANALYSIS OF RESPONSE-LEVEL HIDDEN STATES

As shown in Figs. 7 and 8, our analysis of response-level hidden states across additional LLMs confirms that the insights presented in Sec. 4.1 hold true for various base models and RL paradigms.

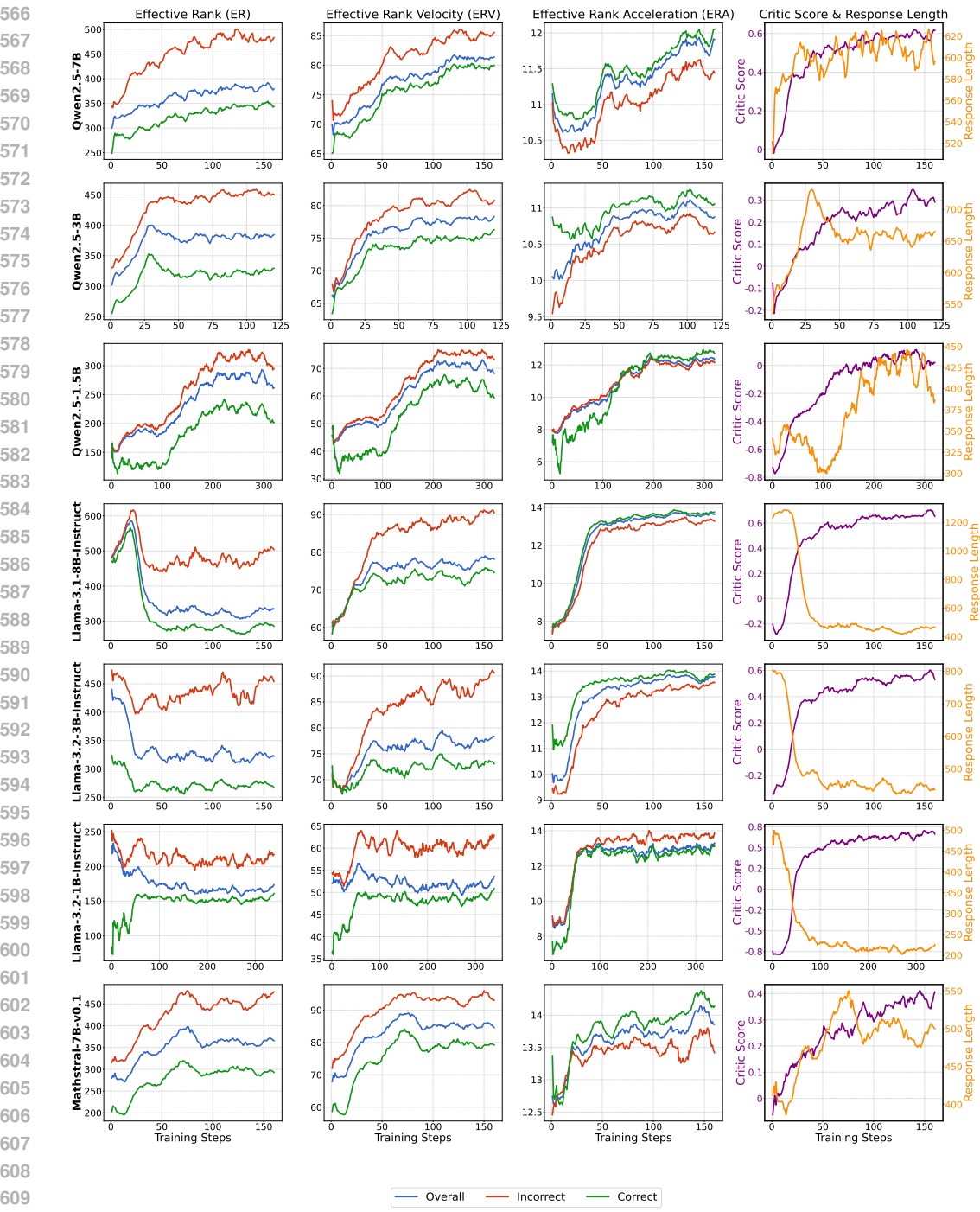

Figure 7: Visualization of response-level metrics for GRPO post-training. 'Overall' (blue) represents the metric across the entire data batch, while 'Correct' (green) and 'Incorrect' (red) show the metrics specifically for correctly and incorrectly classified samples, respectively. Data is smoothed using a rolling window of 10 steps to highlight underlying trends.

## H.2 ANALYSIS OF DATASET-LEVEL HIDDEN STATES

As shown in Figs. 9 and 10, our analysis of dataset-level hidden states across additional LLMs confirms that the insights presented in Sec. 4.2 hold for various base models and RL paradigms.

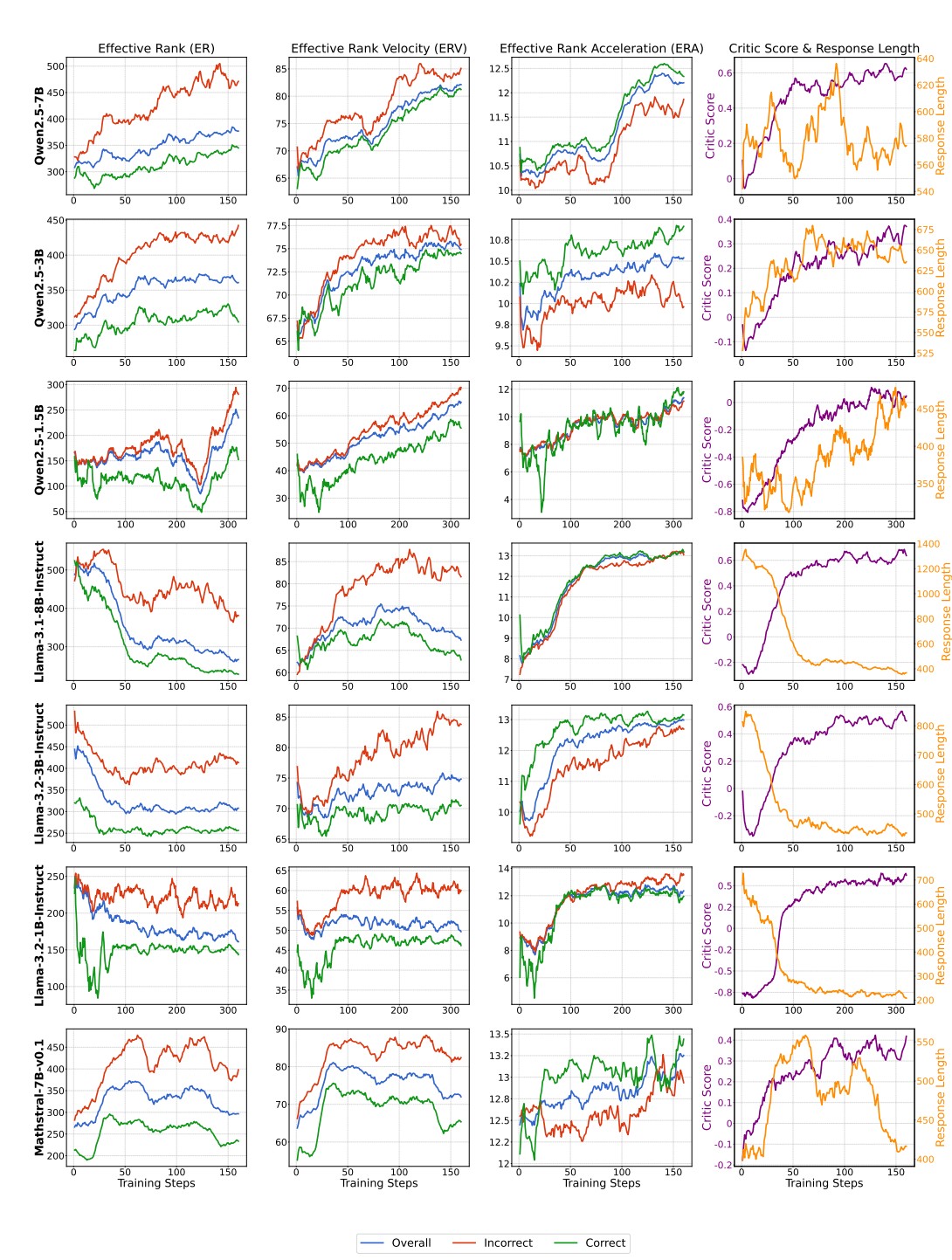

Figure 8: Visualization of response-level metrics for PPO post-training. 'Overall' (blue) represents the metric across the entire data batch, while 'Correct' (green) and 'Incorrect' (red) show the metrics specifically for correctly and incorrectly classified samples, respectively. Data is smoothed using a rolling window of 10 steps to highlight underlying trends.

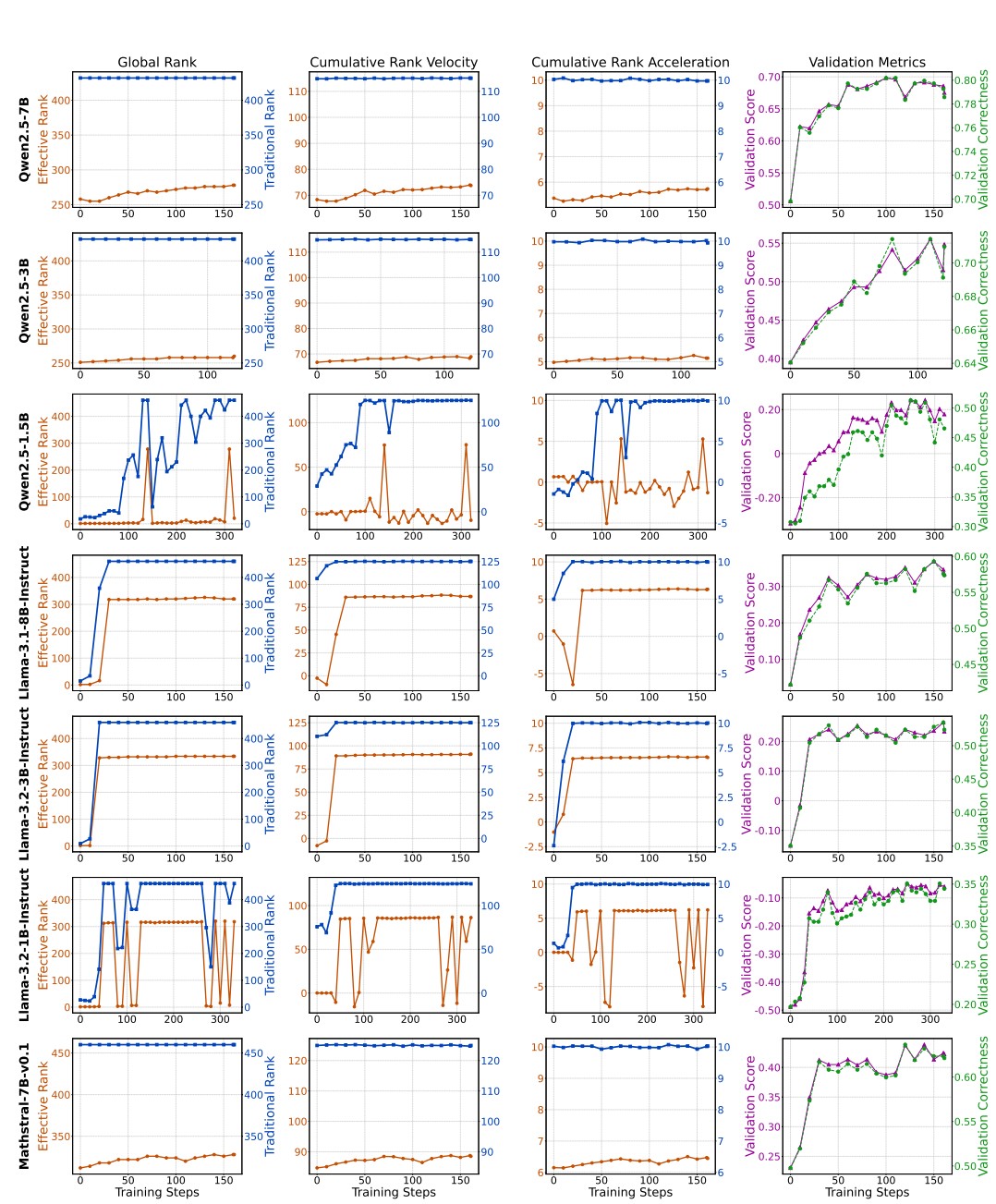

Figure 9: Visualization of dataset-level metrics for GRPO post-training

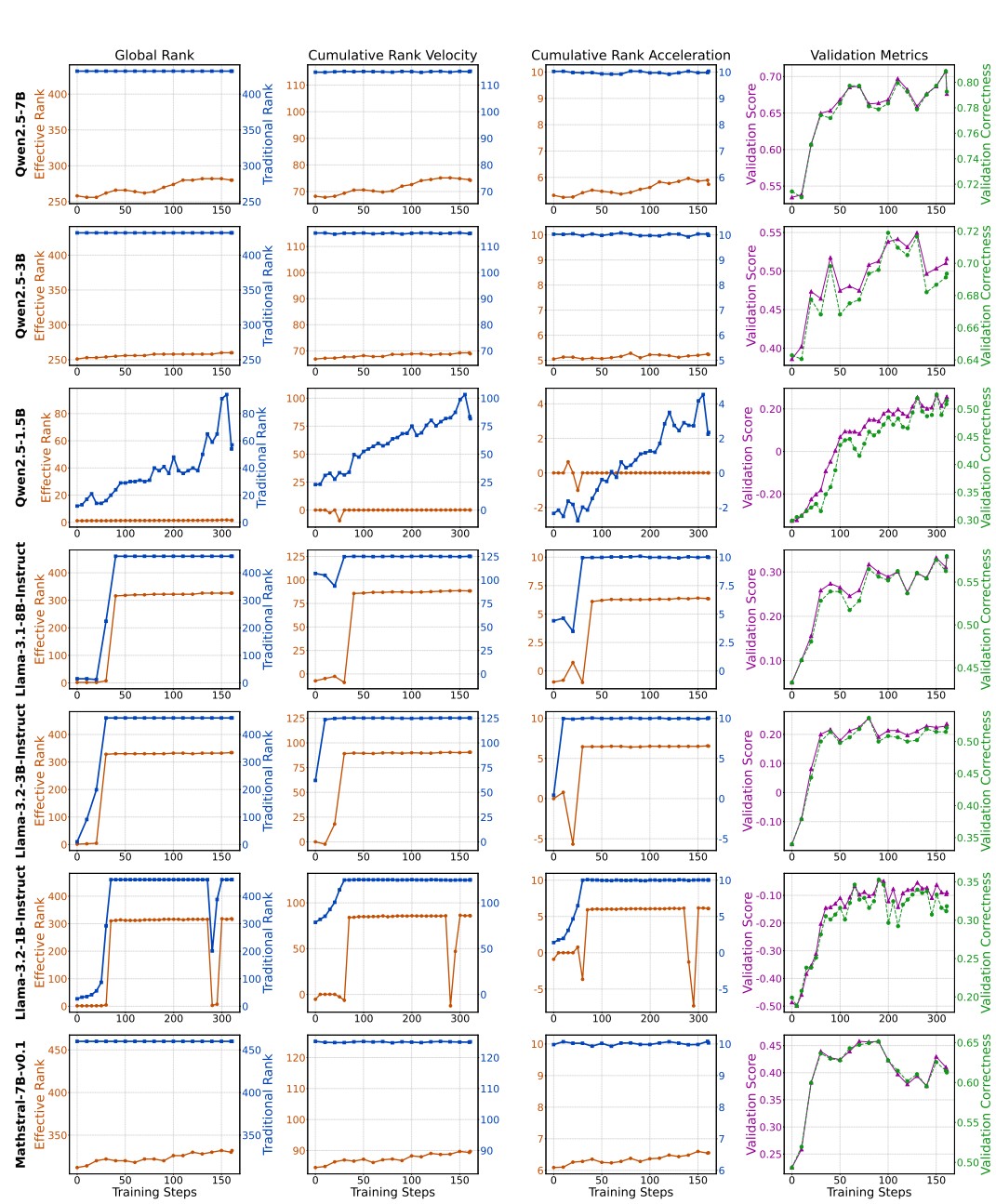

Figure 10: Visualization of dataset-level metrics for PPO post-training

## H.3 DETAILED ANALYSIS OF PASS@1 PERFORMANCE

As shown in Tab. 4, Pass@1 measures the model's ability to generate a correct answer in a single attempt, which directly reflects its exploitation ability. We fine-tune the base model by integrating our VERL-based Advantage method into two reinforcement learning paradigms, GRPO and PPO.

Table 4: Performance comparison of instruction-tuned models on mathematical reasoning benchmarks (Pass@1). "+ GRPO" and "+ PPO" denote reinforcement learning fine-tuning from the base model using GRPO and PPO, respectively. "w/ VERL." indicates the application of our VERL-based advantage to the corresponding RL algorithm. $\Delta$ represents the performance difference between the baseline RL method and its VERL-advanced variant. All results are reported in percentage (%).

| Model | AIME24 | AIME25 | AMC23 | AMC24 | ASDiv | Carp_En | CMATH | Gaokao 2024_I | Gaokao 2024_Mix | Gaokao MathCloze | GSM8K | MAWPS | Olympiad Bench | SVAMP | TabMWP | Avg. |
|---|---|---|---|---|---|---|---|---|---|---|---|---|---|---|---|---|
| **Llama-3.2-3B-Instruct** | 0.0 | 0.0 | 25.0 | 11.1 | 74.6 | 26.5 | 10.2 | 14.3 | 14.3 | 6.8 | 66.6 | 86.9 | 12.7 | 74.1 | 41.4 | 31.0 |
| + GRPO | 3.3 | 0.0 | 27.5 | 8.9 | 88.8 | 45.0 | 28.3 | 21.4 | 20.9 | 23.7 | 80.7 | 96.0 | 16.7 | 87.7 | 71.7 | 41.4 |
| + GRPO w/ VERL. | 13.3 | 6.7 | 25.0 | 11.1 | 89.3 | 45.4 | 46.2 | 14.3 | 22.0 | 22.9 | 81.7 | 96.0 | 17.6 | 87.8 | 72.3 | 43.4 |
| $\Delta_{GRPO}$ | +10.0 | +6.7 | -2.5 | +2.2 | +0.5 | +0.4 | +17.9 | -7.1 | +1.1 | -0.8 | +1.0 | +0.0 | +0.9 | +0.1 | +0.6 | +2.0 |
| + PPO | 10.0 | 3.3 | 22.5 | 13.3 | 87.9 | 46.4 | 21.2 | 7.1 | 16.5 | 20.3 | 81.4 | 95.5 | 17.8 | 86.8 | 71.0 | 40.1 |
| + PPO w/ VERL. | 10.0 | 3.3 | 25.0 | 11.1 | 88.7 | 46.0 | 30.7 | 14.3 | 19.8 | 27.1 | 82.9 | 95.7 | 17.3 | 85.8 | 71.3 | 41.9 |
| $\Delta_{PPO}$ | +0.0 | +0.0 | +2.5 | -2.2 | +0.8 | -0.4 | +9.5 | +7.2 | +3.3 | +6.8 | +1.5 | +0.2 | -0.5 | -1.0 | +0.3 | +1.9 |
| **Llama-3.1-8B-Instruct** | 0.0 | 3.3 | 17.5 | 8.9 | 48.0 | 34.1 | 18.5 | 0.0 | 15.4 | 16.9 | 47.4 | 43.5 | 10.4 | 48.5 | 34.3 | 23.1 |
| + GRPO | 6.7 | 0.0 | 22.5 | 15.6 | 90.3 | 42.4 | 60.7 | 7.1 | 14.3 | 32.2 | 88.4 | 96.4 | 19.7 | 88.5 | 82.7 | 44.5 |
| + GRPO w/ VERL. | 10.0 | 3.3 | 32.5 | 15.6 | 90.7 | 45.0 | 72.7 | 14.3 | 14.3 | 30.5 | 88.6 | 96.9 | 21.3 | 88.4 | 83.1 | 47.2 |
| $\Delta_{GRPO}$ | +3.3 | +3.3 | +10.0 | +0.0 | +0.4 | +2.6 | +12.0 | +7.2 | +0.0 | -1.7 | +0.2 | +0.5 | +1.6 | -0.1 | +0.4 | +2.7 |
| + PPO | 6.7 | 0.0 | 30.0 | 17.8 | 89.8 | 42.0 | 60.0 | 0.0 | 14.3 | 25.4 | 86.4 | 95.7 | 18.2 | 88.6 | 82.3 | 43.8 |
| + PPO w/ VERL. | 10.0 | 0.0 | 35.0 | 13.3 | 90.7 | 42.6 | 62.0 | 14.3 | 22.0 | 28.8 | 87.3 | 96.6 | 19.1 | 88.1 | 83.0 | 46.2 |
| $\Delta_{PPO}$ | +3.3 | +0.0 | +5.0 | -4.5 | +0.9 | +0.6 | +2.0 | +14.3 | +7.7 | +3.4 | +0.9 | +0.9 | +0.9 | -0.5 | +0.7 | +2.4 |
| **Qwen2.5-3B** | 6.7 | 0.0 | 20.0 | 24.4 | 90.7 | 54.7 | 76.7 | 0.0 | 22.0 | 41.5 | 80.7 | 95.1 | 23.0 | 84.3 | 71.3 | 46.1 |
| + GRPO | 3.3 | 0.0 | 40.0 | 22.2 | 92.6 | 56.0 | 82.7 | 7.1 | 27.5 | 42.4 | 82.8 | 96.6 | 23.6 | 89.0 | 81.4 | 49.8 |
| + GRPO w/ VERL. | 6.7 | 0.0 | 30.0 | 17.8 | 92.6 | 56.9 | 84.8 | 21.4 | 33.0 | 49.2 | 82.2 | 96.4 | 24.4 | 88.5 | 81.0 | 51.0 |
| $\Delta_{GRPO}$ | +3.4 | +0.0 | -10.0 | -4.4 | +0.0 | +0.9 | +2.1 | +14.3 | +5.5 | +6.8 | -0.6 | -0.2 | +0.8 | -0.5 | -0.4 | +1.2 |
| + PPO | 3.3 | 0.0 | 32.5 | 15.6 | 92.8 | 56.5 | 83.2 | 0.0 | 28.6 | 50.0 | 81.7 | 96.6 | 24.4 | 86.0 | 80.8 | 48.8 |
| + PPO w/ VERL. | 6.7 | 0.0 | 32.5 | 17.8 | 92.6 | 57.0 | 84.3 | 21.4 | 29.7 | 47.5 | 81.8 | 96.5 | 24.6 | 88.3 | 81.4 | 50.8 |
| $\Delta_{PPO}$ | +3.4 | +0.0 | +0.0 | +2.2 | -0.2 | +0.5 | +1.1 | +21.4 | +1.1 | -2.5 | +0.1 | -0.1 | +0.2 | +2.3 | +0.6 | +2.0 |
| **Qwen2.5-7B** | 6.7 | 0.0 | 45.0 | 15.6 | 91.4 | 55.8 | 86.7 | 42.9 | 33.0 | 49.2 | 85.8 | 95.4 | 25.8 | 88.5 | 82.8 | 53.6 |
| + GRPO | 10.0 | 6.7 | 55.0 | 26.7 | 94.8 | 60.2 | 91.7 | 14.3 | 34.1 | 64.4 | 90.2 | 97.6 | 36.1 | 92.8 | 91.3 | 57.7 |
| + GRPO w/ VERL. | 13.3 | 10.0 | 50.0 | 28.9 | 95.0 | 60.8 | 90.7 | 35.7 | 35.2 | 69.5 | 89.2 | 97.7 | 35.4 | 92.9 | 91.9 | 59.8 |
| $\Delta_{GRPO}$ | +3.3 | +3.3 | -5.0 | +2.2 | +0.2 | +0.6 | -1.0 | +21.4 | +1.1 | +5.1 | -1.0 | +0.1 | -0.7 | +0.1 | +0.6 | +2.1 |
| + PPO | 6.7 | 3.3 | 50.0 | 33.3 | 94.9 | 59.6 | 89.8 | 28.6 | 31.9 | 63.6 | 89.1 | 97.3 | 36.1 | 92.8 | 90.8 | 57.9 |
| + PPO w/ VERL. | 10.0 | 6.7 | 52.5 | 33.3 | 94.8 | 60.0 | 90.3 | 28.6 | 34.1 | 66.9 | 90.2 | 97.8 | 36.1 | 92.5 | 90.6 | 59.0 |
| $\Delta_{PPO}$ | +3.3 | +3.3 | +2.5 | +0.0 | -0.1 | +0.4 | +0.5 | +0.0 | +2.2 | +3.3 | +1.1 | +0.5 | +0.0 | -0.3 | -0.2 | +1.1 |
| **Mathstral-7B-v0.1** | 0.0 | 0.0 | 12.5 | 8.9 | 87.1 | 51.1 | 74.2 | 28.6 | 33.0 | 31.4 | 81.6 | 93.8 | 17.9 | 87.7 | 54.7 | 44.2 |
| + GRPO | 0.0 | 0.0 | 47.5 | 17.8 | 92.9 | 55.9 | 81.3 | 35.7 | 44.0 | 49.2 | 88.1 | 97.6 | 25.6 | 93.0 | 81.5 | 54.0 |
| + GRPO w/ VERL. | 6.7 | 0.0 | 45.0 | 20.0 | 93.3 | 55.5 | 81.5 | 50.0 | 40.7 | 46.6 | 89.5 | 97.2 | 29.3 | 90.7 | 83.5 | 55.3 |
| $\Delta_{GRPO}$ | +6.7 | +0.0 | -2.5 | +2.2 | +0.4 | -0.4 | +0.2 | +14.3 | -3.3 | -2.6 | +1.4 | -0.4 | +3.7 | -2.3 | +2.0 | +1.3 |
| + PPO | 6.7 | 3.3 | 32.5 | 20.0 | 90.9 | 51.8 | 78.3 | 42.9 | 37.4 | 49.2 | 87.0 | 96.0 | 28.4 | 89.9 | 70.7 | 52.3 |
| + PPO w/ VERL. | 10.0 | 0.0 | 27.5 | 22.2 | 93.0 | 53.8 | 78.2 | 42.9 | 51.6 | 48.3 | 87.4 | 96.7 | 26.1 | 89.6 | 84.1 | 54.1 |
| $\Delta_{PPO}$ | +3.3 | -3.3 | -5.0 | +2.2 | +2.1 | +2.0 | -0.1 | +0.0 | +14.2 | -0.9 | +0.4 | +0.7 | -2.3 | -0.3 | +13.4 | +1.8 |
| **Mistral-7B-v0.3** | 0.0 | 0.0 | 10.0 | 0.0 | 40.5 | 12.4 | 21.8 | 14.3 | 13.2 | 3.4 | 24.0 | 50.8 | 1.6 | 39.1 | 30.6 | 17.4 |
| + GRPO | 0.0 | 0.0 | 2.5 | 4.4 | 58.2 | 11.1 | 42.3 | 0.0 | 15.4 | 5.1 | 52.4 | 79.2 | 3.0 | 47.6 | 37.7 | 23.9 |
| + GRPO w/ VERL. | 0.0 | 0.0 | 7.5 | 2.2 | 59.1 | 15.0 | 43.0 | 0.0 | 6.6 | 4.2 | 40.3 | 69.5 | 2.8 | 57.5 | 53.0 | 24.0 |
| $\Delta_{GRPO}$ | +0.0 | +0.0 | +5.0 | -2.2 | +0.9 | +3.9 | +0.7 | +0.0 | -8.8 | -0.9 | -12.1 | -9.7 | -0.2 | +9.9 | +15.3 | +0.1 |
| + PPO | 0.0 | 0.0 | 0.0 | 0.0 | 8.9 | 6.6 | 7.7 | 7.1 | 11.0 | 2.5 | 3.3 | 8.6 | 2.1 | 6.9 | 12.0 | 5.1 |
| + PPO w/ VERL. | 0.0 | 0.0 | 2.5 | 0.0 | 44.7 | 10.6 | 35.7 | 7.1 | 16.5 | 5.1 | 28.8 | 70.7 | 2.4 | 57.5 | 35.1 | 21.1 |
| $\Delta_{PPO}$ | +0.0 | +0.0 | +2.5 | +0.0 | +35.8 | +4.0 | +28.0 | +0.0 | +5.5 | +2.6 | +25.5 | +62.1 | +0.3 | +50.6 | +23.1 | +16.0 |

## H.4 DETAILED ANALYSIS OF PASS@$k$ PERFORMANCE

As shown in Tab. 5, which provides a comprehensive analysis of the models' performance on the Pass@$k$ metric, which is a direct measure of the model's exploration ability. As a supplement to the main paper's discussion, it presents the detailed performance of our VERL-based models across a variety of mathematical reasoning benchmarks. These results demonstrate that our method consistently leads to significant improvements, confirming its effectiveness in enhancing the models' exploration capabilities.

Table 5: Performance comparison of instruction-tuned models under diverse decoding settings (Pass@$k$). All results are reported in percentage (%).

| Model | MATH500 (Pass@16) | AMC23 (Pass@128) | AMC24 (Pass@128) | AIME24 (Pass@256) | AIME25 (Pass@256) | Avg. |
|---|---|---|---|---|---|---|
| **Llama-3.2-3B-Instruct** | 79.8 | 93.5 | 51.5 | 40.0 | 30.0 | 58.96 |
| + GRPO | 80.2 | 95.4 | 60.6 | 40.0 | 30.0 | 61.24 |
| + GRPO w/ VERL. | 80.6 | 95.7 | 59.0 | 50.0 | 36.7 | 64.40 |
| $\Delta_{\text{GRPO}}$ | +0.4 | +0.3 | −1.6 | +10.0 | +6.7 | +3.16 |
| + PPO | 82.2 | 94.5 | 57.0 | 46.7 | 36.7 | 63.42 |
| + PPO w/ VERL. | 82.4 | 94.7 | 57.8 | 46.7 | 40.0 | 64.32 |
| $\Delta_{\text{PPO}}$ | +0.2 | +0.2 | +0.8 | +0.0 | +3.3 | +0.90 |
| **Llama-3.1-8B-Instruct** | 79.8 | 94.6 | 57.4 | 46.7 | 36.7 | 63.04 |
| + GRPO | 83.4 | 94.9 | 56.9 | 53.3 | 36.7 | 65.04 |
| + GRPO w/ VERL. | 83.4 | 95.1 | 63.1 | 50.0 | 36.7 | 65.66 |
| $\Delta_{\text{GRPO}}$ | +0.0 | +0.2 | +6.2 | −3.3 | +0.0 | +0.62 |
| + PPO | 79.2 | 92.4 | 59.0 | 46.7 | 36.7 | 62.80 |
| + PPO w/ VERL. | 82.4 | 91.9 | 60.0 | 53.3 | 36.7 | 64.86 |
| $\Delta_{\text{PPO}}$ | +3.2 | −0.5 | +1.0 | +6.6 | +0.0 | +2.06 |
| **Qwen2.5-3B** | 86.0 | 96.7 | 69.0 | 56.7 | 40.0 | 69.68 |
| + GRPO | 86.6 | 92.2 | 68.5 | 46.7 | 40.0 | 66.80 |
| + GRPO w/ VERL. | 87.6 | 95.9 | 67.8 | 53.3 | 43.3 | 69.58 |
| $\Delta_{\text{GRPO}}$ | +1.0 | +3.7 | −0.7 | +6.6 | +3.3 | +2.78 |
| + PPO | 87.8 | 96.5 | 67.9 | 43.3 | 43.3 | 67.76 |
| + PPO w/ VERL. | 88.2 | 96.8 | 67.3 | 53.3 | 43.3 | 69.78 |
| $\Delta_{\text{PPO}}$ | +0.4 | +0.3 | −0.6 | +10.0 | +0.0 | +2.02 |
| **Qwen2.5-7B** | 90.6 | 98.4 | 73.7 | 60.0 | 60.0 | 76.54 |
| + GRPO | 90.8 | 97.8 | 78.3 | 56.7 | 50.0 | 74.72 |
| + GRPO w/ VERL. | 91.4 | 98.3 | 79.0 | 63.3 | 60.0 | 78.40 |
| $\Delta_{\text{GRPO}}$ | +0.6 | +0.5 | +0.7 | +6.6 | +10.0 | +3.68 |
| + PPO | 91.2 | 98.6 | 74.3 | 53.3 | 56.7 | 74.82 |
| + PPO w/ VERL. | 91.4 | 98.0 | 74.4 | 56.7 | 66.7 | 77.44 |
| $\Delta_{\text{PPO}}$ | +0.2 | −0.6 | +0.1 | +3.4 | +10.0 | +2.62 |
| **Mathstral-7B-v0.1** | 80.4 | 88.5 | 60.9 | 43.3 | 36.7 | 61.96 |
| + GRPO | 84.8 | 87.3 | 69.2 | 36.7 | 40.0 | 63.60 |
| + GRPO w/ VERL. | 87.0 | 97.0 | 76.9 | 50.0 | 50.0 | 72.18 |
| $\Delta_{\text{GRPO}}$ | +2.2 | +9.7 | +7.7 | +13.3 | +10.0 | +8.58 |
| + PPO | 82.4 | 91.7 | 70.7 | 53.3 | 40.0 | 67.62 |
| + PPO w/ VERL. | 84.8 | 93.8 | 69.9 | 53.3 | 46.7 | 69.70 |
| $\Delta_{\text{PPO}}$ | +2.4 | +2.1 | −0.8 | +0.0 | +6.7 | +2.08 |
| **Mistral-7B-v0.3** | 36.0 | 73.5 | 39.6 | 20.0 | 16.7 | 37.16 |
| + GRPO | 33.0 | 63.2 | 36.0 | 10.0 | 10.0 | 30.44 |
| + GRPO w/ VERL. | 34.4 | 64.5 | 38.0 | 16.7 | 13.3 | 33.38 |
| $\Delta_{\text{GRPO}}$ | +1.4 | +1.3 | +2.0 | +6.7 | +3.3 | +2.94 |
| + PPO | 21.8 | 46.4 | 25.1 | 6.7 | 6.7 | 21.34 |
| + PPO w/ VERL. | 19.2 | 46.5 | 30.1 | 3.3 | 13.3 | 22.48 |
| $\Delta_{\text{PPO}}$ | −2.6 | +0.1 | +5.0 | −3.4 | +6.6 | +1.14 |

## H.5 ABLATION ON THE CHOICE OF HIDDEN LAYER

We focus on the final hidden layer because our exploration/exploitation metrics are defined in the *semantic* space along the reasoning trajectory, and prior interpretability work (Jing et al., 2025; Sajjad et al., 2022; Valeriani et al., 2023; Matthews et al., 2024; Servedio et al., 2025; Zhang et al.,

Table 6: Comparison of GRPO + VERL using an intermediate layer (layer 14) versus the final layer, evaluated by pass@1 on multiple math benchmarks. Using the last layer yields the strongest average improvement.

| Method | aime24 | aime25 | amc23 | amc24 | asdiv | carp_en | cmath | gaokao24_I | gaokao24_mix | gaokao_math_cloze | gsm8k | mawps | olympiadbench | svamp | tabmwp | Avg. |
|---|---|---|---|---|---|---|---|---|---|---|---|---|---|---|---|---|
| Llama-3.2-3B-Instruct | 0.0 | 0.0 | 25.0 | 11.1 | 74.6 | 26.5 | 10.2 | 14.3 | 14.3 | 6.8 | 66.6 | 86.9 | 12.7 | 74.1 | 41.4 | 30.97 |
| GRPO | 3.3 | 0.0 | 27.5 | 8.9 | 88.8 | 45.0 | 28.3 | 21.4 | 20.9 | 23.7 | 80.7 | 96.0 | 16.7 | 87.7 | 71.7 | 41.37 |
| GRPO w/ VERL (layer = 14) | 10.0 | 0.0 | 27.5 | 11.1 | 88.6 | 43.6 | 30.7 | 21.4 | 16.5 | 22.0 | 81.9 | 95.5 | 18.1 | 87.0 | 71.4 | 41.69 |
| GRPO w/ VERL (layer = last) | 13.3 | 6.7 | 25.0 | 11.1 | 89.3 | 45.4 | 46.2 | 14.3 | 22.0 | 22.9 | 81.7 | 96.0 | 17.6 | 87.8 | 72.3 | **43.44** |

Table 7: Comparison of GRPO + VERL using an intermediate layer (layer 14) versus the final layer, evaluated by pass@k on several math benchmarks. Again, using the last layer yields the best average improvement.

| Method | math500@16 | amc23@128 | amc24@128 | aime24@256 | aime25@256 | Avg. |
|---|---|---|---|---|---|---|
| Llama-3.2-3B-Instruct | 79.8 | 93.5 | 51.5 | 40.0 | 30.0 | 58.96 |
| GRPO | 80.2 | 95.4 | 60.6 | 40.0 | 30.0 | 61.24 |
| GRPO w/ VERL (layer = 14) | 81.0 | 92.9 | 57.0 | 40.0 | 36.7 | 61.52 |
| GRPO w/ VERL (layer = last) | 80.6 | 95.7 | 59.0 | 50.0 | 36.7 | **64.40** |

2025) suggests that the last layers are most aligned with semantic meaning and model predictions. Empirically, using the final layer gives consistently better performance than using an intermediate layer: for example, GRPO + VERL with the last layer improves the average pass@1 from 41.69% (layer 14) to 43.44%, and the average pass@k from 61.52% (layer 14) to 64.40%. This subsection provides the detailed analysis supporting our design choice to base VERL on the final layer.

Intermediate layers in large language models can encode rich features. However, our notion of exploration and exploitation is explicitly defined in the semantic space of a reasoning trajectory. Existing interpretability studies (Jing et al., 2025; Sajjad et al., 2022; Valeriani et al., 2023; Matthews et al., 2024; Servedio et al., 2025; Zhang et al., 2025) indicate that hidden states in the last layers are most tightly aligned with token-level semantics and the model's predictive distribution, while mid-layer representations tend to mix morphology, syntax, and other lower-level or task-specific signals. For this reason, we consider the final layer more suitable for semantic diversity metrics such as ER and ERV.

**Intermediate-layer vs. final-layer VERL (pass@1).** We implemented VERL on an intermediate layer (layer 14) and on the last layer, keeping everything else fixed. The pass@1 results are reported in Table 6. Both VERL variants improve over GRPO, but the last-layer version clearly gives the strongest overall gains in average pass@1.

**Intermediate-layer vs. final-layer VERL (pass@k).** A similar pattern holds for pass@k, summarized in Table 7. Using the last layer yields the best average improvement, especially on the more challenging AIME-style benchmarks, further supporting the choice of the final hidden layer as the basis for our semantic exploration and exploitation metrics.

# I CASE STUDY

For all case studies presented in this section, we employ Qwen2.5-7B as the base model. The answers shown in the gray boxes are generated by Qwen2.5-7B trained with vanilla GRPO for 120 steps, while those in the purple boxes are produced by Qwen2.5-7B trained with GRPO augmented with the auxiliary shaping term—our proposed method VERL—for 120 steps.

## I.1 CASE STUDY FOR PASS@1 SETTING

**Case Study** I. For the case shown in Fig. 11, the model trained with vanilla GRPO incorrectly judges $-1 < -13$, whereas the model trained with GRPO augmented with the auxiliary shaping term gives the correct answer. This example highlights the limitation of vanilla GRPO in exploitation: it fails to consistently apply basic numerical commonsense—namely, that larger absolute values correspond to smaller magnitudes for negative numbers. By contrast, introducing the Auxiliary Shaping Term strengthens exploitation capacity, enabling the model to make such comparisons more reliably.

---

**Question**

What is the smallest possible integer value of $x$ if $12 \div (x + 1)$ is an integer?

---

**Ground Truth**

-13

---

**Vanilla GRPO**

To find the smallest possible integer value of $x$ such that $\frac{12}{x+1}$ is an integer, we need to consider the factors of 12. The expression $\frac{12}{x+1}$ will be an integer if $x + 1$ is a divisor of 12. Let's list the divisors of 12:
1. The divisors of 12 are: $\pm 1, \pm 2, \pm 3, \pm 4, \pm 6, \pm 12$.
2. We need to find the smallest $x$, so we start with the smallest positive divisor, which is 1:

$$x + 1 = 1 \implies x = 1 - 1 \implies x = 0$$

3. To confirm, we substitute $x = 0$ back into the original expression:

$$\frac{12}{0 + 1} = \frac{12}{1} = 12$$

Since 12 is an integer, $x = 0$ is indeed a valid solution.
4. We should also consider the negative divisors to ensure we have the smallest possible integer value. The next smallest divisor is -1:

$$x + 1 = -1 \implies x = -1 - 1 \implies x = -2$$

5. Substituting $x = -2$ back into the original expression:

$$\frac{12}{-2 + 1} = \frac{12}{-1} = -12$$

Since -12 is an integer, $x = -2$ is also a valid solution, but it is not smaller than $x = 0$.
Therefore, the smallest possible integer value of $x$ is

$$\boxed{0}$$

---

**GRPO with Auxiliary Shaping Term**

To find the smallest possible integer value of $x$ such that $\frac{12}{x+1}$ is an integer, we need to determine the divisors of 12. The divisors of 12 are $\pm 1, \pm 2, \pm 3, \pm 4, \pm 6, \pm 12$. For $\frac{12}{x+1}$ to be an integer, $x + 1$ must be one of these divisors. We want the smallest possible $x$, so we should consider the smallest positive divisor, which is 1.
If $x + 1 = 1$, then $x = 0$.
We should also consider the negative divisors to ensure we find the smallest possible integer value of $x$. The smallest negative divisor is $-12$. If $x + 1 = -12$, then $x = -13$.
Comparing $x = 0$ and $x = -13$, the smallest possible integer value of $x$ is $-13$.
Therefore, the final answer is

$$\boxed{-13}$$

---

Figure 11: Case study: Vanilla GRPO fails to exploit numerical commonsense when comparing negative values (-1 vs. -13), while our method gives the correct answer.

**Case Study** II.    For the case in Fig. 12, the crucial distinction lies in how the models interpret the condition "No house faces two different streets." The vanilla GRPO–trained model incorrectly infers that each street contains only 10 houses, overlooking the structural implication of the condition. In contrast, the model trained with GRPO augmented by the auxiliary shaping term exploits this commonsense constraint more effectively: it decomposes the problem step by step, correctly deduces that each street has 20 houses, and thereby arrives at the correct total.

---

**Question**

In my neighborhood, there are six streets. There are 10 houses on each side of each street. No house faces two different streets. How many houses are in my neighborhood?

---

**Ground Truth**

120

---

**Vanilla GRPO**

To determine the total number of houses in the neighborhood, we can follow these steps:
1. **Identify the number of streets and houses per street:** - There are 6 streets. - Each street has 10 houses on each side.
2. **Calculate the number of houses on one street:** Since there are 10 houses on each side of a street, and each house faces only one street, the total number of houses on one street is 10 (since they are all on the same street).
3. **Calculate the total number of houses in the neighborhood:** There are 6 streets, and each street has 10 houses. Therefore, the total number of houses is $6 \times 10 = 60$.
Putting the final answer within ☐, we get:

$$\boxed{60}$$

---

**GRPO with Auxiliary Shaping Term**

To find the total number of houses in the neighborhood, let's break down the problem step by step.
1. There are 6 streets in the neighborhood.
2. Each street has houses on both sides.
3. There are 10 houses on each side of every street.
4. No house faces two different streets, meaning each house is counted only once.
First, calculate the number of houses on one side of a single street:

$$10 \text{ houses/side}$$

Since there are houses on both sides of each street, multiply by 2:

$$10 \text{ houses/side} \times 2 \text{ sides/street} = 20 \text{ houses/street}$$

Now, since there are 6 streets, multiply the number of houses per street by the number of streets:

$$20 \text{ houses/street} \times 6 \text{ streets} = 120 \text{ houses}$$

So, the total number of houses in the neighborhood is

$$\boxed{120}$$

---

Figure 12: Case study: Vanilla GRPO overlooks the constraint and infers 10 houses per street, while our method exploits it correctly and infers 20.

**Case Study** III.    As shown in Fig. 13, the task concerns solving a differential equation under the constraint of finding a purely exponential solution rather than a general one. The model trained with vanilla GRPO overlooks this constraint and defaults to the generic procedure for obtaining the full solution. In contrast, the model trained with GRPO augmented by the auxiliary shaping term respects the constraint and produces the correct purely exponential answer. This case illustrates how the auxiliary shaping term enhances exploitation by enforcing adherence to problem-specific constraints.

I.2    CASE STUDY FOR PASS@16 SETTING

**Case Study** I.    As shown in Fig. 14, the question is a simple elementary-level math problem requiring the integer solution $x \geqslant 1, x \neq 2$. While the vanilla GRPO model consistently outputs 3, which is incorrect, the model trained with the auxiliary shaping term produces 1 in 5 out of 16 rollouts, the correct answer. This case highlights that the auxiliary shaping term enhances the model's exploration ability, enabling it to consider a wider range of possibly valid solutions.

**Question**

Find a purely exponential solution of $\frac{d^4 x}{dt^4} - x = e^{-2t}$.

**Ground Truth**

$\frac{e^{-2t}}{15}$

**Vanilla GRPO**

To find a purely exponential solution of the differential equation ($\frac{d^4 x}{dt^4} - x = e^{-2t}$), we will follow these steps:

1. **Find the complementary solution by solving the homogeneous equation:**

$$\frac{d^4 x}{dt^4} - x = 0$$

The characteristic equation is:

$$r^4 - 1 = 0$$

Solving for $r$, we get:

$$r^4 = 1 \implies r = \pm 1, \pm i$$

Therefore, the complementary solution is:

$$x_c(t) = C_1 e^t + C_2 e^{-t} + C_3 \cos(t) + C_4 \sin(t)$$

2. **Find a particular solution using the method of undetermined coefficients:**
Since the right-hand side of the non-homogeneous equation is $e^{-2t}$, we assume a particular solution of the form:

$$x_p(t) = A e^{-2t}$$

Taking the fourth derivative of $x_p(t)$:

$$\frac{d^4}{dt^4}(A e^{-2t}) = A(-2)^4 e^{-2t} = 16 A e^{-2t}$$

Substituting $x_p(t)$ and its fourth derivative into the original differential equation:

$$16 A e^{-2t} - A e^{-2t} = e^{-2t}$$

Simplifying, we get:

$$(16A - A)e^{-2t} = e^{-2t} \implies 15 A e^{-2t} = e^{-2t}$$

Therefore:

$$15A = 1 \implies A = \frac{1}{15}$$

So, the particular solution is:

$$x_p(t) = \frac{1}{15} e^{-2t}$$

3. **Combine the complementary and particular solutions:**
The general solution to the differential equation is:

$$x(t) = x_c(t) + x_p(t) = C_1 e^t + C_2 e^{-t} + C_3 \cos(t) + C_4 \sin(t) + \frac{1}{15} e^{-2t}$$

Thus, the final answer is:

$$\boxed{C_1 e^t + C_2 e^{-t} + C_3 \cos(t) + C_4 \sin(t) + \frac{1}{15} e^{-2t}}$$

Figure 13: Case study: Vanilla GRPO ignores the constraint of a purely exponential solution, while our method enforces it and yields the correct result.

**GRPO with Auxiliary Shaping Term**

To find a purely exponential solution to the differential equation

$$\frac{d^4 x}{dt^4} - x = e^{-2t}$$

, we will follow these steps:

1. **Find the complementary solution by solving the homogeneous equation:**

$$\frac{d^4 x}{dt^4} - x = 0$$

The characteristic equation is:

$$r^4 - 1 = 0$$

Solving for $r$, we get:

$$r^4 = 1 \implies r = \pm 1, \pm i$$

Therefore, the complementary solution is:

$$x_c(t) = C_1 e^t + C_2 e^{-t} + C_3 \cos(t) + C_4 \sin(t)$$

2. **Find a particular solution using the method of undetermined coefficients:** Since the right-hand side of the non-homogeneous equation is $e^{-2t}$, we assume a particular solution of the form:

$$x_p(t) = A e^{-2t}$$

Taking the fourth derivative of $x_p(t)$:

$$\frac{d^4 x_p}{dt^4} = 16 A e^{-2t}$$

Substituting $x_p(t)$ and its fourth derivative into the original differential equation:

$$16 A e^{-2t} - A e^{-2t} = e^{-2t}$$

Simplifying, we get:

$$15 A e^{-2t} = e^{-2t}$$

Solving for $A$:

$$15 A = 1 \implies A = \frac{1}{15}$$

Therefore, the particular solution is:

$$x_p(t) = \frac{1}{15} e^{-2t}$$

3. **Combine the complementary and particular solutions:** The general solution to the differential equation is:

$$x(t) = x_c(t) + x_p(t) = C_1 e^t + C_2 e^{-t} + C_3 \cos(t) + C_4 \sin(t) + \frac{1}{15} e^{-2t}$$

Thus, the purely exponential solution is:

$$\boxed{\frac{1}{15} e^{-2t}}$$

Figure 13: Case study: Vanilla GRPO ignores the constraint of a purely exponential solution, while our method enforces it and yields the correct result (continued).

**Question**

Given $f(x) = \frac{\sqrt{x-1}}{x-2}$, what is the smallest possible integer value for $x$ such that $f(x)$ has a real number value?

**Ground Truth**

1

Figure 14: Case study demonstrating improved exploration: the auxiliary shaping term enables the model to explore and reach the correct solution.

**Vallina GRPO#1-#8**

---

*(Details omitted as they are not crucial for this case study.)*

1. **Numerator Condition**: The numerator is $\sqrt{x-1}$. For the square root to be defined and result in a real number, the expression inside the square root must be non-negative.

$$x - 1 \geqslant 0 \implies x \geqslant 1$$

2. **Denominator Condition**: The denominator is $x - 2$. For the function to be defined, the denominator must not be zero.

$$x - 2 \neq 0 \implies x \neq 2$$

Combining these two conditions, we get:

$$x \geqslant 1 \quad \text{and} \quad x \neq 2$$

The smallest integer that satisfies $x \geqslant 1$ and $x \neq 2$ is $x = 3$.
Therefore, the smallest possible integer value for $x$ such that $f(x)$ has a real number value is $\boxed{3}$.

---

To determine the smallest possible integer value for $x$ such that $f(x) = \frac{\sqrt{x-1}}{x-2}$ has a real number value, we need to ensure that both the numerator and the denominator satisfy their respective conditions for real numbers.

1. **Numerator Condition**: The expression inside the square root must be non-negative.

$$x - 1 \geqslant 0 \implies x \geqslant 1$$

2. **Denominator Condition**: The denominator must not be zero to avoid division by zero.

$$x - 2 \neq 0 \implies x \neq 2$$

Combining these conditions, we need $x \geqslant 1$ and $x \neq 2$. The smallest integer that satisfies these conditions is $x = 3$.
Therefore, the smallest possible integer value for $x$ such that $f(x)$ has a real number value is $\boxed{3}$.

---

To determine the smallest possible integer value for $x$ such that $f(x) = \frac{\sqrt{x-1}}{x-2}$ has a real number value, we need to ensure that both the numerator and the denominator satisfy certain conditions.

1. **Numerator Condition**: The expression inside the square root must be non-negative because the square root of a negative number is not a real number.

$$x - 1 \geqslant 0 \implies x \geqslant 1$$

2. **Denominator Condition**: The denominator must not be zero because division by zero is undefined.

$$x - 2 \neq 0 \implies x \neq 2$$

Combining these conditions, we get:

$$x \geqslant 1 \quad \text{and} \quad x \neq 2$$

The smallest integer that satisfies $x \geqslant 1$ and $x \neq 2$ is $x = 3$.

---

*(Details omitted as they are not crucial for this case study.)*

1. **Square Root Condition**: The expression inside the square root, $x - 1$, must be greater than or equal to zero for the square root to be a real number.

$$x - 1 \geqslant 0 \implies x \geqslant 1$$

2. **Denominator Condition**: The denominator, $x - 2$, must not be zero to avoid division by zero.

$$x - 2 \neq 0 \implies x \neq 2$$

Combining these two conditions, we get:

$$x \geqslant 1 \quad \text{and} \quad x \neq 2$$

The smallest integer that satisfies $x \geqslant 1$ and $x \neq 2$ is $x = 3$.
Therefore, the smallest possible integer value for $x$ such that $f(x)$ has a real number value is $\boxed{3}$.

---

*(Details omitted as they are not crucial for this case study.)*

1. For the square root to be defined, the expression inside it must be greater than or equal to zero:

$$x - 1 \geqslant 0$$

Solving this inequality:

$$x \geqslant 1$$

2. For the denominator to be non-zero:

$$x - 2 \neq 0$$

Solving this equation:

$$x \neq 2$$

*(Details omitted as they are not crucial for this case study.)*
Therefore, the smallest possible integer value for $x$ such that $f(x)$ has a real number value is:

$$\boxed{3}$$

---

*(Details omitted as they are not crucial for this case study.)*

1. For the square root to be defined, the expression inside it must be greater than or equal to zero:

$$x - 1 \geqslant 0$$

Solving this inequality:

$$x \geqslant 1$$

2. For the function to be defined, the denominator must not be zero:

$$x - 2 \neq 0$$

Solving this equation:

$$x \neq 2$$

*(Details omitted as they are not crucial for this case study.)*
Therefore, the smallest possible integer value for $x$ such that $f(x)$ has a real number value is $\boxed{3}$.

---

*(Details omitted as they are not crucial for this case study.)*

1. For the square root to be defined, we need:

$$x - 1 \geqslant 0$$

Solving this inequality:

$$x \geqslant 1$$

2. For the denominator to be non-zero, we need:

$$x - 2 \neq 0$$

Solving this equation:

$$x \neq 2$$

Combining these two conditions, $x$ must be greater than or equal to 1 but not equal to 2. The smallest integer that satisfies this condition is $x = 3$.
Therefore, the smallest possible integer value for $x$ such that $f(x)$ has a real number value is $\boxed{3}$.

---

*(Details omitted as they are not crucial for this case study.)*

1. **Square Root Condition**: The expression inside the square root, $x - 1$, must be greater than or equal to zero for the square root to be a real number.

$$x - 1 \geqslant 0 \implies x \geqslant 1$$

2. **Denominator Condition**: The denominator, $x - 2$, must not be zero to avoid division by zero.

$$x - 2 \neq 0 \implies x \neq 2$$

Combining these two conditions, we get:

$$x \geqslant 1 \quad \text{and} \quad x \neq 2$$

The smallest integer that satisfies $x \geqslant 1$ and $x \neq 2$ is $x = 3$.
Therefore, the smallest possible integer value for $x$ such that $f(x)$ has a real number value is $\boxed{3}$.

---

Figure 14: Case study demonstrating improved exploration: the auxiliary shaping term enables the model to explore and reach the correct solution (continued).

**Vanilla GRPO#9-#16**

*(Details omitted as they are not crucial for this case study.)*

1. **Numerator Condition**: The expression inside the square root, $x-1$, must be non-negative because the square root of a negative number is not a real number. Therefore, we need:

$$x - 1 \geqslant 0 \implies x \geqslant 1$$

2. **Denominator Condition**: The denominator, $x - 2$, must not be zero because division by zero is undefined. Therefore, we need:

$$x - 2 \neq 0 \implies x \neq 2$$

Combining these two conditions, we get:

$$x \geqslant 1 \quad \text{and} \quad x \neq 2$$

The smallest integer that satisfies $x \geqslant 1$ and $x \neq 2$ is $x = 3$. Therefore, the smallest possible integer value for $x$ such that $f(x)$ has a real number value is: $\boxed{3}$

---

*(Details omitted as they are not crucial for this case study.)*

1. **Numerator Condition**: The expression inside the square root, $x-1$, must be non-negative because the square root of a negative number is not a real number. Therefore, we need:

$$x - 1 \geqslant 0 \implies x \geqslant 1$$

2. **Denominator Condition**: The denominator, $x - 2$, must not be zero because division by zero is undefined. Therefore, we need:

$$x - 2 \neq 0 \implies x \neq 2$$

Combining these two conditions, we get:

$$x \geqslant 1 \quad \text{and} \quad x \neq 2$$

The smallest integer that satisfies $x \geqslant 1$ and $x \neq 2$ is $x = 3$. Therefore, the smallest possible integer value for $x$ such that $f(x)$ has a real number value is $\boxed{3}$.

---

*(Details omitted as they are not crucial for this case study.)*

1. For the square root to be defined, we need:

$$x - 1 \geqslant 0$$
$$x \geqslant 1$$

2. For the denominator to be non-zero, we need:

$$x - 2 \neq 0$$
$$x \neq 2$$

Combining these two conditions, we get:

$$x \geqslant 1$$
$$x \neq 2$$

The smallest integer that satisfies $x \geqslant 1$ and $x \neq 2$ is $x = 3$. Therefore, the smallest possible integer value for $x$ such that $f(x)$ has a real number value is $\boxed{3}$.

---

*(Details omitted as they are not crucial for this case study.)*

1. **Numerator Condition**: The expression under the square root, $x-1$, must be non-negative because the square root of a negative number is not a real number. Therefore, we need:

$$x - 1 \geqslant 0 \implies x \geqslant 1$$

2. **Denominator Condition**: The denominator, $x - 2$, must not be zero because division by zero is undefined. Therefore, we need:

$$x - 2 \neq 0 \implies x \neq 2$$

Combining these two conditions, we get:

$$x \geqslant 1 \quad \text{and} \quad x \neq 2$$

The smallest integer that satisfies $x \geqslant 1$ and $x \neq 2$ is $x = 3$. Therefore, the smallest possible integer value for $x$ such that $f(x)$ has a real number value is: $\boxed{3}$

---

*(Details omitted as they are not crucial for this case study.)*

1. **Numerator Condition**: The numerator is $\sqrt{x-1}$. For the square root to be defined and result in a real number, the expression inside the square root must be non-negative.

$$x - 1 \geqslant 0 \implies x \geqslant 1$$

2. **Denominator Condition**: The denominator is $x - 2$. For the function to be defined, the denominator must not be zero.

$$x - 2 \neq 0 \implies x \neq 2$$

Combining these two conditions, we need $x$ to be greater than or equal to 1, but not equal to 2. Therefore, the smallest possible integer value for $x$ that satisfies both conditions is 3.

$\boxed{3}$

---

*(Details omitted as they are not crucial for this case study.)*

1. **Square Root Condition**: The expression inside the square root, $x - 1$, must be greater than or equal to zero for the square root to be a real number.

$$x - 1 \geqslant 0 \implies x \geqslant 1$$

2. **Denominator Condition**: The denominator, $x - 2$, must not be zero to avoid division by zero.

$$x - 2 \neq 0 \implies x \neq 2$$

Combining these two conditions, we get:

$$x \geqslant 1 \quad \text{and} \quad x \neq 2$$

The smallest integer that satisfies $x \geqslant 1$ and $x \neq 2$ is $x = 3$. Therefore, the smallest possible integer value for $x$ such that $f(x)$ has a real number value is $\boxed{3}$.

---

To determine the smallest possible integer value for $x$ such that $f(x) = \frac{\sqrt{x-1}}{x-2}$ has a real number value, we need to ensure that both the numerator and the denominator satisfy their respective conditions for real numbers.

1. **Numerator Condition**: The expression inside the square root must be non-negative.

$$x - 1 \geqslant 0 \implies x \geqslant 1$$

2. **Denominator Condition**: The denominator must not be zero.

$$x - 2 \neq 0 \implies x \neq 2$$

Combining these conditions, we need $x$ to be greater than or equal to 1 but not equal to 2. Therefore, the smallest possible integer value for $x$ that satisfies both conditions is 3.

$\boxed{3}$

---

*(Details omitted as they are not crucial for this case study.)*

1. **Numerator Condition**: The numerator is $\sqrt{x-1}$. For the square root to be a real number, the expression inside the square root must be non-negative.

$$x - 1 \geqslant 0 \implies x \geqslant 1$$

2. **Denominator Condition**: The denominator is $x - 2$. For the function to be defined, the denominator must not be zero.

$$x - 2 \neq 0 \implies x \neq 2$$

Combining these two conditions, we get:

$$x \geqslant 1 \quad \text{and} \quad x \neq 2$$

The smallest integer that satisfies $x \geqslant 1$ and $x \neq 2$ is $x = 3$. Therefore, the smallest possible integer value for $x$ such that $f(x)$ has a real number value is $\boxed{3}$.

Figure 14: Case study demonstrating improved exploration: the auxiliary shaping term enables the model to explore and reach the correct solution (continued).

**GRPO with Auxiliary Shaping Term#1-#8**

*(Details omitted as they are not crucial for this case study.)*

1. **Numerator Condition**: The numerator is $\sqrt{x-1}$. For the square root to be defined and real, the expression inside the square root must be non-negative.

$$x - 1 \geqslant 0 \implies x \geqslant 1$$

2. **Denominator Condition**: The denominator is $x - 2$. For the function to be defined, the denominator must not be zero.

$$x - 2 \neq 0 \implies x \neq 2$$

Combining these conditions, we need $x$ to be greater than or equal to 1, but not equal to 2. The smallest integer that satisfies these conditions is $x = 3$. Therefore, the smallest possible integer value for $x$ such that $f(x)$ has a real number value is $\boxed{3}$.

---

*(Details omitted as they are not crucial for this case study.)*

1. For the square root to be defined in the real numbers, the expression inside it must be greater than or equal to zero:

$$x - 1 \geqslant 0$$

Solving this inequality:

$$x \geqslant 1$$

2. For the function to be defined, the denominator must not be zero:

$$x - 2 \neq 0$$

Solving this equation:

$$x \neq 2$$

*(Details omitted as they are not crucial for this case study.)*

Thus, the smallest possible integer value for $x$ such that $f(x)$ has a real number value is $\boxed{3}$.

---

*(Details omitted as they are not crucial for this case study.)*

1. The expression inside the square root must be non-negative. 2. The denominator must not be zero.

*(Details omitted as they are not crucial for this case study.)*

### Step 3: Combine the conditions From Step 1, we know $x \geqslant 1$. From Step 2, we know $x \neq 2$. Therefore, the smallest integer value for $x$ that satisfies both conditions is the smallest integer greater than 1, which is 3.

### Final Answer The smallest possible integer value for $x$ such that $f(x)$ has a real number value is $\boxed{3}$.

---

*(Details omitted as they are not crucial for this case study.)*

### Combining the Conditions From Condition 1, we have $x \geqslant 1$. From Condition 2, we have $x \neq 2$. Combining these, the smallest integer value for $x$ that satisfies both conditions is $x = 3$.

### Verification Let's verify that $x = 3$ works: - For $x = 3$, $\sqrt{x-1} = \sqrt{3-1} = \sqrt{2}$, which is a real number. - The denominator $x - 2 = 3 - 2 = 1$, which is not zero. Since both conditions are satisfied, the smallest possible integer value for $x$ is indeed 3.

Thus, the final answer is:

$$\boxed{3}$$

---

*(Details omitted as they are not crucial for this case study.)*

1. **Numerator Condition**: The expression under the square root, $x - 1$, must be non-negative because the square root of a negative number is not a real number. Therefore, we need:

$$x - 1 \geqslant 0 \implies x \geqslant 1$$

2. **Denominator Condition**: The denominator, $x - 2$, must not be zero because division by zero is undefined. Therefore, we need:

$$x - 2 \neq 0 \implies x \neq 2$$

Combining these two conditions, we get:

$$x \geqslant 1 \quad \text{and} \quad x \neq 2$$

The smallest integer that satisfies $x \geqslant 1$ but not $x = 2$ is $x = 1$. Therefore, the smallest possible integer value for $x$ such that $f(x)$ has a real number value is $\boxed{1}$.

---

*(Details omitted as they are not crucial for this case study.)*

### Combining the Conditions From Condition 1, we know $x \geqslant 1$. From Condition 2, we know $x \neq 2$. The smallest integer that satisfies $x \geqslant 1$ but not $x = 2$ is $x = 1$.

*(Details omitted as they are not crucial for this case study.)*

Since $f(1)$ is a real number, $x = 1$ is indeed a valid solution. Therefore, the smallest possible integer value for $x$ such that $f(x)$ has a real number value is $\boxed{1}$.

---

*(Details omitted as they are not crucial for this case study.)*

1. For $\sqrt{x-1}$ to be defined, we need:

$$x - 1 \geqslant 0 \implies x \geqslant 1$$

2. For the denominator $x - 2$ to be non-zero, we need:

$$x - 2 \neq 0 \implies x \neq 2$$

Combining these two conditions, we get:

$$x \geqslant 1 \quad \text{and} \quad x \neq 2$$

The smallest integer that satisfies $x \geqslant 1$ and $x \neq 2$ is $x = 1$.

*(Details omitted as they are not crucial for this case study.)*

Since $f(1) = 0$ is a real number, our solution is confirmed. Therefore, the smallest possible integer value for $x$ such that $f(x)$ has a real number value is $\boxed{1}$.

---

*(Details omitted as they are not crucial for this case study.)*

1. For $\sqrt{x-1}$ to be defined in the real numbers, we need:

$$x - 1 \geqslant 0 \implies x \geqslant 1$$

2. For the denominator $x - 2$ to not be zero, we need:

$$x - 2 \neq 0 \implies x \neq 2$$

Combining these two conditions, $x$ must be greater than or equal to 1 but not equal to 2. The smallest integer that satisfies this condition is $x = 3$. Therefore, the smallest possible integer value for $x$ such that $f(x)$ has a real number value is $\boxed{3}$.

Figure 14: Case study demonstrating improved exploration: the auxiliary shaping term enables the model to explore and reach the correct solution (continued).

**GRPO with Auxiliary Shaping Term#9-#16**

*(Details omitted as they are not crucial for this case study.)*
1. **Numerator Condition**: The expression inside the square root, $x - 1$, must be non-negative because the square root of a negative number is not a real number. Therefore, we need:

$$x - 1 \geqslant 0 \implies x \geqslant 1$$

2. **Denominator Condition**: The denominator, $x - 2$, must not be zero because division by zero is undefined. Therefore, we need:

$$x - 2 \neq 0 \implies x \neq 2$$

Combining these two conditions, we get:

$$x \geqslant 1 \quad \text{and} \quad x \neq 2$$

The smallest integer that satisfies $x \geqslant 1$ and $x \neq 2$ is $x = 1$.
Therefore, the smallest possible integer value for $x$ such that $f(x)$ has a real number value is $\boxed{1}$.

---

*(Details omitted as they are not crucial for this case study.)*
1. **Numerator Condition**: The expression under the square root must be non-negative.

$$x - 1 \geqslant 0 \implies x \geqslant 1$$

2. **Denominator Condition**: The denominator must not be zero.

$$x - 2 \neq 0 \implies x \neq 2$$

Combining these two conditions, we get:

$$x \geqslant 1 \quad \text{and} \quad x \neq 2$$

The smallest integer that satisfies $x \geqslant 1$ and $x \neq 2$ is $x = 3$.
Therefore, the smallest possible integer value for $x$ such that $f(x)$ has a real number value is $\boxed{3}$.

---

*(Details omitted as they are not crucial for this case study.)*
1. **Numerator Condition**: The expression inside the square root must be non-negative.

$$x - 1 \geqslant 0 \implies x \geqslant 1$$

This ensures that $\sqrt{x - 1}$ is a real number.
2. **Denominator Condition**: The denominator must not be zero.

$$x - 2 \neq 0 \implies x \neq 2$$

This ensures that the function is defined.
Combining these two conditions, we get:

$$x \geqslant 1 \quad \text{and} \quad x \neq 2$$

The smallest integer that satisfies $x \geqslant 1$ and $x \neq 2$ is $x = 3$.
Therefore, the smallest possible integer value for $x$ such that $f(x)$ has a real number value is $\boxed{3}$.

---

*(Details omitted as they are not crucial for this case study.)*
1. For the square root to be defined, we need:

$$x - 1 \geqslant 0$$

Solving this inequality:

$$x \geqslant 1$$

2. For the denominator to be non-zero, we need:

$$x - 2 \neq 0$$

Solving this equation:

$$x \neq 2$$

Combining these two conditions, $x$ must be greater than or equal to 1 but not equal to 2. The smallest integer that satisfies this condition is $x = 3$. Therefore, the smallest possible integer value for $x$ such that $f(x)$ has a real number value is $\boxed{3}$.

---

*(Details omitted as they are not crucial for this case study.)*
1. The expression inside the square root must be non-negative. 2. The denominator must not be zero.
Let's address these conditions one by one:
1. For the square root to be defined, the expression inside it must be greater than or equal to zero:

$$x - 1 \geqslant 0 \implies x \geqslant 1$$

2. For the denominator to be non-zero:

$$x - 2 \neq 0 \implies x \neq 2$$

Combining these conditions, we need $x$ to be greater than or equal to 1 but not equal to 2. The smallest integer that satisfies this is $x = 3$.
Therefore, the smallest possible integer value for $x$ such that $f(x)$ has a real number value is $\boxed{3}$.

---

*(Details omitted as they are not crucial for this case study.)*
1. **Numerator Condition**: The expression inside the square root must be non-negative because the square root of a negative number is not a real number.

$$x - 1 \geqslant 0 \implies x \geqslant 1$$

2. **Denominator Condition**: The denominator must not be zero because division by zero is undefined.

$$x - 2 \neq 0 \implies x \neq 2$$

Combining these two conditions, we get:

$$x \geqslant 1 \quad \text{and} \quad x \neq 2$$

The smallest integer that satisfies $x \geqslant 1$ and $x \neq 2$ is $x = 3$.
Therefore, the smallest possible integer value for $x$ such that $f(x)$ has a real number value is $\boxed{3}$.

---

*(Details omitted as they are not crucial for this case study.)*
1. **Numerator Condition**: The expression under the square root, $x - 1$, must be non-negative because the square root of a negative number is not a real number.

$$x - 1 \geqslant 0 \implies x \geqslant 1$$

2. **Denominator Condition**: The denominator, $x - 2$, must not be zero because division by zero is undefined.

$$x - 2 \neq 0 \implies x \neq 2$$

Combining these conditions, we need $x \geqslant 1$ and $x \neq 2$. The smallest integer that satisfies these conditions is $x = 1$.
*(Details omitted as they are not crucial for this case study.)*
Therefore, the smallest possible integer value for $x$ such that $f(x)$ has a real number value is $\boxed{1}$.

---

To determine the smallest possible integer value for $x$ such that $f(x) = \frac{\sqrt{x-1}}{x-2}$ has a real number value, we need to ensure that both the numerator and the denominator satisfy their respective conditions for real numbers.
1. **Numerator Condition**: The expression inside the square root must be non-negative.

$$x - 1 \geqslant 0 \implies x \geqslant 1$$

2. **Denominator Condition**: The denominator must not be zero.

$$x - 2 \neq 0 \implies x \neq 2$$

Combining these conditions, we need $x$ to be greater than or equal to 1 but not equal to 2. The smallest integer that satisfies this condition is $x = 3$.
Therefore, the smallest possible integer value for $x$ such that $f(x)$ has a real number value is $\boxed{3}$.

Figure 14: Case study demonstrating improved exploration: the auxiliary shaping term enables the model to explore and reach the correct solution (continued).

**Case Study** II.    As shown in Fig. 15, this high-school level geometry problem illustrates the contrast between vanilla GRPO and GRPO with the auxiliary shaping term. The vanilla GRPO model fails to produce the correct answer in any rollout. In contrast, the model trained with the auxiliary shaping term succeeds in 5 out of 16 rollouts. Moreover, its reasoning traces display higher exploratory diversity: it applies the law of sines 9 times, the law of cosines 4 times, and other general geometric properties 3 times. By comparison, the vanilla GRPO model relies more narrowly on the law of sines (11 times), with only limited use of other strategies. This case highlights that the auxiliary shaping term enhances the model's exploration capacity, enabling it to leverage a broader range of problem-solving strategies rather than over-committing to a single path.

**Question**

Abdul and Chiang are standing 48 feet apart in a field. Bharat is standing in the same field as far from Abdul as possible so that the angle formed by his lines of sight to Abdul and Chiang measures $60°$. What is the square of the distance (in feet) between Abdul and Bharat?

**Ground Truth**

3072

**Vanilla GRPO#1-#8**

*(Details omitted as they are not crucial for this case study.)*
1. **Identify the triangle and its properties:**
*(Details omitted as they are not crucial for this case study.)*
2. **Use the Law of Sines:**
*(Details omitted as they are not crucial for this case study.)*
3. **Apply the Law of Sines:**
*(Details omitted as they are not crucial for this case study.)*
4. **Find the square of the distance:**
*(Details omitted as they are not crucial for this case study.)*
Therefore, the square of the distance between Abdul and Bharat is $\boxed{2304}$.

---

To solve this problem, we can use the Law of Sines in trigonometry. Let's denote the distance between Abdul and Bharat as $x$ and the distance between Chiang and Bharat as $y$. We know that the angle between Abdul and Chiang as seen from Bharat is 60°, and the distance between Abdul and Chiang is 48 feet. Given: - $\angle ABC = 60°$ - $AB = 48$ feet - We need to find $x^2$
First, we apply the Law of Sines in triangle $ABC$:
*(Details omitted as they are not crucial for this case study.)*
Thus, the square of the distance between Abdul and Bharat is $\boxed{9216}$.

---

To solve this problem, we can use the Law of Sines in trigonometry. Let's denote the distance between Abdul and Bharat as $AB$, and the distance between Chiang and Bharat as $BC$. Since Bharat is as far from Abdul as possible to form a 60° angle, triangle $ABC$ is a right triangle with the right angle at Chiang. This means $\angle ACB = 90°$ and $\angle BAC = 60°$.
*(Details omitted as they are not crucial for this case study.)*
Thus, the square of the distance between Abdul and Bharat is $\boxed{9216}$.

---

First, we need to find the distance $AB$ using the Law of Sines. In triangle $ABC$:
*(Details omitted as they are not crucial for this case study.)*
Now, we need to find the square of the distance $AB$:
*(Details omitted as they are not crucial for this case study.)*
Thus, the square of the distance between Abdul and Bharat is $\boxed{1728}$.
To verify the accuracy of this result, we can use Python to perform the calculations.
*(Details omitted as they are not crucial for this case study.)*

---

To solve this problem, we can use the Law of Sines and some geometric properties. Let's break it down step by step.
1. **Identify the triangle and its properties:**
*(Details omitted as they are not crucial for this case study.)*
2. **Apply the Law of Sines:**
*(Details omitted as they are not crucial for this case study.)*
3. **Calculate the distances:**
*(Details omitted as they are not crucial for this case study.)*
4. **Find the square of the distance:**
*(Details omitted as they are not crucial for this case study.)*
So, the square of the distance between Abdul and Bharat is $\boxed{768}$.

---

*(Details omitted as they are not crucial for this case study.)*
1. Recognize that the triangle $ABC$ is a right triangle with $\angle BAC = 90°$ and $\angle ABC = 60°$. Therefore, $\angle ACB = 30°$.
*(Details omitted as they are not crucial for this case study.)*
3. Since $AC = 48$ feet, and in a 30-60-90 triangle, the side opposite the 60° angle is $\sqrt{3}$ times the side opposite the 30° angle, we can find $AB$:

$$AB = 48\sqrt{3}$$

*(Details omitted as they are not crucial for this case study.)*
Therefore, the square of the distance between Abdul and Bharat is $\boxed{6912}$ feet.

---

To solve this problem, we need to use some geometric principles, specifically involving the Law of Sines and properties of triangles.
1. **Identify the triangle and the given information:**
*(Details omitted as they are not crucial for this case study.)*
2. **Use the Law of Sines to find the distance BC (the distance from Bharat to Chiang):**
*(Details omitted as they are not crucial for this case study.)*
3. **Find the distance AC (the distance from Abdul to Bharat):**
*(Details omitted as they are not crucial for this case study.)*
Therefore, the square of the distance between Abdul and Bharat is $\boxed{11520}$.

---

*(Details omitted as they are not crucial for this case study.)*
1. **Identify the triangle and the given information:**
*(Details omitted as they are not crucial for this case study.)*
2. **Use the Law of Sines:**
*(Details omitted as they are not crucial for this case study.)*
3. **Maximize $AB$:**
*(Details omitted as they are not crucial for this case study.)*
4. **Apply the Pythagorean theorem:**
*(Details omitted as they are not crucial for this case study.)*
Thus, the square of the distance between Abdul and Bharat is $\boxed{9216}$.

Figure 15: Case study: Geometry problem. GRPO with auxiliary shaping term not only achieves higher accuracy but also explores more diverse reasoning strategies than vanilla GRPO.

**Vanilla GRPO#9-#16**

To solve this problem, we can use the Law of Sines and some geometric properties. Let's break it down step by step.
1. **Identify the triangle and the given information:**
*(Details omitted as they are not crucial for this case study.)*
2. **Determine the largest possible angle at A:**
*(Details omitted as they are not crucial for this case study.)*
3. **Use the Pythagorean theorem:**
*(Details omitted as they are not crucial for this case study.)*
4. **Find the square of the distance between Abdul and Bharat:**
*(Details omitted as they are not crucial for this case study.)*
So, the square of the distance between Abdul and Bharat is 9216 .

To solve this problem, we can use some geometric properties and trigonometry. Let's break it down step by step:
1. **Identify the triangle formed**:
*(Details omitted as they are not crucial for this case study.)*
2. **Use the Law of Sines**:
*(Details omitted as they are not crucial for this case study.)*
3. **Maximize the distance**:
*(Details omitted as they are not crucial for this case study.)*
4. **Square the distance**:
*(Details omitted as they are not crucial for this case study.)*
Let's calculate it:

$$48^2 = 2304$$

So, the square of the distance between Abdul and Bharat is 2304 .

To solve the problem, we need to determine the distance between Abdul and Bharat, given that the angle between their lines of sight to Chiang is 60° and that Bharat is positioned to be as far from Abdul as possible.
*(Details omitted as they are not crucial for this case study.)*
Using the Law of Cosines in triangle $ABC$:
*(Details omitted as they are not crucial for this case study.)*

$$BC^2 = d^2 + 2304 - 48d$$

*(Details omitted as they are not crucial for this case study.)*
The square of the distance between Abdul and Bharat is:

$$d^2 = 96^2 = 9216$$

Thus, the square of the distance between Abdul and Bharat is 9216 .

*(Details omitted as they are not crucial for this case study.)*
1. **Identify the known values:**
*(Details omitted as they are not crucial for this case study.)*
2. **Set up the triangle:**
*(Details omitted as they are not crucial for this case study.)*
3. **Apply the Law of Sines:**
*(Details omitted as they are not crucial for this case study.)*
4. **Determine the angles:**
*(Details omitted as they are not crucial for this case study.)*
5. **Simplify the Law of Sines equation:**
*(Details omitted as they are not crucial for this case study.)*
Therefore, the square of the distance between Abdul and Bharat is 2304 .

To solve this problem, we need to use some geometric properties and trigonometry. Let's break it down step by step.
1. **Understanding the Problem:**
*(Details omitted as they are not crucial for this case study.)*
2. **Geometric Insight:**
*(Details omitted as they are not crucial for this case study.)*
3. **Using Trigonometry:**
*(Details omitted as they are not crucial for this case study.)*
4. **Calculating the Distance:** - Let the distance AB be x. Since ∠ABC is 60° and we need to maximize AB, we use the Law of Cosines in triangle ABC:
*(Details omitted as they are not crucial for this case study.)*
Therefore, the square of the distance between Abdul and Bharat is 432 .

*(Details omitted as they are not crucial for this case study.)*
1. **Identify the given information:**
*(Details omitted as they are not crucial for this case study.)*
2. **Visualize the problem:**
*(Details omitted as they are not crucial for this case study.)*
3. **Use the Law of Sines:**
*(Details omitted as they are not crucial for this case study.)*
4. **Properties of an equilateral triangle:**
*(Details omitted as they are not crucial for this case study.)*
5. **Calculate the square of the distance:**
*(Details omitted as they are not crucial for this case study.)*
Thus, the square of the distance between Abdul and Bharat is 2304 .
Providethefinalanswerwithin□.

To solve this problem, we need to use some geometric properties and trigonometry. Let's break it down step by step.
1. **Understand the Problem:**
*(Details omitted as they are not crucial for this case study.)*
2. **Visualize the Problem:**
*(Details omitted as they are not crucial for this case study.)*
3. **Use Geometric Properties:**
*(Details omitted as they are not crucial for this case study.)*
4. **Apply Trigonometry:**
*(Details omitted as they are not crucial for this case study.)*
5. **Calculate the Distance:** - AB = 2 * AC = 2 * 48 = 96 feet. - The square of the distance AB is $96^2$.
6. **Final Calculation:** - $96^2 = 9216$.
So, the square of the distance between Abdul and Bharat is 9216 .

To solve this problem, we can use the Law of Sines in trigonometry. Let's break it down step by step:
1. **Identify the given information:**
*(Details omitted as they are not crucial for this case study.)*
2. **Set up the triangle:**
*(Details omitted as they are not crucial for this case study.)*
3. **Use the Law of Sines:**
*(Details omitted as they are not crucial for this case study.)*
4. **Find the square of the distance:** The distance between Abdul and Bharat is 24 feet. The square of this distance is:

$$24^2 = 576$$

Therefore, the square of the distance between Abdul and Bharat is 576 .

Figure 15: Case study: Geometry problem. GRPO with auxiliary shaping term not only achieves higher accuracy but also explores more diverse reasoning strategies than vanilla GRPO (continued).

**GRPO with Auxiliary Shaping Term#1-#8**

*(Details omitted as they are not crucial for this case study.)*
Given that $\angle ABC = 60°$ and $AC = 48$ feet, we can use the Law of Sines to find $AB$:
*(Details omitted as they are not crucial for this case study.)*
The square of the distance $AB$ is:

$$(32\sqrt{3})^2 = 32^2 \times 3 = 1024 \times$$

Thus, the square of the distance between Abdul and Bharat is $\boxed{3072}$.

---

To solve this problem, we can use the Law of Sines in trigonometry.
*(Details omitted as they are not crucial for this case study.)*
Given that $AC = 48$ feet, we can use the Law of Sines to find $AB$:

$$\frac{AB}{\sin(60°)} = \frac{AC}{\sin(30°)}$$

*(Details omitted as they are not crucial for this case study.)*
Solving for $AB$:

$$AB = 96 \times \frac{\sqrt{3}}{2} = 48\sqrt{3}$$

*(Details omitted as they are not crucial for this case study.)*
Therefore, the square of the distance between Abdul and Bharat is $\boxed{6912}$.

---

*(Details omitted as they are not crucial for this case study.)*
1. **Identify the given information:**
*(Details omitted as they are not crucial for this case study.)*
2. **Visualize the problem:**
*(Details omitted as they are not crucial for this case study.)*
3. **Determine the type of triangle:** - Since angle ABC and angle ACB are both 60°, triangle ABC is an equilateral triangle. In an equilateral triangle, all sides are equal.
4. **Calculate the side length of the equilateral triangle:** - Since AC = 48 feet and triangle ABC is equilateral, AB (the distance from Abdul to Bharat) is also 48 feet.
*(Details omitted as they are not crucial for this case study.)*
So, the square of the distance between Abdul and Bharat is $\boxed{2304}$.

---

To solve this problem, we can use the Law of Sines and some geometric properties. Let's denote the positions of Abdul, Chiang, and Bharat as points $A$, $C$, and $B$ respectively. The distance between $A$ and $C$ is given as 48 feet. The angle $\angle ACB$ is 60°. We need to find the square of the distance $AB$, which is the farthest distance Bharat can be from Abdul.
*(Details omitted as they are not crucial for this case study.)*
Therefore, the square of the distance between Abdul and Bharat is $\boxed{6912}$.

---

To solve this problem, we need to use some geometric principles. Let's denote the positions of Abdul, Chiang, and Bharat as points A, C, and B respectively. We are given that the distance between A and C (AC) is 48 feet, and the angle ABC is 60°. We need to find the distance between A and B (AB), and then square it.
*(Details omitted as they are not crucial for this case study.)*
Step 3: Recognize that if triangle ABC is equilateral, then all sides are equal. Therefore, AB = BC = AC = 48 feet.
Step 4: Calculate the square of the distance AB. Since AB = 48 feet, we square this value to get AB².

$$AB^2 = 48^2 = 2304$$

So, the square of the distance between Abdul and Bharat is $\boxed{2304}$.

---

To find the square of the distance between Abdul and Bharat, we can use the Law of Sines in trigonometry. Let's denote the distance between Abdul and Bharat as $AB$, and the distance between Chiang and Bharat as $BC$. Since Bharat is as far from Abdul as possible, triangle $ABC$ is a right triangle with $\angle ACB = 90°$.
Given: - $\angle ABC = 60°$ - $AC = 48$ feet
We want to find $AB^2$.
First, we apply the Law of Sines in triangle $ABC$:
*(Details omitted as they are not crucial for this case study.)*
Thus, the square of the distance between Abdul and Bharat is $\boxed{3072}$.

---

*(Details omitted as they are not crucial for this case study.)*
In a right triangle, if one of the angles is 60°, the triangle is a 30-60-90 triangle. In a 30-60-90 triangle, the sides are in the ratio $1 : \sqrt{3} : 2$. Here, $AC$ is the hypotenuse, so we can set up the relationship as follows:

$$AC = 2 \times AB$$
$$48 = 2 \times AB$$
$$AB = \frac{48}{2} = 24 \text{ feet}$$

We need to find the square of the distance $AB$:

$$AB^2 = 24^2 = 576$$

Thus, the square of the distance between Abdul and Bharat is $\boxed{576}$.

---

*(Details omitted as they are not crucial for this case study.)*
Using the properties of a 30-60-90 triangle, we know that the side opposite the 60° angle is $\sqrt{3}$ times the side opposite the 30° angle. Let's denote the side opposite the 30° angle (which is half the distance between Abdul and Chiang) as $x$. Therefore, $AB = x\sqrt{3}$ and $BC = 2x$.
*(Details omitted as they are not crucial for this case study.)*

$$AB^2 = 2304(21 - 12\sqrt{3})$$

Since we are asked to find the square of the distance, and we have already calculated it, we can box the final answer:

$$\boxed{2304}$$

Figure 15: Case study: Geometry problem. GRPO with auxiliary shaping term not only achieves higher accuracy but also explores more diverse reasoning strategies than vanilla GRPO (continued).

**GRPO with Auxiliary Shaping Term#9-#16**

To solve this problem, we can use the Law of Sines in trigonometry.
*(Details omitted as they are not crucial for this case study.)*
Using the Law of Sines:

$$\frac{AB}{\sin(120°)} = \frac{AC}{\sin(60°)}$$

*(Details omitted as they are not crucial for this case study.)*
Therefore, the square of the distance between Abdul and Bharat is $\boxed{3072}$.

---

To solve this problem, we need to use some geometric properties and trigonometric relationships. Let's break it down step by step.
1. **Identify the given information:**
*(Details omitted as they are not crucial for this case study.)*
2. **Visualize the problem:**
*(Details omitted as they are not crucial for this case study.)*
3. **Use the Law of Cosines:**
*(Details omitted as they are not crucial for this case study.)*
4. **Apply the Law of Cosines:**
*(Details omitted as they are not crucial for this case study.)*
5. **Find the square of the distance:** - The square of the distance between Abdul and Bharat is $d^2 = 2304$.
Therefore, the square of the distance between Abdul and Bharat is $\boxed{2304}$.

---

To solve this problem, we can use the Law of Sines and some geometric properties.
*(Details omitted as they are not crucial for this case study.)*
Given that the angle ABC is 60° and AB is maximized, the triangle ABC is a right triangle with angle ACB being 90° (since the angle at Bharat is 60° and the sum of angles in a triangle is 180°, the remaining angle at A must be 30°, making AB the hypotenuse of a 30-60-90 triangle).
*(Details omitted as they are not crucial for this case study.)*
Therefore, the square of the distance between Abdul and Bharat is $\boxed{3072}$.

---

To solve this problem, we need to use some geometric principles. Let's denote the positions of Abdul, Chiang, and Bharat as points $A$, $C$, and $B$ respectively. The distance between $A$ and $C$ is 48 feet, and we need to find the distance $AB$ when the angle $\angle ABC$ is 60° and $B$ is as far from $A$ as possible.
1. **Identify the configuration**:
*(Details omitted as they are not crucial for this case study.)*
2. **Use the Law of Cosines**:
*(Details omitted as they are not crucial for this case study.)*
3. **Substitute into the Law of Cosines**:
*(Details omitted as they are not crucial for this case study.)*
So, the square of the distance between Abdul and Bharat is $\boxed{3840}$ square feet.

---

To solve this problem, we need to use some geometric principles, specifically involving the Law of Sines and properties of triangles. Let's break it down step by step:
1. **Identify the given information:**
*(Details omitted as they are not crucial for this case study.)*
2. **Visualize the problem:**
*(Details omitted as they are not crucial for this case study.)*
3. **Use the Law of Sines:**
*(Details omitted as they are not crucial for this case study.)*
4. **Determine the angles:**
*(Details omitted as they are not crucial for this case study.)*
5. **Calculate the distance AB:**
*(Details omitted as they are not crucial for this case study.)*
Therefore, the square of the distance between Abdul and Bharat is $\boxed{3072}$.

---

To find the square of the distance between Abdul and Bharat, let's denote the distance between Abdul and Bharat as $d$, and the distance between Chiang and Bharat as $d'$. Since the angle formed by the lines of sight to Abdul and Chiang is 60°, we can use the Law of Cosines in the triangle formed by Abdul, Chiang, and Bharat. The Law of Cosines states that for any triangle with sides $a$, $b$, and $c$, and the angle $\theta$ opposite side $c$:

$$c^2 = a^2 + b^2 - 2ab\cos(\theta)$$

*(Details omitted as they are not crucial for this case study.)*
Therefore, the square of the distance between Abdul and Bharat is $\boxed{2304}$.

---

To solve this problem, we'll use some geometric principles, particularly the Law of Sines and properties of triangles. Let's break down the problem step-by-step:
1. **Identify the given information:**
*(Details omitted as they are not crucial for this case study.)*
2. **Visualize the problem:**
*(Details omitted as they are not crucial for this case study.)*
3. **Use the Law of Sines:**
*(Details omitted as they are not crucial for this case study.)*
4. **Apply the Law of Sines:**
*(Details omitted as they are not crucial for this case study.)*
5. **Use trigonometry in the right triangle:**
*(Details omitted as they are not crucial for this case study.)*
Therefore, the square of the distance between Abdul and Bharat is $\boxed{1728}$.

---

To solve this problem, we need to use some geometric properties and trigonometry. Let's denote the positions of Abdul, Chiang, and Bharat as points A, C, and B respectively in the field. We know that AC = 48 feet and the angle $\angle$ACB = 60°. We want to find the distance AB, which we'll denote as $d$, and then find $d^2$.
1. **Identify the type of triangle:**
*(Details omitted as they are not crucial for this case study.)*
2. **Apply the Law of Cosines:**
*(Details omitted as they are not crucial for this case study.)*
3. **Maximize the distance $d$:**
*(Details omitted as they are not crucial for this case study.)*
Therefore, the square of the distance between Abdul and Bharat is $\boxed{1728}$.

Figure 15: Case study: Geometry problem. GRPO with auxiliary shaping term not only achieves higher accuracy but also explores more diverse reasoning strategies than vanilla GRPO (continued).

