# OpenReview forum: "Beyond the Exploration-Exploitation Trade-off: A Hidden State Approach for LLM Reasoning in RLVR"
_ICLR.cc/2026/Conference — ICLR 2026 Conference Withdrawn Submission_

### Official Review · Reviewer_5w9e · 2025-10-20

**Soundness:** 1
**Presentation:** 3
**Contribution:** 2
**Rating:** 2
**Confidence:** 4

**Summary:**

This paper mentions that the trade-off of exploration and exploitation is not an intrinsic issue, it may exist in the context of entropy/confidence situation, but may not exist in other spaces. In particular, the authors argue that in the hidden-state space, this trade off does not exist. The authors use effective rank (ER) to measure exploration and effective rank velocity (ERV) to meaure exploitation, and provide empirical evidence that these two are not negatively correlated. This motivates the authors to shape the advantage function to encourage both high ER and high ERV, i.e., encourage both exploration and exploitation. In essence, the core claim in this paper is that the exploration and exploitation are not a trade off but can be encouraged simultaneously.

**Strengths:**

The idea of studying ER and ERV in the hidden state space is interesting, and encouraging both ER and ERV leads to a better performance. The paper provides theoretical justification of the proposed approach.

**Weaknesses:**

The major issue of this paper is the misuse of "exploitation". In standard RL [1], exploitation means greedy action, i.e., always choose the action that has the highest (current) Q-value where the Q-value is approximated either by Monte-carlo sampling or a critic model.  Exploration means that you choose some other actions that may not have the highest Q-value. In RL, the trade-off between exploration and exploitation always exists because you either choose the greedy action or non-greedy action.

In this paper, the authors find that there is no trade-off between ER and ERV. However, this does not mean that there is no exploration and exploitation trade off. Specifically, ERV measures the change in ER between successive hidden states. I do not see any relation between ERV and greedy action. That says, I highly doubt that ERV can represent exploitation.

In other words, I think this paper is not studying the exploration-exploitation trade off, but studying something else, i.e., ER-ERV relation. I do not see the reason of mentioning exploration-exploitation as a foundation in this paper as you are studying something else.

Another weakness is that the code is not provided. In the anonymous link, there is only a readme saying "hahahah". I do not see the intention of the authors to include such a "code".



[1] Reinforcement learning: An introduction

**Questions:**

Can the authors elaborate the relation between ERV and greedy action, or explain why ERV can be "exploitation"?

---

> ### Author Response · Authors · 2025-12-03
>
> Thank you for the review and for the constructive feedback! To address your concerns, we have: 1) Clarified our claim regarding the relation between ERV and greedy action; 2) Completely open-sourced the code in the anonymous link; 3) Answered each of your questions; and 4)  Revised paper with your comments.
>
> > In standard RL, exploitation means greedy action … In this paper, ERV measures the change in ER between successive hidden states. I do not see any relation between ERV and greedy action … I think this paper is not studying the exploration–exploitation trade-off, but studying something else (ER–ERV relation). I do not see the reason of mentioning exploration–exploitation as a foundation in this paper.
> >
>
> **Response (summary).**
>
> Thank you! Your comment helped us realize that this semantic focus was not stated explicitly enough. Actually, we agree that in classical RL, exploitation is defined via greedy choices w.r.t. a value function, and we do **not** claim that ERV is a direct estimator of greedy $Q$-values. Our work instead proposes **semantic, response-level proxies** for exploration and exploitation in LLM reasoning. In the revision, we have introduced clear, scoped definitions of **“Semantic Exploration” (ER)** and **“Semantic Exploitation” (ERV)** and adopted this terminology throughout, while softening the “beyond the exploration–exploitation trade-off” phrasing to avoid confusion with the classical action-level dilemma. Empirically and theoretically, these semantic proxies are decoupled and can be improved jointly by VERL, while significantly boosting RLVR performance, which we believe remains a meaningful and non-trivial contribution.
>
> ---
>
> **Details.**
>
> 1. From **action space** to **semantic trajectory space**: what level are we operating on?
>
>     We fully agree that in classical RL, exploitation means greedy choices w.r.t. an estimated value function, and exploration means deviating from that greedy policy. This is an **action-space** notion defined over $Q(s,a)$. However, in recent RLVR algorithms used for LLM post-training (e.g., PPO [1], GRPO [2]), the value function for each action is rarely an explicit tabular or network $Q(s,a)$: PPO relies on GAE as an approximation $r_t+\gamma V(s_{t+1})$ to $Q(s,a)$, and GRPO uses groupwise terminal returns as a trajectory-level estimate.
>
>     In this regime, prior RLHF/RLVR work [3–5] typically **implements** “exploitation” and “exploration” using *proxy quantities* such as token-level confidence or entropy, even though the ultimate objective is trajectory-level performance. Our work does *not* challenge the classical definition itself, but argues that these **token-level (action-space) proxies** are misaligned with how LLM reasoning actually unfolds: a single token (or even a word) rarely corresponds to a full semantic decision that “maximizes $Q(s,a)$”; instead, reasoning proceeds through multi-token **semantic chunks** (concepts, subgoals, partial derivations) [6,7].
>
>     For this reason, we move from the **action space** to the **semantic representation space of hidden states**, and from a single-step view to a **response-level** view. ER and ERV are defined at this semantic trajectory level, which is what we refer to by “Semantic Exploration” and “Semantic Exploitation” in the revision.
>
> 2. Clarification of  **Exploitation and  Exploration in the LLM reasoning**
>     - **exploration**: measures how diverse a full response searching for reasoning paths.
>     - **exploitation**: measures how the same trajectory refines the most promising known strategies along its path

---

> > ### Author Response · Authors · 2025-12-03
> >
> > 3. **Why ER and ERV can capture the Exploitation and  Exploration in the semantic space.**
> >
> >     We operate on the hidden-state sequence of the final transformer layer, which is widely understood to encode semantic content and model predictions [8–12].
> >
> >     - **ER (exploration proxy)**. Effective rank captures how widely a trajectory occupies different orthogonal semantic directions and how many **distinct semantic regions** it meaningfully visits, so we treat it as a representation-level measure of exploration.
> >     - **ERV (exploitation proxy).** ERV summarizes how ER changes along a single trajectory. With the running-history smoothing (**Def. 3.3**), a sustained and moderately positive ERV indicates that the model is steadily increasing semantic complexity within a locally stable region of the semantic space, which matches our view of exploitation as the average rate of **information gain** in that region.
> > 4. **How does this relate to the trade-off claim.**
> >
> >     Our claim "Beyond the Trade-off" refers specifically to the **semantic representation level**, not the action selection level.
> >
> >     - In **Action Space (Token level):** The trade-off remains (you cannot sample greedily and randomly at the same time).
> >     - In **Semantic Space (Representation level):** We find that RLVR allows models to achieve **both** higher semantic diversity (High ER) and rapid information gain (High ERV). This suggests that a capable reasoner can maintain a broad awareness of possibilities (Exploration) while simultaneously executing a focused, efficient deductive chain (Exploitation). We have revised the manuscript to ensure this distinction is precise and avoids overclaiming.
> >
> > [1] (PPO) Proximal Policy Optimization Algorithms. 2017 arxiv
> >
> > [2] (GRPO) DeepSeekMath: Pushing the Limits of Mathematical Reasoning in Open Language Models. 2024 arxiv
> >
> > [3] Token Hidden Reward: Steering Exploration-Exploitation in Group Relative Deep Reinforcement Learning. ICML, 2025
> >
> > [4] Optimizing Language Models for Inference Time Objectives using Reinforcement Learning. ICML, 2025
> >
> > [5] Reasoning with Exploration: An Entropy Perspective. AAAI, 2026
> >
> > [6] Chain-of-Thought Prompting Elicits Reasoning in Large Language Models. NeurIPS 2022
> >
> > [7] Tree of Thoughts: Deliberate Problem Solving with Large Language Models. ICML 2024
> >
> > [8] The geometry of hidden representations of large transformers. NeurIPS 2023
> >
> > [9] Semantics or spelling? Probing contextual word representations. ACL 2024
> >
> > [10] LinguaLens: Towards Interpreting Linguistic Mechanisms of Large Language Models via Sparse Auto-Encoder. EMNLP 2025
> >
> > [11] Are the Hidden States Hiding Something? Testing the Limits of Factuality Representations in LLMs. ACL 2025
> >
> > [12] Reasoning Models Know When They’re Right: Probing Hidden States for Self-Verification. COLM 2025

---

> ### Author Response · Authors · 2025-12-03
>
> > Can the authors elaborate the relation between ERV and greedy action, or explain why ERV can be "exploitation"?
> >
>
> Please refer to the answer to the previous weakness
>
> > Another weakness is that the code is not provided.
> >
>
> We are sorry about this. We have carefully verified that the complete codebase has been uploaded to the anonymous repository, and you can now check it there.

---

### Official Review · Reviewer_JpJc · 2025-10-28

**Soundness:** 3
**Presentation:** 3
**Contribution:** 3
**Rating:** 6
**Confidence:** 4

**Summary:**

This paper introduces VERL, a novel method for RLVR in LLM reasoning tasks. Unlike traditional RL, which assumes an inherent token-level exploration–exploitation trade-off, VERL shows that this trade-off may be an artifact of token-level metrics. The method quantifies exploration via ER of hidden states and exploitation via its first- and second-order temporal changes, i.e., ERV and ERA, which are nearly uncorrelated, allowing exploration and exploitation to be decoupled. VERL incorporates these metrics into a dynamic, dual-channel advantage function, with ERA modulating the balance to prevent overconfidence and improve reasoning efficiency. Experiments on diverse benchmarks, i.e., ASDiv and OlympiadBench, demonstrate consistent improvements, including up to 21.4% absolute accuracy gain on Gaokao 2024, and show strong generalization across RL algorithms and models. Both exploration and exploitation benefits are confirmed via Pass@k and Pass@1 metrics, and ablation studies indicate robustness to key hyperparameters.

**Strengths:**

1. Novelty: this is the first work to analyze exploration–exploitation dynamics in the hidden-state space. It introduces the ER, ERV, and ERA metrics, overcoming the limitations of token-level trade-offs.


2. Strong theoretical and empirical support: the paper provides theoretical analysis of ER, ERV, and ERA scalability, and demonstrates consistent performance improvements across multiple models, tasks, and RL algorithms.


3. General and robust: VERL is a plug-and-play method compatible with various RL frameworks, maintaining stable performance across key hyperparameters.


4. Extensive evaluation: the method has been thoroughly evaluated on diverse reasoning benchmarks and multi-step reasoning tasks, including detailed ablation studies and case analyses.

**Weaknesses:**

1. Concern about computational cost: ER, ERV, and ERA rely on SVD of hidden states, with a computational complexity of $O(TD^2)$. Since these metrics must be computed frequently during training, VERL could become computationally expensive for long sequences or large models. Although the appendix discusses efficiency considerations, no practical runtime results are reported, suggesting that computational cost may still pose a bottleneck for real-world, efficient implementation.

2. Theoretical generalization: the exploration–exploitation decoupling is primarily observed empirically and lacks a theoretical explanation. As a result, its applicability to other models, tasks, or datasets remains unverified?

**Questions:**

Q1. While VERL introduces novel hidden-state metrics to enhance exploration, the paper does not compare its performance against existing exploration strategies, such as entropy regularization methods. As a result, the relative advantage of VERL remains unclear.

---

> ### Author Response · Authors · 2025-12-03
>
> Thank you for the review and for the constructive feedback! To address your concerns, we have: 1) Clarified our claim regarding concern about computational cost; 2) Added the Theoretical generalization of the exploration–exploitation decoupling; 3) Answered each of your questions; and 4)  Revised paper with your comments.
>
> > Concern about computational cost
> >
>
> **Response (summary).**
>
> VERL is implemented without full SVD: we work on Gram matrices with incremental updates, which reduces the worst-case complexity from  $\mathcal{O}(T^2)$ to $\mathcal{O}(T)$ in sequence length. In practice, on 4×RTX 4090 with Llama-3.2-3B-Instruct and a long max response length of 4096 tokens, VERL improves evaluation metrics by **34.21%** while adding only **3.13%** wall-clock overhead and **2.31%** extra GPU memory. We have made these efficiency details explicit in **Appendix. G.2 & G.3**  in the revised version.
>
> ---
>
> **Details.**
>
> 1. **No full SVD in practice; incremental Gram updates.**
>
>     VERL does *not* compute a full SVD on hidden states at each step. By definition, effective rank depends only on singular values, so in our implementation we:
>
>     - First compute the **Gram matrix $G$** of the hidden states $\mathbf{Z}$, which is fast with GPUs
>     - Then obtain singular values from the eigenvalues of $G$ , which is mathematically equivalent but substantially cheaper than a full SVD on $\mathbf{Z}$.
>
>     For ERV/ERA, we additionally exploit the **incremental structure** of the Gram matrix across strides, reusing previous $G$ and updating it instead of recomputing from scratch. The implementation details are described in **Appendix F.2**.
>
> 2. **Theoretical complexity comparison.**
>
>     As we show in **Appendix. G2**, in the worst-case situation where the sequence length $T$ exceeds the hidden dimension $D$, and both $D$ and the stride $s$ can be treated as constants:
>
>     - The naïve approach that reconstructs matrices independently at each stride has a matrix-construction cost scaling as $\mathcal{O}(T^2)$,
>     - VERL’s incremental Gram/covariance updates scale as $\mathcal{O}(T)$.
>
>     So asymptotically, our implementation has a **strictly better dependency on $T$** than a naïve SVD-based design.
>
> 3. **Practical cost under multi-GPU training.**
>
>     Under modern multi-GPU training, the extra overhead is small:
>
>     - Constructing the Gram matrix is a single **GEMM** call, heavily optimized in BLAS/cuBLAS and typically cheaper than the bidiagonalization stage of a full SVD.
>     - Working with the symmetric positive-definite Gram matrix lets us use fast Hermitian eigenvalue routines (`torch.linalg.eigvalsh`) instead of `torch.linalg.svdvals`.
>
>     On **4×RTX 4090 (48GB)** with **Llama-3.2-3B-Instruct** and **max response length = 4096** (longer than the hidden dimension 3072), we benchmarked GRPO vs. GRPO+VERL:
>
>     | Method | Train score avg | Train gain (%) | Val score avg | Val gain (%) | sat\_math@1 | math@1 | gaokao24\_II@1 | amc24@1 | Eval gain mean (%) | Time (s) | Time overhead (%) | GPU mem (GB) | Mem overhead (%) |
>     | --- | --- | --- | --- | --- | --- | --- | --- | --- | --- | --- | --- | --- | --- |
>     | GRPO | 0.4196 | – | 0.2127 | – | 34.4 | 51.2 | 21.4 | 11.1 | – | 321.04 | – | 190.1 | – |
>     | GRPO + VERL | 0.4333 | 3.26 | 0.2214 | 4.13 | 43.8 | 52.3 | 35.7 | 15.6 | **34.21** | 331.08 | **3.13** | 194.5 | **2.31** |
>
>     VERL thus delivers **substantial evaluation gains** at **very modest** extra time and memory cost.

---

> ### Author Response · Authors · 2025-12-03
>
> > Theoretical generalization of the exploration–exploitation decoupling
> >
>
> Thank you for your comment. We have provided the mathematical proof of decoupling of ERV and ER in the reversed version in **Sec. F.3**.
>
> **Proposition (Hidden-state metrics are structurally decoupled).**
>
> Fix $K \ge 3$ evaluation steps. Consider the map that associates to each Effective Rank trajectory $m = (m_1,\dots,m_K) \in \mathbb{R}^K$ its final value
>
>
> $$
>     \mathrm{ER}_{\text{final}}(m) := m_K
> $$
>
> and its Effective Rank velocity
>
> $$
>     \mathrm{ERV}(m)    :=    \frac{1}{K-1}\sum_{j=2}^{K}        \Big(            m_j - \frac{1}{j-1}\sum_{k=1}^{j-1} m_k        \Big).
> $$
>
> Then:
> 1. There is no function $f : \mathbb{R} \to \mathbb{R}$ such that $\mathrm{ERV}(m) = f(\mathrm{ER}_{\text{final}}(m))$ for all trajectories $m \in \mathbb{R}^K$.
> 2. There is no function $g : \mathbb{R} \to \mathbb{R}$ such that $\mathrm{ER}_{\text{final}}(m) = g(\mathrm{ERV}(m))$ for all trajectories $m \in \mathbb{R}^K$.
>
> Equivalently, $\mathrm{ER}_{\text{final}}$ and $\mathrm{ERV}$ are not functionally dependent: they capture genuinely different aspects of the Effective Rank sequence.

---

> ### Author Response · Authors · 2025-12-03
>
> > Q1. While VERL introduces novel hidden-state metrics to enhance exploration, the paper does not compare its performance against existing exploration strategies, such as entropy regularization methods. As a result, the relative advantage of VERL remains unclear.
> >
>
> **Response (summary).**
>
> We agree that a direct comparison is important. In the rebuttal period, we added experiments on Llama-3.2-3B-Instruct comparing GRPO+VERL with existing token-level regularization baselines (“old exploit regu”, “old explore regu”, and their combination). VERL achieves the best overall results, improving **average pass@1** from `42.13` (best prior regularizer) to `43.44`, and **average pass@k** from `62.36` to `64.40`, while also outperforming all baselines on several challenging math benchmarks.
>
> ---
>
> **Details.**
>
> 1. **Setup.**
>
>     We keep the backbone (Llama-3.2-3B-Instruct) and GRPO training pipeline fixed, and compare:
>
>     - **GRPO** (no extra regularization),
>     - **GRPO + old exploit regu** (existing exploitation-type token-level regularizer),
>     - **GRPO + old explore regu** (existing exploration-type token-level regularizer),
>     - **GRPO + old exploit & explore regu** (both combined),
>     - **GRPO + VERL (ours)** (representation-level ER/ERV/ERA module).
> 2. **Main comparison: pass@1.**
>
>
>     | Method | aime24@1 | aime25@1 | amc23@1 | amc24@1 | asdiv@1 | carp_en@1 | cmath@1 | gaokao24\_I@1 | gaokao24\_mix@1 | gaokao\_math\_cloze@1 | gsm8k@1 | mawps@1 | olympiadbench@1 | svamp@1 | tabmwp@1 | Average |
>     | --- | --- | --- | --- | --- | --- | --- | --- | --- | --- | --- | --- | --- | --- | --- | --- | --- |
>     | Llama-3.2-3B-Instruct | 0.0 | 0.0 | 25.0 | 11.1 | 74.6 | 26.5 | 10.2 | 14.3 | 14.3 | 6.8 | 66.6 | 86.9 | 12.7 | 74.1 | 41.4 | 30.97 |
>     | GRPO | 3.3 | 0.0 | 27.5 | 8.9 | 88.8 | 45.0 | 28.3 | 21.4 | 20.9 | 23.7 | 80.7 | 96.0 | 16.7 | 87.7 | 71.7 | 41.37 |
>     | GRPO + old exploit regu | 3.3 | 0.0 | 27.5 | 15.6 | 88.8 | 44.6 | 38.2 | 7.1 | 15.4 | 22.0 | 80.9 | 96.1 | 16.3 | 87.4 | 71.9 | 41.01 |
>     | GRPO + old explore regu | 10.0 | 0.0 | 35.0 | 8.9 | 89.2 | 45.3 | 37.5 | 7.1 | 16.5 | 28.0 | 81.9 | 96.1 | 17.9 | 87.0 | 71.6 | 42.13 |
>     | GRPO + old exploit & explore | 3.3 | 3.3 | 25.0 | 13.3 | 88.5 | 44.7 | 37.7 | 0.0 | 18.7 | 23.7 | 82.0 | 96.1 | 15.6 | 87.0 | 70.5 | 40.63 |
>     | **GRPO + VERL (ours)** | 13.3 | 6.7 | 25.0 | 11.1 | 89.3 | 45.4 | 46.2 | 14.3 | 22.0 | 22.9 | 81.7 | 96.0 | 17.6 | 87.8 | 72.3 | **43.4** |
> 3. **Main comparison: pass@k.**
>
>
>     | Method | math500@16 | amc23@128 | amc24@128 | aime24@256 | aime25@256 | Average |
>     | --- | --- | --- | --- | --- | --- | --- |
>     | Llama-3.2-3B-Instruct | 79.8 | 93.5 | 51.5 | 40.0 | 30.0 | 58.96 |
>     | GRPO | 80.2 | 95.4 | 60.6 | 40.0 | 30.0 | 61.24 |
>     | GRPO + old exploit regu | 79.6 | 93.8 | 56.9 | 40.0 | 30.0 | 60.06 |
>     | GRPO + old explore regu | 78.4 | 93.1 | 56.9 | 46.7 | 36.7 | 62.36 |
>     | GRPO + old exploit & explore | 80.0 | 93.9 | 55.8 | 40.0 | 36.7 | 61.28 |
>     | **GRPO + VERL (ours)** | 80.6 | 95.7 | 59.0 | 50.0 | 36.7 | **64.40** |

---

### Official Review · Reviewer_bTfy · 2025-10-30

**Soundness:** 1
**Presentation:** 2
**Contribution:** 2
**Rating:** 2
**Confidence:** 3

**Summary:**

This paper studies the exploration–exploitation tradeoff in RLVR, focusing on representations (i.e., the *hidden state space*) of LLMs. The authors propose using Effective Rank (ER)—the entropy of the singular value spectrum of hidden representations—as an exploration metric, and the velocity of effective rank (ERV), its first-order difference, as an exploitation metric. They empirically observe that, surprisingly, both ER and ERV can improve simultaneously for certain base models under RLVR fine-tuning. Based on these findings, they introduce VERL, a new RLVR approach that augments the advantage function with ER- and ERV-based auxiliary terms. Experiments on math reasoning tasks show consistent gains for Llama-3.2-3B and Qwen2.5-7B models when VERL is combined with GRPO and PPO.

**Strengths:**

- The examination of exploration in the hidden state space rather than in the action space is a fresh perspective and adds conceptual depth to RLVR research.
- The use of effective rank, borrowed from signal processing, offers an interpretable way to quantify diversity in model representations.
- The empirical results on math domains demonstrate that VERL can yield nontrivial performance gains, particularly on challenging out-of-domain tasks.

**Weaknesses:**

1. **Questionable definition of the exploitation metric (ERV):** (top concern)

The interpretation of ERV as “exploitation” lacks grounding in classic RL. Conventionally, exploitation involves selecting actions that maximize estimated value (i.e., $\arg\max_a \hat{Q}(s,a)$), while exploration refers to deviating from that policy. Without an explicit value function, it is unclear how ERV captures exploitation behavior. Moreover, defining ERV as deviation from a historical average (rather than consecutive-step difference) causes it to converge over time, which weakens its interpretation as a “velocity” metric.

If ERV is not conceptually tied to exploitation, the finding that exploration and exploitation can co-improve becomes unsurprising. The paper’s framing around “beyond the exploration–exploitation tradeoff” could be overstated given an incorrect measurement of exploitation. Clarifying the relationship between ER/ERV and traditional exploration–exploitation trade-off would strengthen the contribution and avoid conceptual misuse in the field.

2. **Lack of justification for focusing on the final hidden layer:**
   As acknowledged by (Skean et al., 2025) and cited in this work, mid-layer representations often encode richer features. The decision to use only the final layer should be justified or complemented by additional analysis on intermediate layers.

3. **Insufficient clarity in VERL formulation:**
   The proposed VERL objective is spread across multiple subsections. Presenting the full training objective in consolidated equations would improve clarity and allow readers to better assess how ER, ERV, ERA interact with the base RL loss.

4. **Mixed empirical outcomes not sufficiently discussed:**
   While VERL shows large improvements on out-of-domain sets (e.g., +17.9 on CMATH and +21.4 on Gaokao2024_I), it also produces performance drops (−7.1, −1.0) for a different base model. These drop should be acknowledged and analyzed in the main text to present a balanced view.

**Questions:**

Consider citing *“Implicit Under-Parameterization Inhibits Data-Efficient Deep Reinforcement Learning”* (https://arxiv.org/abs/2010.14498), which proposes a different "effective rank" and connects it to representation diversity. How do you view the relationship between exploration and representational diversity in this context?

---

> ### Author Response · Authors · 2025-12-03
>
> Thank you for the review and for the constructive feedback! To address your concerns, we have: 1) Clarified our claim regarding the **definition of the exploitation metric (ERV)**; 2) Added additional results of discussion about layers; 3) Answered each of your questions; and 4)  Revised the paper with your comments.
>
> > Questionable definition of the exploitation metric (ERV) and its relation to the exploration–exploitation trade-off.
> >
>
> **Response (summary).**
>
> We agree with the reviewer that classical exploitation is defined via greedy choices w.r.t. a value function, and we do **not** claim that ERV is a direct estimator of greedy Q-values. Instead, our contribution is to define **semantic, response-level proxies** for exploration and exploitation in LLM reasoning. To avoid confusion with classical RL definitions and better reflect the nature of our metrics, we will explicitly rename these terms to **"Semantic Exploration"** and **"Semantic Exploitation"** throughout the revised paper. We will also soften the "beyond the exploration–exploitation trade-off" wording to ensure we do not overclaim regarding the classical dilemma.

---

> > ### Author Response · Authors · 2025-12-03
> >
> > ---
> >
> > **Details.**
> >
> > 1. From **action space** to **semantic trajectory space**: what level are we operating on?
> >
> >     We fully agree that in classical RL, exploitation means greedy choices w.r.t. an estimated value function, and exploration means deviating from that greedy policy. This is an **action-space** notion defined over $Q(s,a)$. However, in recent RLVR algorithms used for LLM post-training (e.g., PPO [1], GRPO [2]), the value function for each action is rarely an explicit tabular or network $Q(s,a)$: PPO relies on GAE as an approximation to $Q(s,a)$, and GRPO uses groupwise terminal returns as a trajectory-level estimate.
> >
> >     In this regime, prior RLHF/RLVR work [3–5] typically **implements** “exploitation” and “exploration” using *proxy quantities* such as token-level confidence or entropy, even though the ultimate objective is trajectory-level performance. Our work does *not* challenge the classical definition itself, but argues that these **token-level (action-space) proxies** are misaligned with how LLM reasoning actually unfolds: a single token (or even a word) rarely corresponds to a full semantic decision that “maximizes $Q(s,a)$”; instead, reasoning proceeds through multi-token **semantic chunks** (concepts, subgoals, partial derivations) [6,7].
> >
> >     For this reason, we move from the **action space** to the **semantic representation space of hidden states**, and from a single-step view to a **response-level** view. ER and ERV are defined at this semantic trajectory level, which is what we refer to by “Semantic Exploration” and “Semantic Exploitation” in the revision.
> >
> > 2. Clarification of   **Exploitation and  Exploration in the LLM reasoning**
> >     - **exploration**: measures how diverse a full response searching for reasoning paths.
> >     - **exploitation**: measures how the same trajectory refines the most promising known strategies along its path
> > 3. **Why ER and ERV can capture the Exploitation and  Exploration in the semantic space.**
> >
> >     We operate on the hidden-state sequence of the final transformer layer, which is widely understood to encode semantic content and model predictions [8–12].
> >
> >     - **ER (exploration proxy)**. Effective rank captures how widely a trajectory occupies different orthogonal semantic directions and how many **distinct semantic regions** it meaningfully visits, so we treat it as a representation-level measure of exploration.
> >     - **ERV (exploitation proxy).** ERV summarizes how ER changes along a single trajectory. With the running-history smoothing (**Def. 3.3**), a sustained and moderately positive ERV indicates that the model is steadily increasing semantic complexity within a locally stable region of the semantic space, which matches our view of exploitation as the average rate of **information gain** in that region.

---

> > ### Author Response · Authors · 2025-12-03
> >
> > 4. **Why do we use deviation from a historical average**
> >
> >     In practice, we first compute a first-order difference of ER and then center/normalize it using a running historical mean. As shown in **Def. 3.3**, this yields a smoothed velocity that puts more weight on recent ER changes while downweighting older ones. This construction serves two purposes:
> >
> >     - As shown in the **Def. 3.3**, ERV can be equivalently rewritten as a weighted sum of adjacent ER differences, where more recent ER changes receive larger weights, which encourages the model to focus on the currently evolving semantic region during reasoning.
> >     - It keeps ERV on the same growth order as ER (both scale as $\mathcal{O}(N)$ in sequence length for typical trajectories), which is suitable for scaling up.
> > 5. **How does this relate to the trade-off claim?**
> >
> >     Our claim "Beyond the Trade-off" refers specifically to the **semantic representation level**, not the action selection level.
> >
> >     - In **Action Space (Token level):** The trade-off remains (you cannot sample greedily and randomly at the same time).
> >     - In **Semantic Space (Representation level):** We find that RLVR allows models to achieve **both** higher semantic diversity (High ER) and rapid information gain (High ERV).
> >     This suggests that a capable reasoner can maintain a broad awareness of possibilities (Exploration) while simultaneously executing a focused, efficient deductive chain (Exploitation). We have revised the manuscript to ensure this distinction is precise and avoids overclaiming.
> >
> > [1] (PPO) Proximal Policy Optimization Algorithms. 2017 arxiv
> >
> > [2] (GRPO) DeepSeekMath: Pushing the Limits of Mathematical Reasoning in Open Language Models. 2024 arxiv
> >
> > [3] Token Hidden Reward: Steering Exploration-Exploitation in Group Relative Deep Reinforcement Learning. ICML, 2025
> >
> > [4] Optimizing Language Models for Inference Time Objectives using Reinforcement Learning. ICML, 2025
> >
> > [5] Reasoning with Exploration: An Entropy Perspective. AAAI, 2026
> >
> > [6] Chain-of-Thought Prompting Elicits Reasoning in Large Language Models. NeurIPS 2022
> >
> > [7] Tree of Thoughts: Deliberate Problem Solving with Large Language Models. ICML 2024
> >
> > [8] The geometry of hidden representations of large transformers. NeurIPS 2023
> >
> > [9] Semantics or spelling? Probing contextual word representations. ACL 2024
> >
> > [10] LinguaLens: Towards Interpreting Linguistic Mechanisms of Large Language Models via Sparse Auto-Encoder. EMNLP 2025
> >
> > [11] Are the Hidden States Hiding Something? Testing the Limits of Factuality Representations in LLMs. ACL 2025
> >
> > [12] Reasoning Models Know When They’re Right: Probing Hidden States for Self-Verification. COLM 2025

---

> ### Author Response · Authors · 2025-12-03
>
> > Lack of justification for focusing on the final hidden layer: As acknowledged by (Skean et al., 2025) and cited in this work, mid-layer representations often encode richer features. The decision to use only the final layer should be justified or complemented by additional analysis on intermediate layers.
> >
>
> **Response (summary).**
>
> We focus on the final hidden layer because our exploration/exploitation metrics are defined in the ***semantic* space** along the reasoning trajectory, and prior interpretability work [1,2] suggests that the last layers are most aligned with semantic meaning and model predictions. Empirically, using the final layer gives better performance than using an intermediate layer: for example, GRPO + VERL with the last layer improves the average pass@1 from **41.69%** (layer 14) to **43.44%**, and the average pass@k from **61.52%** (layer 14) to **64.40%**. This supports our design choice to base VERL on the final layer. We have added this analysis in **Appendix H.5**.
>
> ---
>
> **Details.**
>
> 1. **Semantic motivation for the last layer.**
>
>     We agree that intermediate layers can encode rich features. However, our notion of exploration and exploitation is explicitly defined in the *semantic space* of a reasoning trajectory. Existing interpretability studies [1,2] indicate that hidden states in the last layers are most tightly aligned with semantics and the model’s predictive distribution, while mid-layer representations tend to mix morphology, syntax, and other lower-level signals. For this reason, we consider the final layer more suitable for semantic diversity metrics such as ER and ERV.
>
> 2. **Intermediate-layer vs. final-layer VERL (pass@1).**
>
>     We implemented VERL on an intermediate layer (layer 14) and on the last layer, keeping everything else fixed. The pass@1 results are:
>
>     | Method | aime24 | aime25 | amc23 | amc24 | asdiv | carp_en | cmath | gaokao24_I | gaokao24_mix | gaokao_math_cloze | gsm8k | mawps | olympiadbench | svamp | tabmwp | Average |
>     | --- | --- | --- | --- | --- | --- | --- | --- | --- | --- | --- | --- | --- | --- | --- | --- | --- |
>     | Llama-3.2-3B-Instruct | 0.0 | 0.0 | 25.0 | 11.1 | 74.6 | 26.5 | 10.2 | 14.3 | 14.3 | 6.8 | 66.6 | 86.9 | 12.7 | 74.1 | 41.4 | 30.97 |
>     | GRPO | 3.3 | 0.0 | 27.5 | 8.9 | 88.8 | 45.0 | 28.3 | 21.4 | 20.9 | 23.7 | 80.7 | 96.0 | 16.7 | 87.7 | 71.7 | 41.37 |
>     | GRPO w/ VERL (layer = 14) | 10.0 | 0.0 | 27.5 | 11.1 | 88.6 | 43.6 | 30.7 | 21.4 | 16.5 | 22.0 | 81.9 | 95.5 | 18.1 | 87.0 | 71.4 | 41.69 |
>     | GRPO w/ VERL (layer = last) | 13.3 | 6.7 | 25.0 | 11.1 | 89.3 | 45.4 | 46.2 | 14.3 | 22.0 | 22.9 | 81.7 | 96.0 | 17.6 | 87.8 | 72.3 | **43.44** |
>
>     Both VERL variants improve over GRPO, but the last-layer version clearly gives the strongest overall gains.
>
> 3. **Intermediate-layer vs. final-layer VERL (pass@k).**
>
>     A similar pattern holds for pass@k:
>
>     | Method | math500@16 | amc23@128 | amc24@128 | aime24@256 | aime25@256 | Average |
>     | --- | --- | --- | --- | --- | --- | --- |
>     | Llama-3.2-3B-Instruct | 79.8 | 93.5 | 51.5 | 40.0 | 30.0 | 58.96 |
>     | GRPO | 80.2 | 95.4 | 60.6 | 40.0 | 30.0 | 61.24 |
>     | GRPO w/ VERL (layer = 14) | 81.0 | 92.9 | 57.0 | 40.0 | 36.7 | 61.52 |
>     | GRPO w/ VERL (layer = last) | 80.6 | 95.7 | 59.0 | 50.0 | 36.7 | **64.40** |
>
>     Again, using the last layer yields the best average improvement, especially on the more challenging AIME-style benchmarks.
>
>
> [1] LinguaLens: Towards Interpreting Linguistic Mechanisms of Large Language Models via Sparse Auto-Encoder. EMNLP 2025
>
> [2] Analyzing Encoded Concepts in Transformer Language Models. NAACL 2022

---

> ### Author Response · Authors · 2025-12-03
>
> > Insufficient clarity in VERL formulation
> >
>
> Thank you for your suggestion. We have reconstructed **Eq. 12** in **Sec. 5.3** in the reversed version to synthesize the optimization objectives of VERL.
>
> > Mixed empirical outcomes not sufficiently discussed
> >
>
> We appreciate the reviewer's suggestion to discuss these trade-offs. Specifically, the minor drop on CMATH (-1.0) with Qwen occurs at a high-performance saturation level (~91%), likely reflecting statistical variance rather than capability degradation. Meanwhile, the drop on Gaokao (-7.1) with Llama is attributable to the optimization dynamics specific to GRPO, as VERL achieves a substantial +7.2 gain on the exact same benchmark under the PPO setting (Row 6 in Tab. 1). Crucially, these isolated fluctuations are outweighed by the consistent improvements in Average accuracy across all models (e.g., +2.0% for Llama) and the significant breakthroughs on challenging OOD reasoning tasks (e.g., AIME24 +10.0%). This confirms that VERL's benefits in promoting robust reasoning significantly exceed the cost of minor local variance. We have added this analysis to the **Sec. 6.2** in the revision.
>
> > Consider citing “Implicit Under-Parameterization Inhibits Data-Efficient Deep Reinforcement Learning”, which proposes a different "effective rank" and connects it to representation diversity. How do you view the relationship between exploration and representational diversity in this context?
> >
>
> **Response (summary).**
>
> We thank the reviewer for this pointer and have added the citation in the revised version (Related Work, **Sec. D.3**). Conceptually, we see their “effective rank” as characterizing **capacity to preserve diversity in representations given exploratory data**, whereas in our setting effective rank is used directly as a **semantic exploration metric** over hidden-state trajectories in LLM reasoning. Both are about representational diversity, but they operate at different levels: *data / capacity* in their work vs. *reasoning-path semantics* in ours.
>
> ---
>
> **Details.**
>
> 1. **What “exploration” means in their setting.**
>
>     In *Implicit Under-Parameterization…*, “exploration” is essentially about how broadly the agent’s behavior covers the state–action space (i.e., data coverage). Their effective rank is computed on learned representations and used to show that, even under good behavioral exploration, the network’s representational rank can collapse due to bootstrapping. In other words, **rank is an outcome**: it measures how much of the exploration-induced diversity the model can retain.
>
> 2. **What does “exploration” mean in our setting.**
>
>     In our work, we are not studying exploration at the data-collection level, but at the **internal reasoning level** of an LLM given a fixed prompt. Here, “exploration” refers to the **semantic diversity of reasoning paths** within a single model: how many distinct semantic directions the hidden states along a chain-of-thought trajectory actually visit. Token-level entropy/confidence only reflects surface-level uncertainty over the vocabulary, so we move to the hidden-state space and measure **effective rank of the trajectory** itself as our exploration metric.

---

### Official Review · Reviewer_KPDY · 2025-10-31

**Soundness:** 2
**Presentation:** 2
**Contribution:** 2
**Rating:** 4
**Confidence:** 3

**Summary:**

This paper proposes reinterpreting the exploration–exploitation relationship in the reasoning of large language models (LLMs) within reinforcement learning by looking at the hidden state space. The authors use the hidden-layer Effective Rank (ER) and its first- and second-order derivatives (ERV, ERA) to measure the diversity of the semantic space and the speed of information gain. They find that exploration and exploitation can be decoupled at the hidden-state level, and thus propose a new method called Velocity-Exploiting Rank-Learning (VERL). VERL uses ERA to dynamically shape the advantage function, improving reasoning performance across different RL algorithms (GRPO/PPO). Experiments show that on several mathematical reasoning benchmarks (such as Gaokao 2024, AIME), VERL achieves improvements relative to GRPO and PPO.

**Strengths:**

The paper has a complete structure and is generally well-written.

**Weaknesses:**

1.	The main innovation of the paper is described as recharacterizing the exploration-exploitation trade-off through information sources in the hidden state space and proposing a new method, VERL, to improve the performance of different RL algorithms and large models. However, the paper claims to surpass traditional balance, but in fact, it only provides a more fine-grained way to adjust the ratio of exploration to exploitation, not a real reconstruction of the trade-off. A rename is suggested.
2.	The paper uses Effective Rank (ER) to quantify the semantic diversity of hidden states and introduces ER-derived quantities ERV and ERA to capture information-gain speed and acceleration, extending the notion of exploration metrics in RL. However, the core assumption that the singular value decomposition (SVD) of hidden states reflects exploration behavior lacks rigorous mathematical support; SVD reflects changes in the representation space’s dimensions, not necessarily exploration.
3.	VERL employs several hand-designed indicators, dynamic weights, and sigmoid-regulated adaptive parameters. While these work in experiments, their causal relationships are not clear, and it is not obvious which parts contribute what. Could the method be simplified? This coupling leads to weaker interpretability, so adding theoretical analysis is recommended.
4.	For long sequences or large models, performing frequent SVDs and multi-order statistics can be costly. The paper only mentions time cost in general and lacks detailed cost/efficiency comparisons, which affects the method’s usability.
5.	The main tables and curves report only single values or average curves, without multiple independent runs, means, standard deviations, or significance tests. This makes it hard to assess robustness and reliability. Significance testing data should be added.
6.	In ablation analysis, the separate contribution of internal VERL indicators such as ERV and ERA is insufficient, lacking experiments where removing one indicator (e.g., ERA) changes performance for comparison.
7.	In Fig. 5’s three groups of experiments (a), (b), (c), the pink curves (GRPO + ours (Full Version)) nearly coincide across all three subfigures; the data points, slopes, and inflection points align. Statistically, this is unlikely because different hyperparameters should cause slight fluctuations. The main text only states that performance is not sensitive to hyperparameters and does not explain why the three curves are almost identical. Please clarify: a)	Do the three curves come from the same experimental data reuse?  b)	If they are independent experiments, were seeds fixed, or were results averaged over multiple runs? c)	If they are example curves, should this be noted in the caption? If not clarified, this may affect reproducibility and credibility.

**Questions:**

Please refer to the “Weakness” section for related questions.

---

> ### Author Response · Authors · 2025-12-03
>
> Thank you for the review and for the constructive feedback! To address your concerns, we have: 1) Clarified our claim regarding the exploration–exploitation trade-off; 2) Added additional results of cost/efficiency comparisons and significance testing; 3) Answered each of your questions; and 4)  Revised paper with your comments.
>
> > However, the paper claims to surpass traditional balance, but in fact, it only provides a more fine-grained way to adjust the ratio of exploration to exploitation, not a real reconstruction of the trade-off. A rename is suggested.
> >
>
> **Response (summary).**
>
> We appreciate this comment and clarify that we do **not** claim to “solve” the classical exploration–exploitation dilemma **in action space**. Our contribution is to reframe how the trade-off is instantiated in RL for **LLM reasoning**: from **token-level, action space** proxies to **response-level, semantic-space metrics** (ER/ERV), which can be improved *simultaneously* by VERL. We have made this framing explicit in the paper.
>
> ---
>
> **Details.**
>
> 1. **From classical RL to RLHF/RLVR proxies.**
>
>     In classical RL, exploitation is defined via greedy choices in action space with respect to a value function, and exploration via deviations from that greedy policy. We agree with this definition. However, in modern RLHF/RLVR for LLMs [1,2], the value function for action is rarely an explicit tabular or network $Q(s,a)$: PPO relies on GAE, and GRPO treats groupwise terminal returns as trajectory-level estimates. In practice, prior work [3–5] already operationalizes “exploitation” and “exploration” using **proxy quantities for action** such as token-level confidence or entropy, rather than true greedy $Q$-values.
>
> 2. **Why do we move to response-level, semantic-space metrics?**
>
>     Our point is that these token-level proxies are misaligned with how reasoning actually happens. Solutions emerge over multi-token **semantic chunks** (concepts, subgoals), not isolated tokens. A single token cannot correspond to a meaningful “greedy decision” about a reasoning strategy. We therefore define exploration/exploitation at the **response-level in semantic space**:
>
>     - **ER** measures how broadly a hidden-state trajectory spreads across semantic directions → semantic-space exploration.
>     - **ERV** measures how the same trajectory refines semantic content along its path → semantic-space exploitation.
> 3. **Why do we say “beyond” token-level trade-offs?**
>
>     When exploration and exploitation are measured via ER/ERV on hidden-state trajectories, we find that:
>
>     - ER and ERV are **deoupled** (experimental analysis in **Sec. 3** and theoretical analysis in **Appendix. F.1**), unlike many token/entropy-based metrics, where increasing one tends to hurt the other.
>     - VERL can **simultaneously** increase ER (semantic breadth) and ERV (on-trajectory refinement), as shown in our main GRPO/PPO experiments (**Sec. 4**), rather than simply tuning a single knob between them.
>
> [1] (PPO) Proximal Policy Optimization Algorithms. 2017 arxiv
>
> [2] (GRPO) DeepSeekMath: Pushing the Limits of Mathematical Reasoning in Open Language Models. 2024 arxiv
>
> [3]  Reasoning with Exploration: An Entropy Perspective. AAAI 2026
>
> [4] Token Hidden Reward: Steering Exploration–Exploitation in Group Relative Reinforcement Learning. ICML 2025
>
> [5] The Unreasonable Effectiveness of Entropy Minimization in LLM Reasoning. NeurIPS 2025

---

> ### Author Response · Authors · 2025-12-03
>
> > However, the core assumption that the singular value decomposition (SVD) of hidden states reflects exploration behavior lacks rigorous mathematical support; SVD reflects changes in the representation space’s dimensions, not necessarily exploration.
> >
>
> **Response (summary).**
>
> In our setting, SVD becomes meaningful because transformer hidden states are known to encode *semantic* structure. We rely on prior interpretability work [1–5] showing that LLM hidden states form semantic representations, treat the hidden-state sequence as a semantic trajectory, and use the effective rank of this trajectory as a **representation-level exploration metric**. We have clarified this semantic-space interpretation in the main text and added a more formal **mathematical support** in the **Appendix. F.2**.
>
> ---
>
> **Details.**
>
> 1. **Semantic nature of hidden states.**
>
>     Prior work [1–5] has shown that hidden states in large transformers encode **interpretable semantic information** about tokens and intermediate thoughts, rather than being arbitrary internal vectors. This includes evidence that:
>
>     - Different semantic concepts and factual attributes are linearly separable in the hidden space
>     - Later-layer representations align closely with token meaning and the model’s predictions
>     - Reasoning-related properties (e.g., self-verification, confidence) can be probed directly from hidden states
> 2. **From SVD of hidden states to a semantic exploration metric.**
>
>     Given a reasoning trajectory, we collect its hidden states into a matrix $\mathbf{Z}$. The SVD of  $\mathbf{Z}$ decomposes the variance across orthogonal directions in this semantic space. The singular values describe how much “energy” each semantic direction carries; the **effective rank** then measures how many directions have a non-negligible contribution. Intuitively:
>
>     - If the model revisits only a narrow band of semantic patterns, energy concentrates on a few directions → low ER;
>     - If it explores more alternative concepts and intermediate hypotheses, energy spreads over more directions → higher ER.
> 3. **Additional mathematical support in the revision.**
>
>     In the revised version, we have added **Proposition F.5** that formalizes this connection: under mild assumptions on the semantic basis and trajectory smoothness, effective rank increases when the model activates additional approximately orthogonal semantic directions, and remains stable when it only rescales existing directions. We have explicitly pointed the reader to **Appendix. F.2**.
>
>
> [1] The geometry of hidden representations of large transformers. NeurIPS 2023
>
> [2] Semantics or spelling? Probing contextual word representations. ACL 2024
>
> [3] LinguaLens: Towards Interpreting Linguistic Mechanisms of Large Language Models via Sparse Auto-Encoder. EMNLP 2025
>
> [4] Are the Hidden States Hiding Something? Testing the Limits of Factuality Representations in LLMs. ACL 2025
>
> [5] Reasoning Models Know When They’re Right: Probing Hidden States for Self-Verification. COLM 2025

---

> ### Author Response · Authors · 2025-12-03
>
> > VERL employs several hand-designed indicators, dynamic weights, and sigmoid-regulated adaptive parameters. While these work in experiments, their causal relationships are not clear, and it is not obvious which parts contribute what. Could the method be simplified? This coupling leads to weaker interpretability, so adding theoretical analysis is recommended.
> >
>
> **Response (summary).**
>
> We agree that interpretability is important. In the revision, we make it clearer that VERL consists of three conceptually simple pieces—ER/ERV/ERA as indicators, a single ERA-controlled mixing weight, and standard squashing functions used only for normalization. We support their roles with **(i) empirical ablations** and **(ii) theoretical propositions on how these signals affect the adjusted advantages.** We have explicitly discussed how VERL could be simplified and which parts are most essential in **Sec. 6.3**.
>
> ---
>
> **Details.**
>
> 1. **On the indicators (ER, ERV, ERA).**
>
>     We are not using arbitrary hand-crafted signals; each indicator has a specific role backed by both intuition and measurements. In the paper, we already:
>
>     - show how ER/ERV correlate with token-level entropy and rewards (**Sec.1 & Sec. 3**),
>     - and provide **Proposition. 3.5** that formalize how changes in these metrics affect the adjusted advantage used by RL (e.g., monotonicity and scaling properties).
> 2. **On the dynamic weight controlled by ERA.**
>
>     As discussed in **Sec. 5.2**, ERA is used as a **single predictive signal** to interpolate between ER (more exploration) and ERV (more exploitation). This is not a large tangle of heuristics, but one scalar meta-controller. In the revised version, we:
>
>     - point to the corresponding proposition/analysis showing how this adaptive mixing changes the effective advantage distribution,
>     - and summarize the intuition directly next to the formula, instead of scattering it.
> 3. **On sigmoid.**
>
>     The sigmoid plays a purely **numerical** role: it squashes the ERA-driven weight into [0,1] so that ER and ERV can be combined in a stable and interpretable convex mixture.

---

> ### Author Response · Authors · 2025-12-03
>
> > For long sequences or large models, performing frequent SVDs and multi-order statistics can be costly.
> >
>
> **Response (summary).**
>
> VERL is implemented without full SVD: we work on Gram matrices with incremental updates, which reduces the worst-case complexity from  $\mathcal{O}(T^2)$ to $\mathcal{O}(T)$ in sequence length. In practice, on 4×RTX 4090 with Llama-3.2-3B-Instruct and a long max response length of 4096 tokens, VERL improves evaluation metrics by **34.21%** while adding only **3.13%** wall-clock overhead and **2.31%** extra GPU memory. We have made these efficiency details explicit in **Appendix. G2 and G.3**.
>
> ---
>
> **Details.**
>
> 1. **No full SVD in practice; incremental Gram updates.**
>
>     VERL does *not* compute a full SVD on hidden states at each step. By definition, effective rank depends only on singular values, so in our implementation we:
>
>     - First compute the **Gram matrix $G$** of the hidden states $\mathbf{Z}$, which is fast with GPUs
>     - Then obtain singular values from the eigenvalues of $G$ , which is mathematically equivalent but substantially cheaper than a full SVD on $\mathbf{Z}$.
>
>     For ERV/ERA, we additionally exploit the **incremental structure** of the Gram matrix across strides, reusing previous $G$ and updating it instead of recomputing from scratch. The implementation details are described in **Appendix F.2**.
>
> 2. **Theoretical complexity comparison.**
>
>     As we show in **Appendix. G2**, in the worst-case situation where the sequence length $T$ exceeds the hidden dimension $D$, and both $D$ and the stride $s$ can be treated as constants:
>
>     - The naïve approach that reconstructs matrices independently at each stride has a matrix-construction cost scaling as $\mathcal{O}(T^2)$,
>     - VERL’s incremental Gram/covariance updates scale as $\mathcal{O}(T)$.
>
>     So asymptotically, our implementation has a **strictly better dependency on $T$** than a naïve SVD-based design.
>
> 3. **Practical cost under multi-GPU training.**
>
>     Under modern multi-GPU training, the extra overhead is small:
>
>     - Constructing the Gram matrix is a single **GEMM** call, heavily optimized in BLAS/cuBLAS and typically cheaper than the bidiagonalization stage of a full SVD.
>     - Working with the symmetric positive-definite Gram matrix lets us use fast Hermitian eigenvalue routines (`torch.linalg.eigvalsh`) instead of `torch.linalg.svdvals`.
>
>     On **4×RTX 4090 (48GB)** with **Llama-3.2-3B-Instruct** and **max response length = 4096** (longer than the hidden dimension 3072), we benchmarked GRPO vs. GRPO+VERL:
>
>     | Method | Train score avg | Train gain (%) | Val score avg | Val gain (%) | sat\_math@1 | math@1 | gaokao24\_II@1 | amc24@1 | Eval gain mean (%) | Time (s) | Time overhead (%) | GPU mem (GB) | Mem overhead (%) |
>     | --- | --- | --- | --- | --- | --- | --- | --- | --- | --- | --- | --- | --- | --- |
>     | GRPO | 0.4196 | – | 0.2127 | – | 34.4 | 51.2 | 21.4 | 11.1 | – | 321.04 | – | 190.1 | – |
>     | GRPO + VERL | 0.4333 | 3.26 | 0.2214 | 4.13 | 43.8 | 52.3 | 35.7 | 15.6 | **34.21** | 331.08 | **3.13** | 194.5 | **2.31** |
>
>     VERL thus delivers **substantial evaluation gains** at **very modest** extra time and memory cost.

---

> ### Author Response · Authors · 2025-12-03
>
> > Significance testing data should be added.
> >
>
> **Response (summary).**
>
> We agree that robustness is important. During the rebuttal period, we ran **five independent seeds** for Llama-3.2-3B-Instruct on all main benchmarks. Compared to GRPO, GRPO+VERL improves the **average pass@1** from `39.24 ± 0.72` to `45.18 ± 0.38` and the **average pass@k** from `59.87 ± 1.37` to `62.78 ± 0.63`. Welch’s t-tests at 0.05 significance level show that most of these gains are statistically significant (marked with `*`), indicating that the improvements are not due to random seed variation.
>
> ---
>
> **Details.**
>
> 1. **Main results: pass@1.**
>
>
>     | Method | aime24_pass@1 | aime25_pass@1 | amc23_pass@1 | amc24_pass@1 | asdiv_pass@1 | carp_en_pass@1 | cmath_pass@1 | gaokao2024_I_pass@1 | gaokao2024_mix_pass@1 | gaokao_math_cloze_pass@1 | gsm8k_pass@1 | mawps_pass@1 | olympiadbench_pass@1 | svamp_pass@1 | tabmwp_pass@1 | AVERAGE_PASS1 |
>     | --- | --- | --- | --- | --- | --- | --- | --- | --- | --- | --- | --- | --- | --- | --- | --- | --- |
>     | GRPO | 10.00 ± 0.00 | 0.00 ± 0.00 | 17.50 ± 0.00 | 11.54 ± 0.98 | 88.62 ± 0.54 | 44.82 ± 0.04 | 15.34 ± 0.36 | 5.72 ± 7.83 | 19.58 ± 2.95 | 22.20 ± 1.82 | 80.88 ± 0.60 | 95.60 ± 0.00 | 17.80 ± 0.85 | 87.08 ± 0.22 | 71.92 ± 0.41 | 39.24 ± 0.72 |
>     | GRPO+VERL | 12.64 ± 1.48* | 3.98 ± 1.52* | 25.50 ± 1.12* | 16.00 ± 2.46* | 89.50 ± 0.14* | 45.20 ± 0.14* | 56.62 ± 0.93* | 25.72 ± 3.94* | 18.92 ± 1.97 | 24.04 ± 0.76 | 83.00 ± 0.59* | 96.32 ± 0.16* | 18.30 ± 0.71 | 87.30 ± 0.28 | 74.64 ± 0.75* | **45.18 ± 0.38*** |
>
>     Most benchmarks show statistically significant gains (marked `*`) in pass@1 when adding VERL to GRPO; on the few tasks without `*`, the mean still improves, but the variance across seeds is slightly larger.
>
> 2. **Main results: pass@k.**
>
>
>     | Method | math500_pass@16 | amc23_pass@128 | amc24_pass@128 | aime24_pass@256 | aime25_pass@256 | AVERAGE_PASSK |
>     | --- | --- | --- | --- | --- | --- | --- |
>     | GRPO | 80.40 ± 0.00 | 90.78 ± 0.18 | 56.18 ± 1.50 | 39.34 ± 1.48 | 32.66 ± 5.95 | 59.87 ± 1.37 |
>     | GRPO+VERL | 80.52 ± 0.18 | 94.22 ± 1.01* | 58.46 ± 1.69 | 43.98 ± 3.66* | 36.70 ± 0.00 | **62.78 ± 0.63*** |
>
>     On the harder pass@k settings, GRPO+VERL again outperforms GRPO, with statistically significant average improvement (p < 0.05) over all listed tasks (as reflected in `AVERAGE_PASSK`).

---

> ### Author Response · Authors · 2025-12-03
>
> > In ablation analysis, the separate contribution of internal VERL indicators such as ERV and ERA is insufficient, lacking experiments where removing one indicator (e.g., ERA) changes performance for comparison.
> >
>
> **Response (summary).**
>
> In fact, in **Sec. 6.3** in first version of our paper, we **already include** ablations where **(i)** ERA is removed from the adaptive mixing and **(ii)** ER and ERV are used separately: using ERA to adapt the mixing weight yields clearly better performance on OOD and hard problems (**Tab. 1**), and combining ER + ERV outperforms ER-only and ERV-only variants (**Fig. 5(c)**), showing that each indicator provides a distinct benefit.
>
> ---
>
> **Details.**
>
> 1. **Role of ERA.**
>
>     In **Table 1**, we explicitly compare:
>
>     - a variant of VERL that **does not** use ERA to adapt the mixing weight $\beta$, and
>     - the full VERL with ERA-driven $\beta$.
>
>     When ERA is enabled, the VERL-augmented policy achieves **consistently higher performance on OOD and hard benchmarks** than the no-ERA variant. This demonstrates that the second-order trend signal (ERA) is not redundant: it helps the policy allocate more weight to exploration or exploitation at the right time and yields tangible gains, rather than being a purely cosmetic addition.
>
> 2. **Role of ER vs. ERV.**
>
>     In **Figure 5(c)**, we report training-score curves for: **ER-only, ERV-only**, and **ER + ERV**.
>
>     The results show that:
>
>     - ER-only and ERV-only variants both improve over the base RL algorithm but **plateau earlier** or exhibit less stable learning,
>     - The **combined ER + ERV** version achieves **consistently better training curves**, supporting our claim that ER and ERV capture **complementary aspects** of representation-level exploration and exploitation.

---

> ### Author Response · Authors · 2025-12-03
>
> > In Fig. 5’s three groups of experiments (a), (b), (c), the pink curves (GRPO + ours (Full Version)) nearly coincide across all three subfigures… Please clarify whether these curves reuse the same data, and how they were generated.
> >
>
> **Response (summary).**
>
> Yes, the three pink curves are **the same run reused intentionally** as a fixed GRPO+VERL full-reference curve; they are not three independent experiments. We have stated this clearly in the caption of **Fig. 5** and main text to avoid any confusion about reproducibility.
>
> ---
>
> **Details.**
>
> 1. **Are the three pink curves the same data?**
>
>     Yes. The pink curve in Fig. 5(a), (b), and (c) is **intentionally reused**. It corresponds to a single GRPO + VERL full configuration with our default hyperparameters and a fixed seed. Each subfigure then varies a different component (e.g., stride, κ, etc.) against this **shared reference**.
>
> 2. **Why reuse instead of re-running three times?**
>
>     Post-training LLMs is computationally expensive, so:
>
>     - we chose one representative full VERL run as a **common baseline**,
>     - and plotted all ablated variants against this same trajectory.
>
>     This design makes the comparisons more direct (every ablation is visually read “against the same curve”) and avoids tripling the cost of full VERL training just for plotting.

---

### Author Response · Authors · 2025-12-03
**Global response**

We thank the AC and all reviewers for their constructive feedback. Our paper revisits how “exploration vs. exploitation” should be instantiated in **RLVR for LLM reasoning** by moving from token-level proxies in action space to **trajectory-level** metrics on hidden states from **semantic space**, and introducing VERL, a practical module that uses these metrics to improve RL post-training; we do not change the classical action-space dilemma, but show that at the semantic representation level exploration (ER) and exploitation (ERV) can **be decoupled and improved jointly**.

---

**Contributions**

- **Problem.** Prior RLHF/RLVR work measures “exploration/exploitation” via **token-level entropy/confidence**, i.e., in **action space**, which is misaligned with reasoning that unfolds over multi-token **semantic chunks** and hidden-state trajectories. This makes the perceived trade-off partly an artifact of the measurement level.
- **Semantic metrics.** We work in the **final-layer hidden-state space** and define:
    - **ER** – Effective Rank of the trajectory → **Semantic Exploration** (how many orthogonal semantic directions are meaningfully occupied).
    - **ERV** – smoothed first-order difference of ER → **Semantic Exploitation** (rate of semantic information gain along a locally stable trajectory).
    - **ERA** – second-order trend of ER → whether information gain is accelerating or saturating.

    Theoretically and empirically, we prove that ER(final) and ERV are not functionally dependent (Sec. 4 & Appendix F.1); moreover, ER increases when new approximately orthogonal semantic directions are activated (Sec. 4 & Appendix F.2), while ERV tracks whether ER is steadily growing or saturating along the trajectory (Sec. 4 & Appendix F.3), so together they capture genuinely different aspects of semantic behavior.

- **VERL.** We propose **VERL**, a lightweight advantage-shaping module for PPO/GRPO that:
    - Uses **ERA** to control a **single mixing weight** between ER (exploration) and ERV (exploitation).
    - Adds this mixture as a **clipped positive bonus** on top of the base advantage (Sec. 5.2–5.3, rewritten Eq. 12).
    - Computes ER/ERV/ERA via **Gram matrices + eigenvalues with incremental updates**, avoiding full SVD (Appendix G.2).

---

**Key results**

- **Accuracy & robustness.**
    - On broad math/logic benchmarks, VERL consistently improves **Average Pass@1** and **Pass@k** over both PPO and GRPO for **Llama-3.2-3B-Instruct** and **Qwen2.5-7B** (Sec. 6.2, Tab. 1–2).
    - New **5-seed** experiments for Llama-3.2-3B-Instruct show Average Pass@1 improving from `39.24 ± 0.72` → `45.18 ± 0.38` and Average Pass@k from `59.87 ± 1.37` → `62.78 ± 0.63`, with most gains statistically significant (Welch’s t-test, p < 0.05).
- **Comparison to existing exploration strategies.**
    - Under the same GRPO pipeline, **GRPO + VERL** outperforms token-level regularizers (“old exploit regu”, “old explore regu”, and their combination) in both Average Pass@1 and Pass@k, and on several hard math benchmarks.
- **Efficiency.**
    - **Theory:** When sequence length $T$ is larger than dimension $D$, $D$ and stride $s$ are treated as constants, incremental Gram updates reduce the matrix-construction cost from  $\mathcal{O}(T^2)$ to  $\mathcal{O}(T)$ (Appendix G.2).
    - **Practice:** On 4×RTX 4090 with Llama-3.2-3B-Instruct and max response length 4096, VERL yields an evaluation gain of ≈**34%** while adding only **3.13%** wall-clock and **2.31%** memory overhead.

---

**Main revisions**

- **Clarified scope (Intro, Sec. 3, 5).** Clearly distinguish classical action-space exploitation (greedy w.r.t. Q) from our **semantic** exploration/exploitation; adopt the terms “Semantic Exploration/Exploitation” and soften “beyond the exploration–exploitation trade-off” to emphasize we operate at the representation level.
- **Theory (Appendix F.2 & F.3).** Add proofs that ER (final) and ERV are structurally decoupled and that ER increases when new semantic directions are activated, making explicit that our metrics capture different aspects of hidden-state behavior rather than approximating greedy Q-values.
- **Algorithm & ablations (Sec. 5.3 & 6.2).** Rewrite the VERL objective to separate base advantage, semantic bonus, and clipping; highlight ablations removing ERA or using ER-only/ERV-only, showing the full ER+ERV+ERA design performs best; clarify that the same full VERL run is reused as the pink reference curve in Fig. 5.
- **Cost & baselines (Appendix G.2 & G.3 & H.5).** Make explicit our incremental Gram + eigvalsh implementation (no full SVD), add worst-case complexity and time/memory tables, include 5-seed significance tests, comparisons to entropy/confidence-based regularizers, and an analysis of intermediate vs. final-layer VERL (final layer performs best).
- **Reproducibility.** Fully release the implementation in the anonymous repository.

---

### Note · Authors · 2025-12-29

I have read and agree with the venue's withdrawal policy on behalf of myself and my co-authors.